# TMBIM6/BI-1 contributes to cancer progression through assembly with mTORC2 and AKT activation

Hyun-Kyoung Kim [1], Kashi Raj Bhattarai [1], Raghu Patil Junjappa [1], Jin Hee Ahn [2], Suvarna H. Pagire[2], Hyun Ju Yoo [3], Jaeseok Han[4], Duckgue Lee [4], Kyung-Woon Kim [5], Hyung-Ryong Kim [6 ✉] & Han-Jung Chae [1 ✉]

Transmembrane B cell lymphoma 2-associated X protein inhibitor motif-containing (TMBIM) 6, a $Ca^{2+}$ channel-like protein, is highly up-regulated in several cancer types. Here, we show that TMBIM6 is closely associated with survival in patients with cervical, breast, lung, and prostate cancer. TMBIM6 deletion or knockdown suppresses primary tumor growth. Further, mTORC2 activation is up-regulated by TMBIM6 and stimulates glycolysis, protein synthesis, and the expression of lipid synthesis genes and glycosylated proteins. Moreover, ER-leaky $Ca^{2+}$ from TMBIM6, a unique characteristic, is shown to affect mTORC2 assembly and its association with ribosomes. In addition, we identify that the BIA compound, a potentialTMBIM6 antagonist, prevents TMBIM6 binding to mTORC2, decreases mTORC2 activity, and also regulates TMBIM6-leaky $Ca^{2+}$, further suppressing tumor formation and progression in cancer xenograft models. This previously unknown signaling cascade in which mTORC2 activity is enhanced via the interaction with TMBIM6 provides potential therapeutic targets for various malignancies.

[1] Department of Pharmacology and New Drug Development Research Institute, Jeonbuk National University Medical School, Jeonju 54896, Republic of Korea. [2] Department of Chemistry, Gwangju Institute of Science and Technology, Gwangju 61005, Republic of Korea. [3] Department of Convergence Medicine, Asan Institute for Life Sciences, Asan Medical Center, University of Ulsan College of Medicine, Seoul 05505, Republic of Korea. [4] Soonchunhyang Institute of Med-bio Science (SIMS), Sooncynhyang University, Cheonan-si 31151, Republic of Korea. [5] Animal Biotechnology Division, National Institute of Animal Science, Rural Development Administration (RDA), Wanju-gun, Jeonbuk 54875, Republic of Korea. [6] College of Dentistry, Dankook University, Cheonan 31116, Republic of Korea. ✉email: hrkimdp@gmail.com; hjchae@jbnu.ac.kr

Transmembrane B cell lymphoma 2-associated X protein (BAX) inhibitor motif-containing (TMBIM)6, an inhibitor of ER stress[1], was initially named BAX inhibitor (BI)-1[2]. BI-1/TMBIM6 is now known to be a member of the transmembrane BI-1 motif-containing family of proteins[3]. TMBIM6 is a $Ca^{2+}$ channel-like protein that lowers the steady-state $[Ca^{2+}]_{ER}$, which is expressed in the endoplasmic reticulum (ER) membrane surface[1]. It is up-regulated in many cancer types including breast, lung, prostate, nasopharyngeal, and liver cancer[4–8]. In our previous study, the overexpression of TMBIM6 promoted cancer metastasis by regulating cell mobility and invasiveness and glucose metabolism[9]. Suppression of TMBIM6 leads to cell death, which results in reduced tumor development. Recently, we reported that Sp1 and PKC regulate the transcriptional expression of TMBIM6[8]. PKC was also highly expressed in various cancer types, as shown in liver, prostate, and breast cancers[10,11]. Although many pieces of evidence supported the involvement of TMBIM6 in the development of various cancers, the molecular mechanism underlying the role of TMBIM6 in cancer progression has been less studied.

Mechanistic target of rapamycin (mTOR) plays a key role in cellular metabolism, cell growth, and nutrient sensing[12]. mTOR complex (mTORC)1 and mTORC2 are two structurally and functionally distinct protein complexes and are frequently overexpressed in cancer and diabetes[13–16]. The former consists of mTOR, regulatory-associated protein of mTORC1 (RAPTOR), and mammalian lethal with SEC13 protein (mLST)8, and is regulated by the presence of amino acids[17,18]; amino acid transporters; Rag, Rheb, and Rab GTPases; and GATOR[19,20]. mTORC2—which comprises mTOR, RICTOR, SIN1, and mLST8—plays an important role in the control of metabolism as part of the PI3K/AKT signaling pathway[21]. mTOR and mLST8 are the core components of both complexes, whereas RAPTOR and RICTOR are unique to mTORC1 and mTORC2, respectively. Dual phosphorylation of SIN1 at Threonine 86 and 398 negatively regulates mTORC2 activation[22]. Protein observed with RICTOR-1/2 and DEP domain-containing mTOR-interacting protein also bind to mTORC2[23,24].

The AGC kinase family members AKT, SGK1, and PKCα are major substrates of mTORC2[25–27]. mTORC2-mediated phosphorylation of AKT (pAKT-S473) of the hydrophobic motif—a hallmark of mTORC2 activation—is induced by stimulation with insulin or other growth factors in serum-starved cells. The constitutive phosphorylation of AKT (pAKT-T450) in the turn motif, which prevents co-translational AKT ubiquitination, maintains the basal activity of mTORC2[28], whereas the phosphorylation of AKT (pAKT-S473) regulated by upstream signaling. The association of mTORC2 with ribosomes by enhanced PI3K signaling increases AKT activation[29]. However, although the mechanism underlying mTORC1 activation has been widely studied, the activation of mTORC2 is not fully understood. For example, the Tel2-Tti1-Tti2-RuvB-like 1/2 complex directly regulates the assembly of mTORC1 based on the metabolic state of the cell[30], whereas mTORC2 assembly has yet to be elucidated. Moreover, mTORC1 is mainly localized in lysosomes, while mTORC2 is present in the ER or plasma membrane. Several studies have demonstrated that mTORC2 may associate with the ER via interaction with ER-bound ribosomes[29,31], although the molecular details remain unclear. We speculate that the ER membrane harbors specific proteins that are involved in mTORC2 activation.

In the present study, we identify TMBIM6 as a major binding partner of mTORC2 at the ER membrane. Our results provide evidence for mTORC2–TMBIM6–ribosome axis that regulates AKT activation and tumorigenesis.

## Results

**TMBIM6 expression increases in tumor samples.** To investigate the oncogenic role of TMBIM6 in cancer progression, we first analyzed *TMBIM6* mRNA expression profiling datasets of multiple tumor samples from the NCBI/GEO. These analyses revealed that TMBIM6 significantly overexpressed in fibrosarcoma, cervical, endometrial and vulvar, breast, lung, and prostate cancers (Fig. 1a–e). Next, we compared the expression levels of TMBIM6 in same cancer tissues using tissue microarrays and obtained the similar results (Fig. 1f). To further examine whether the TMBIM6 expression level in tumors is associated with prognosis, we analyzed the correlations between TMBIM6 expression and overall survival (OS) using GEPIA2 from the TCGA and the GTEx projects[32] and OncoLnc from the TCGA[33]. We found that patients with high TMBIM6 expression had poor survival in breast invasive carcinoma (BRCA), cervical squamous cell carcinoma and endocervical adenocarcinoma (CESC), sarcoma (SARC), and lung adenocarcinoma (LUAD) (Fig. 1g, Supplementary Fig. 1A). In addition, we confirmed OS in several cancers including pancreatic adenocarcinoma, esophageal carcinoma, skin cutaneous melanoma, head and neck squamous cell carcinoma, and brain lower-grade glioma (Supplementary Fig. 1B). These data suggest that TMBIM6 has a potential clinical value as a predictive biomarker for disease outcome in several cancers.

Next, we generated TMBIM6 knockout (KO) cells in the HT1080 and HeLa cell line (TMBIM6 KO) by using CRISPR/Cas9 technology (Supplementary Fig. 2). We analyzed expression profiles in WT and TMBIM6 KO HT1080 cells by microarray and selected Gene Ontology related to cancer characteristics on Quick GO (https://www.ebi.ac.uk/QuickGO/) supplied at EMBL-EBI. There were several differentially expressed genes (DEGs) in TMBIM6 KO HT1080 cells compared with WT cells (Fig. 1h, Supplementary Data 1), and most of the DEGs related to apoptotic process, migration, proliferation, and metabolic pathways were decreased (Fig. 1i, j). On the other hand, TMBIM6-overexpressing HT1080 cells showed upregulation of genes related to cancer progression and metastasis (Supplementary Fig. 1C–E). Thus, TMBIM6 may be an important regulator of cancer-related signaling.

**TMBIM6 depletion suppresses the tumorigenicity of cancer.** To validate the above results, we performed cell proliferation, migration, and invasion assay. TMBIM6 KO HT1080, HeLa cells, and mouse embryonic fibroblasts (MEFs) both exhibited slow growth relative to WT cells (Fig. 2a), which was restored in TMBIM6 KO cells with re-expressing TMBIM6 (Supplementary Fig. 3A, B). Cell migration and invasion were inhibited in cells lacking TMBIM6 (Fig. 2b, c, Supplementary Fig. 3C, D). To investigate the role of TMBIM6 in the growth of tumor cells in animals, we subcutaneously injected TMBIM6 WT and KO HT1080 cells into the left and right flanks of immunocompromised mice (Supplementary Fig. 3E). Tumor formation and the weight of tumors originating from TMBIM6 KO HT1080 cells was significantly reduced compared with that in WT cells (Fig. 2d–f). Immunohistochemistry staining of Ki67-positive proliferative cells showed a significant decrease in xenografts from TMBIM6 KO cells (Fig. 2g). Consistently, tumor formation and weight, and the expressions of Ki-67 was apparently reduced in TMBIM6 KO HeLa cells than WT cells (Fig. 2h–k, Supplementary Fig. 3F). In addition, tumor formation as well as Ki67 expression were reduced in TMBIM6 knockdown by injection of self-assembled micelle inhibitory RNA (SAMiRNA), a stable siRNA silencing platform for efficient in vivo targeting of genes[34] (Supplementary Fig. 3G–L). Taken together, these in vitro and

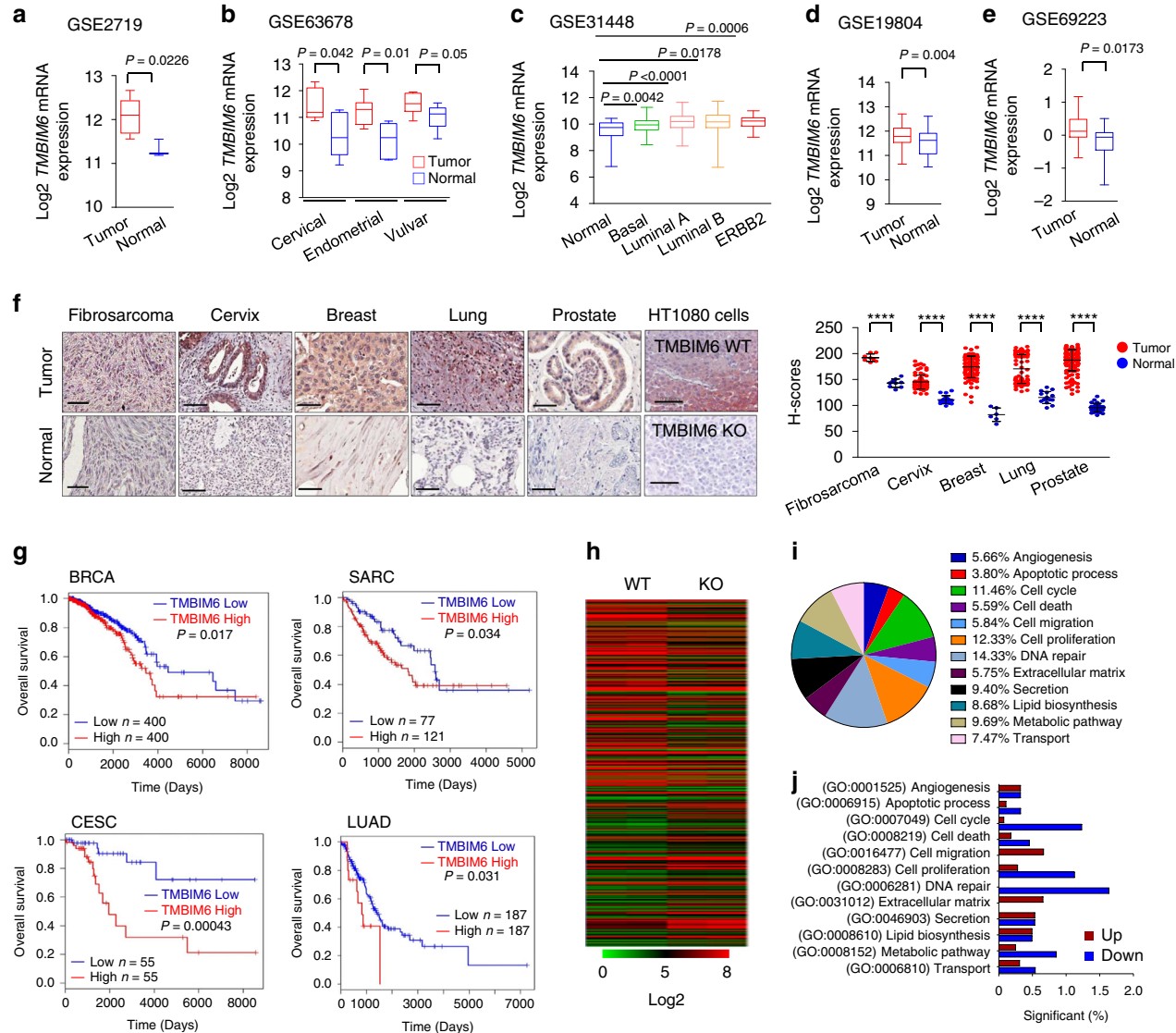

**Fig. 1 TMBIM6 expression increased in cancer patient samples. a–e** TMBIM6 expression was analyzed using the GEO database from NCBI. Fibrosarcoma (GSE2719; normal $n = 3$; tumor $n = 7$), cervix (GSE63678; cervical normal $n = 5$; tumor $n = 5$; endometrial normal $n = 5$; tumor $n = 7$; vulvar normal $n = 7$; tumor $n = 6$), breast (GSE31448; normal $n = 31$; basal $n = 98$; luminal A $n = 89$; luminal B $n = 49$; ERBB2 $n = 25$), lung (GSE19804; normal $n = 60$; tumor $n = 60$), and prostate (GSE69223; normal $n = 15$; tumor $n = 15$) datasets are presented. Center line of the box represents median; box bounds represent 25th and 75th percentiles; whiskers represent minimum and maximum values. **f** Representative immunohistochemical staining of TMBIM6 on tissue microarrays containing fibrosarcoma, cervix, breast, lung, and prostate tissue and adjacent normal tissues. TMBIM6 WT and KO HT1080 cells were used as a control for validation of the method. Right; quantification data of TMBIM6 expression (fibrosarcoma normal $n = 9$; tumor $n = 8$; cervix normal $n = 20$; tumor $n = 80$; breast normal $n = 6$; tumor $n = 97$; lung normal $n = 15$; tumor $n = 75$; prostate normal $n = 32$; tumor $n = 160$). Scale bar, 100 μm. (brown: positive antibody staining, blue: hematoxylin for nuclear staining). Data are presented as means ± SD. ****$p < 0.0001$, two-tailed unpaired $t$-test. **g** Kaplan–Meier curves showing the overall survival analysis in patients with high and low expression of TMBIM6 using GEPIA2 tool. $P$ value with log-rank analysis. BRCA breast invasive carcinoma, CESC cervical squamous cell carcinoma and endocervical adenocarcinoma, SARC sarcoma, LUAD lung adenocarcinoma. **h** Differentially expressed genes by microarray analysis of mRNA expression levels in TMBIM6 KO and WT HT1080 cells. **i** Significant ratios in TMBIM6 KO and WT HT1080 cells determined by Gene Ontology analysis. **j** The graph indicates significant differences in downregulation and upregulation of the indicated category genes in the TMBIM6 KO cells compared with those in TMBIM6 WT cells.

in vivo experiments demonstrate that TMBIM6 regulates tumor growth.

**TMBIM6 activates AKT pathway through mTORC2-ribosome axis.** To evaluate the signaling protein molecule, which regulates the cancer progression in WT and TMBIM6 KO HT1080 cells, we performed protein phospho-kinase profiling analysis. The results showed that the phosphorylation of AKT (pAKT-S473), PRAS40, mTOR, GSK3-α/β, and WNK1 were decreased in TMBIM6 KO HT1080 cells (Fig. 3a). Since PRAS40, GSK3-α/β, and WNK1 are

the known substrates of AKT, we next investigated whether mTORC2, upstream regulator of AKT, is altered in TMBIM6 KO cells. TMBIM6 deletion decreased the phosphorylation of AKT and NDRG1 as mTORC2 substrates, and TSC2 as AKT substrate (Fig. 3b, Supplementary Fig. 4A–C). Immunofluorescence staining confirmed that the phosphorylation of AKT was decreased by TMBIM6 deletion (Supplementary Fig. 4D). Consistently, over-expressing TMBIM6 in HeLa cells increased mTORC2 activity (Supplementary Fig. 4E). Reintroducing TMBIM6 into TMBIM6 KO HT1080 cells restored the phosphorylation of AKT

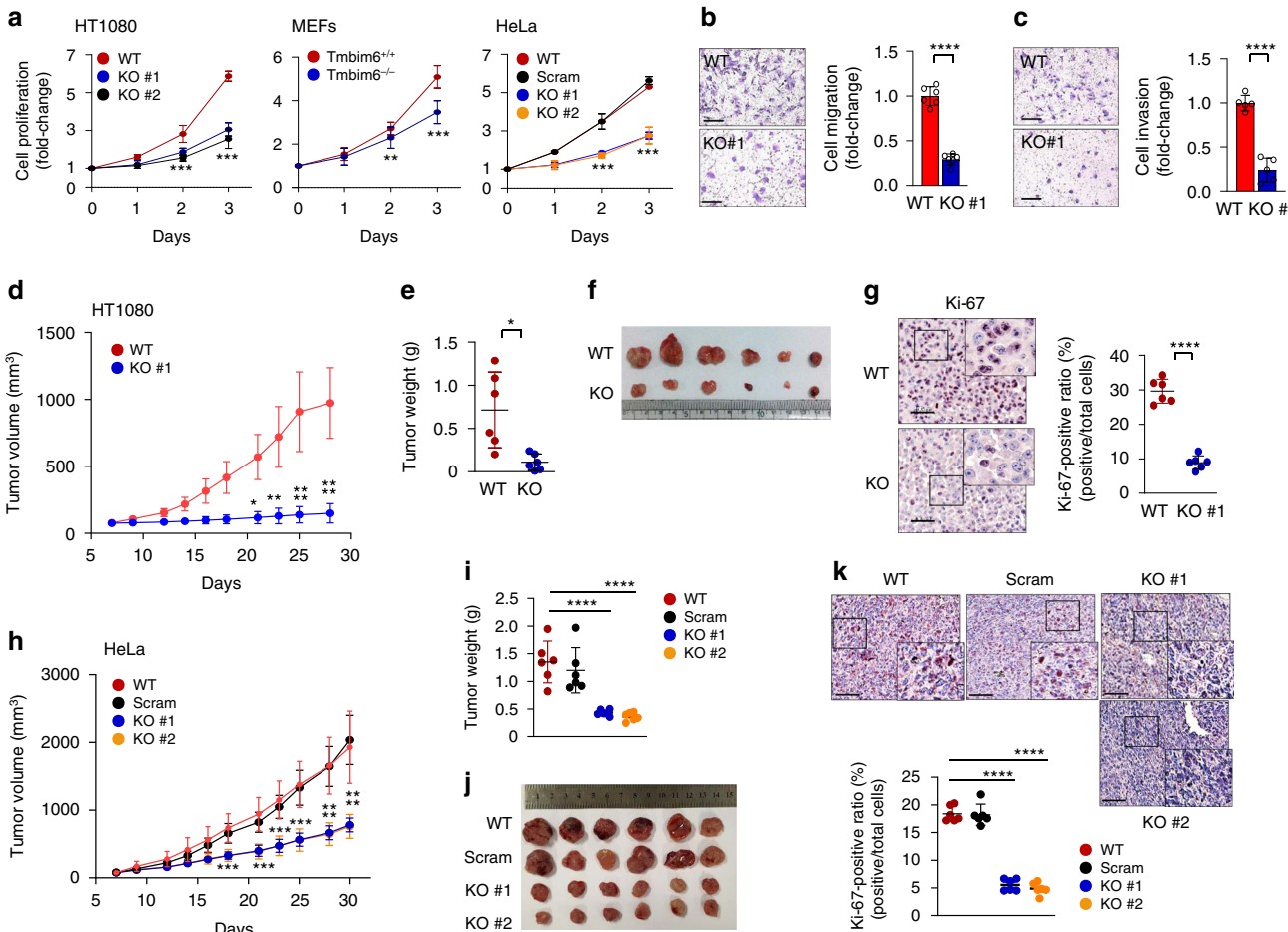

**Fig. 2 TMBIM6 enhances tumor growth. a** Proliferation of TMBIM6 KO and WT HT1080 cells, HeLa cells, and MEFs ($n = 3$ independent experiments). Data are presented as means ± SD. **$p < 0.01$, ***$p < 0.001$, two-way ANOVA followed by Bonferroni's post hoc test. **b** Images of migrated cells in TMBIM6 KO and WT HT1080. Right, quantification of WT cells normalized to KO cells ($n = 6$ independent experiments). Scale bars, 100 μm. Data are presented as means ± SD. ****$p < 0.0001$, two-tailed unpaired $t$-test. **c** Images of invasive cells in TMBIM6 KO and WT HT1080. Right, quantification of WT cells normalized to KO cells ($n = 5$ independent experiments). Scale bars, 100 μm. Data are presented as means ± SD. ****$p < 0.001$, two-tailed unpaired $t$-test. **d-f** Tumor volume, weight, and size derived from TMBIM6 KO or WT HT1080 cells injected into the flank of 6-week-old nude mice ($n = 6$ mice per group). Data are presented as means ± SD. *$p < 0.05$; **$p < 0.01$; ****$p < 0.0001$, two-way ANOVA followed by Bonferroni's post hoc test in D and two-tailed unpaired $t$-test in (**e**). **g** Histological images by immunohistochemical detection of Ki-67. Right, quantification of Ki-67-positive cells in xenograft tumors derived from TMBIM6 KO and WT HT1080 cells ($n = 6$ mice per group). Scale bars, 100 μm. Data are presented as means ± SD. ****$p < 0.0001$, two-tailed unpaired $t$-test. **h-j** Tumor volume, weight, and size derived from TMBIM6 KO or WT HeLa cells injected into the flank of 6-week-old nude mice ($n = 6$ mice per group). Data are presented as means ± SD. ***$p < 0.001$; ****$p < 0.0001$, two-way ANOVA followed by Bonferroni's post hoc test in (**h**) and one-way ANOVA followed by Tukey's post hoc test in (**i**). **k** Histological images by immunohistochemical detection of Ki-67. Right, quantification of Ki-67-positive cells in xenograft tumors derived from TMBIM6 KO and WT HeLa cells ($n = 6$ mice per group). Scale bars, 100 μm. Data are presented as means ± SD. ****$p < 0.0001$, two-way ANOVA followed by Bonferroni's post hoc test.

(pAKT-S473) and NDRG1 (pNDRG1-S939) (Fig. 3c). The phosphorylation of AKT upon insulin, IGF1, and EGF stimulation following serum starvation, was highly induced in WT cells compared with that in TMBIM6 KO (Fig. 3d). Consistent with the above results, the phosphorylation of AKT (pAKT-S473) in TMBIM6 KO MEF cells (MEF$^{-/-}$) with TMBIM6-HA overexpression increased upon insulin stimulation after serum starvation (Supplementary Fig. 4F). To further examine whether AKT phosphorylation is dependent on TMBIM6, we established a stable T-Rex-293 cell with tetracycline-inducible TMBIM6 expression. TMBIM6 level was increased by doxycycline treatment in a dose-dependent manner, with a concomitant increase in AKT phosphorylation (Supplementary Fig. 4G), suggesting that TMBIM6 is one of the essential genes for mTORC2 signaling, which regulates AKT activity.

Since the assembly of mTORC2 and its association with ribosomes is closely related to AKT phosphorylation[29], this was evaluated in TMBIM6 KO cells. Gel filtration assay using MEFs showed that mTORC2 was downregulated by TMBIM6 deletion (Fig. 3e). In in situ proximity ligation assay (PLA), the interactions between RICTOR and mTOR, RPL19, and RPS16 were markedly decreased in TMBIM6 KO HT1080 and HeLa cells (Fig. 3f, Supplementary Fig. 4H). In a co-immunoprecipitation (Co-IP) assay with RPL19, the binding of mTOR, RICTOR, SIN1, and GβL [also known as mLST8] to RPL19 was mostly abrogated in KO cells (Fig. 3g). Moreover, the expression levels of protein and mRNA of these genes were same in TMBIM6 WT and KO cells (Fig. 3g, Supplementary Fig. 4I). To further determine whether the binding between mTORC2 and ribosomes is dependent on TMBIM6, we performed Co-IP assay

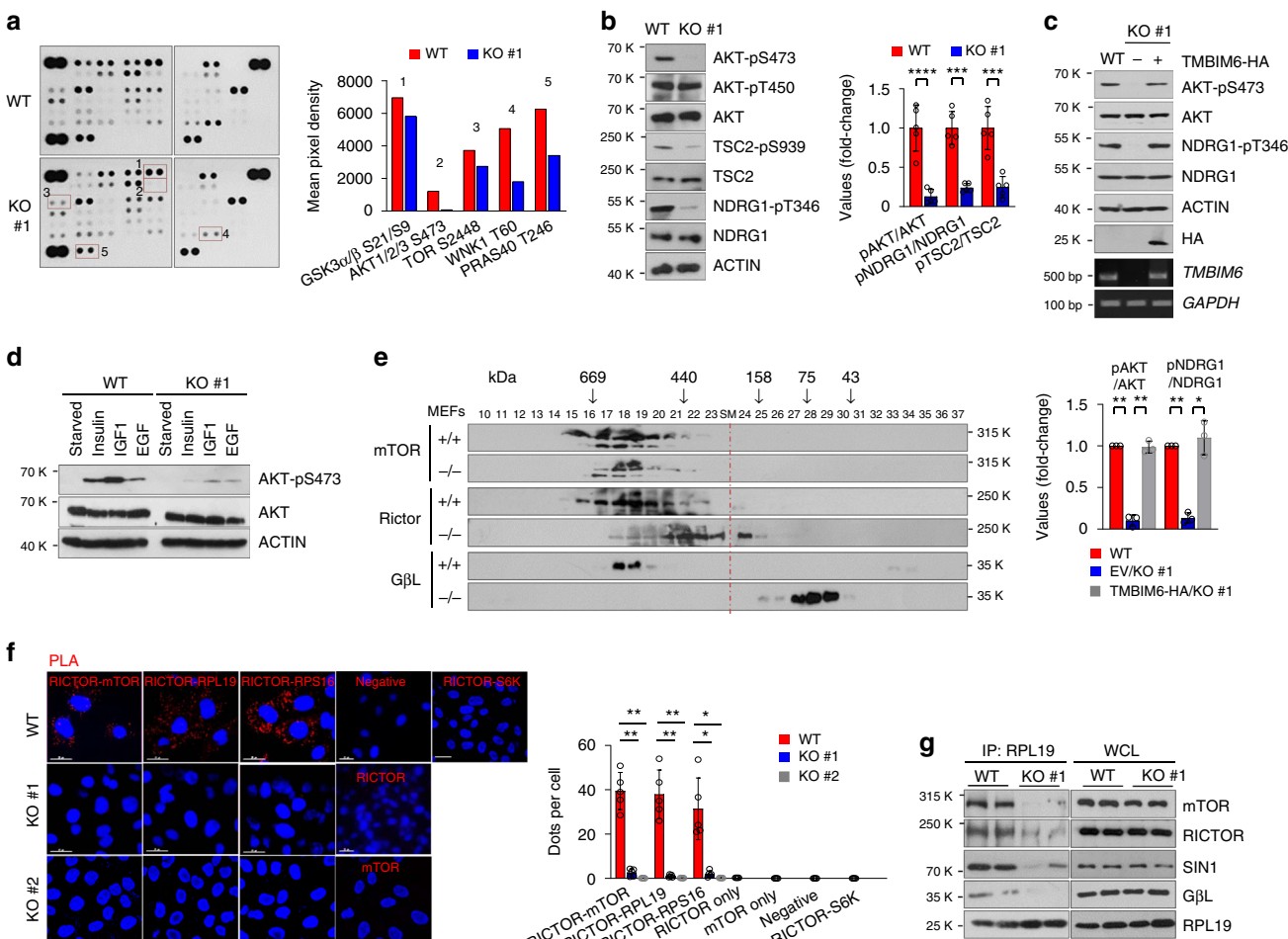

**Fig. 3 TMBIM6 regulates mTORC2 activation. a** Expression and phosphorylation of 43 proteins were examined by Proteome Profile Human Phospho-Kinase Array in WT and TMBIM6 KO HT1080 cells. Right; the relative phosphorylations of indicated proteins were quantitated by ImagJ. Blots represent one of two experiments, with similar results obtained. **b** pAKT, pTSC2, and pNDRG1 were analyzed in TMBIM6 KO and WT HT1080 cells using Western blots (left) and normalized to total proteins of WT cells (right; $n = 5$ independent experiments). Data are presented as means ± SD. ***$p < 0.001$, ****$p < 0.0001$, two-way ANOVA followed by Bonferroni's post hoc test. **c** Immunoblotting, RT-PCR, and its gene quantification analysis were performed in TMBIM6-HA-stably expressing or non-expressing TMBIM6 KO cells ($n = 3$ independent experiments). Data are presented as means ± SD. *$p < 0.05$, **$p < 0.01$, two-way ANOVA followed by Bonferroni's post hoc test. **d** After serum starvation for 12 h, TMBIM6 KO and WT HT1080 cells were stimulated with insulin (100 ng/ml), IGF1 (100 ng/ml), or EGF (100 ng/ml), and immunoblotting was performed with indicated antibodies. **e** Gel filtration assay of extracts of TMBIM6 KO and WT MEFs. The red line represents the size marker. **f** PLA between indicated proteins (red dots) in TMBIM6 KO and WT HT1080 cells. For the PLA, the ribosomal protein S6 kinase beta-1 (S6K1) was also applied as a negative control. Scale bar, 15 μm. Right, quantification of red dots ($n = 5$ independent experiments). Data are presented as means ± SD. *$p < 0.05$, **$p < 0.01$, two-way ANOVA followed by Bonferroni's post hoc test. **g** Immunoblot analysis of anti-RPL19 immunoprecipitate (IP) and whole cell lysate (WCL) of TMBIM6 KO and WT HT1080 cells.

with anti-RICTOR antibody upon insulin stimulation following serum starvation. Anti-RICTOR antibody pulled down with mTOR, GβL, and RPS16 in WT cells, but not in TMBIM6 KO cells (Supplementary Fig. 4J).

To identify whether reducing mTORC2 activity in TMBIM6 KO cells is related to impairment of ribosome maturation, we performed fractionation in a sucrose gradient assay to separate polysomes from 80S, 60S, and 40S ribosomes. The pattern of ribosome profiling was same between TMBIM6 WT and KO cells, indicating TMBIM6 is not related with ribosome maturation (Supplementary Fig. 5A). However, mTORC2 components were relatively less detected in both the polysomal and ribosomal fractions from TMBIM6 KO HT1080 cells compared with those from WT cells (Supplementary Fig. 5B). In addition, TMBIM6 was copurified with polysome and ribosome fractions in TMBIM6-rescued cells (Supplementary Fig. 5C). Since mTORC2 physically interacts with translating (mRNA-bound) and non-

translating 80S ribosomes[29] and TMBIM6 bind to the mTORC2, we next determined whether TMBIM6 is copurified with mTORC2 at mRNA-bound ribosomes. In mRNA-bound ribosomes purified by pull-down of poly(A) mRNA with oligo(dT) cellulose, TMBIM6 was copurified with mTOR, RICTOR, and RPL19 (Supplementary Fig. 5D). Collectively, these results suggest that TMBIM6 regulates the assembly of mTORC2 components and promotes the physical association between mTORC2 and ribosomes. mTORC2 interacts with ER-bound ribosomes[29,31] on the ER membrane, which is required for its kinase activity[29]. To identify whether the localization of mTORC2 on ER is different between TMBIM6 WT and KO cells, we carried out immunofluorescence analysis. The co-localization of mTORC2 components with the ER marker protein disulfide isomerase (PDI) was also decreased in TMBIM6 KO cells (Supplementary Fig. 6), indicating that TMBIM6 regulates mTORC2 residency on the ER.

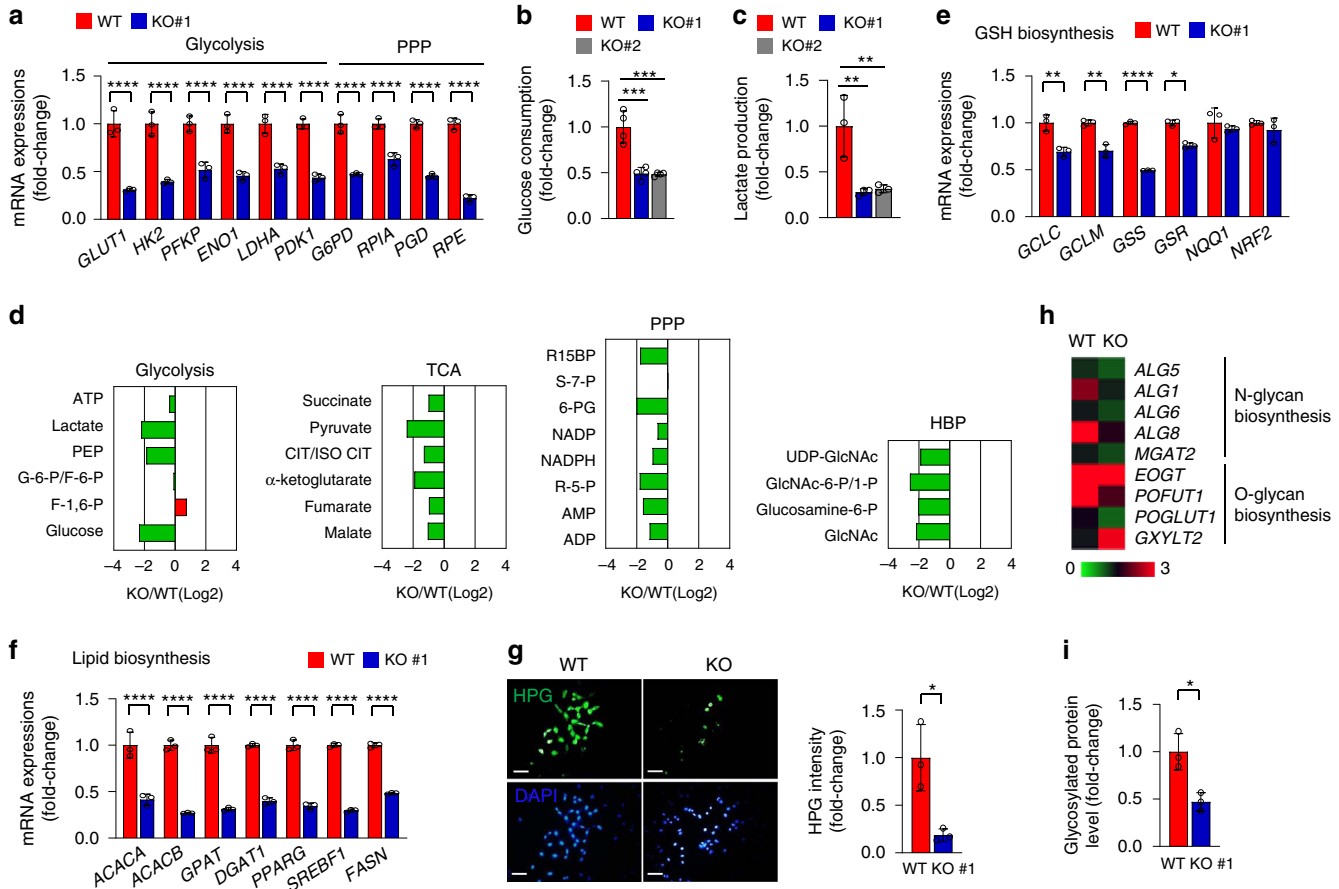

**Fig. 4 TMBIM6 regulates AKT-dependent metabolism. a** mRNA levels of glycolysis- and PPP-related genes in TMBIM6 KO and WT HT1080 cells, as determined by qRT-PCR. Quantification data represent the expression level of genes in the KO cells compared with those in normalized WT cells ($n = 3$ independent experiments). Data are presented as means ± SD. ****$p < 0.0001$, two-way ANOVA followed by Bonferroni's post hoc test. **b**, **c** Glucose consumption and lactate production in TMBIM6 KO and WT HT1080 cells ($n = 3$ independent experiments). Data are presented as means ± SD. **$p < 0.01$; ***$p < 0.001$, one-way ANOVA followed by Tukey's post hoc test. **d** Metabolite analysis in TMBIM6 KO and WT HT1080 cells ($n = 2$ independent experiments). **e**, **f** mRNA levels of GSH biosynthesis genes and de novo lipid biosynthesis genes in TMBIM6 KO and WT HT1080 cells, as determined by qRT-PCR. Quantification data represent the expression level of genes in the KO cells compared with those in normalized WT cells ($n = 3$ independent experiments). Data are presented as means ± SD. *$p < 0.05$, **$p < 0.01$, ****$p < 0.0001$, two-way ANOVA followed by Bonferroni's post hoc test. **g** Immunofluorescence images of protein synthesis in TMBIM6 KO and WT HT1080 cells. Right, quantification data represent the expression intensity compared with that in normalized WT cells ($n = 3$ independent experiments). Data are presented as means ± SD. *$p < 0.05$, two-tailed unpaired $t$-test. Scale bar, 15 μm. **h** Glycosylation-related gene lists in TMBIM6 KO and WT HT1080 cells by microarray. **i** Glycosylated protein levels in TMBIM6 KO and WT HT1080 cells ($n = 3$ independent experiments). Data are presented as means ± SD. *$p < 0.05$, two-tailed unpaired $t$-test.

**mTORC2 activation by TMBIM6 regulates cellular metabolism.** mTORC2 regulates cellular bioenergetics by modulating glycolytic gene expression, aerobic glycolysis, glutathione (GSH) biosynthesis, hexosamine biosynthesis pathway (HBP), and glycosylation[35–38]. TMBIM6 KO cells showed downregulation of glycolytic genes (Fig. 4a), resulting in reduced glucose consumption and lactate production (Fig. 4b, c). The expression of genes related to the pentose phosphate pathway (PPP) was also decreased in TMBIM6 KO, which was reversed in TMBIM6-overexpressing HeLa cells (Fig. 4a and Supplementary Fig. 7). An MS analysis showed that the levels of metabolites from glycolysis, tricarboxylic acid cycle (TCA), PPP, and HBP were decreased in TMBIM6 KO cells relative to those in WT cells (Fig. 4d) indicating that metabolic pathways are dysregulated in the absence of TMBIM6, where is linked to the inhibition of mTORC2 activity.

We next analyzed the expression of genes related to AKT-associated GSH biosynthesis, de novo lipogenesis, and protein synthesis. TMBIM6 KO cells showed decreased expression of *GCLC*, *GCLM*, *GSS*, and *GSR* (Fig. 4e), which are targets of the transcription factor NRF2, a master regulator of the cellular

antioxidant response, which is in agreement with a previous report[37]. Notably, the expression of genes related to de novo lipogenesis involving *SREBF1*, which is required for cholesterol, fatty acid, triglyceride, and phospholipid synthesis[39] was also decreased in these cells (Fig. 4f). Overall, protein synthesis was significantly reduced by the loss of TMBIM6 (Fig. 4g).

An examination of glycoprotein folding status revealed that basal levels of glycosylated proteins were lower in TMBIM6 KO than in WT HT1080 cells (Fig. 4h). Microarray analysis data showed that the expression of glycosylation-related genes including *ALG5*, *ALG1*, *ALG6*, *ALG8*, *MGAT2*, *EOGT*, *POFUT1*, and *POGLUT1* was decreased in TMBIM6 KO cells (Fig. 4i). Taken together, these data indicate that TMBIM6 regulates mTORC2 activity by altering signal transduction in metabolism.

**TMBIM6 directly interacts with mTORC2 and ribosomes.** We investigated whether TMBIM6 directly interacts with mTORC2 components and ribosomes by transfecting TMBIM6 KO HT1080 cells with human influenza hemagglutinin (HA)-tagged TMBIM6 (TMBIM6-HA). The gel filtration assay and

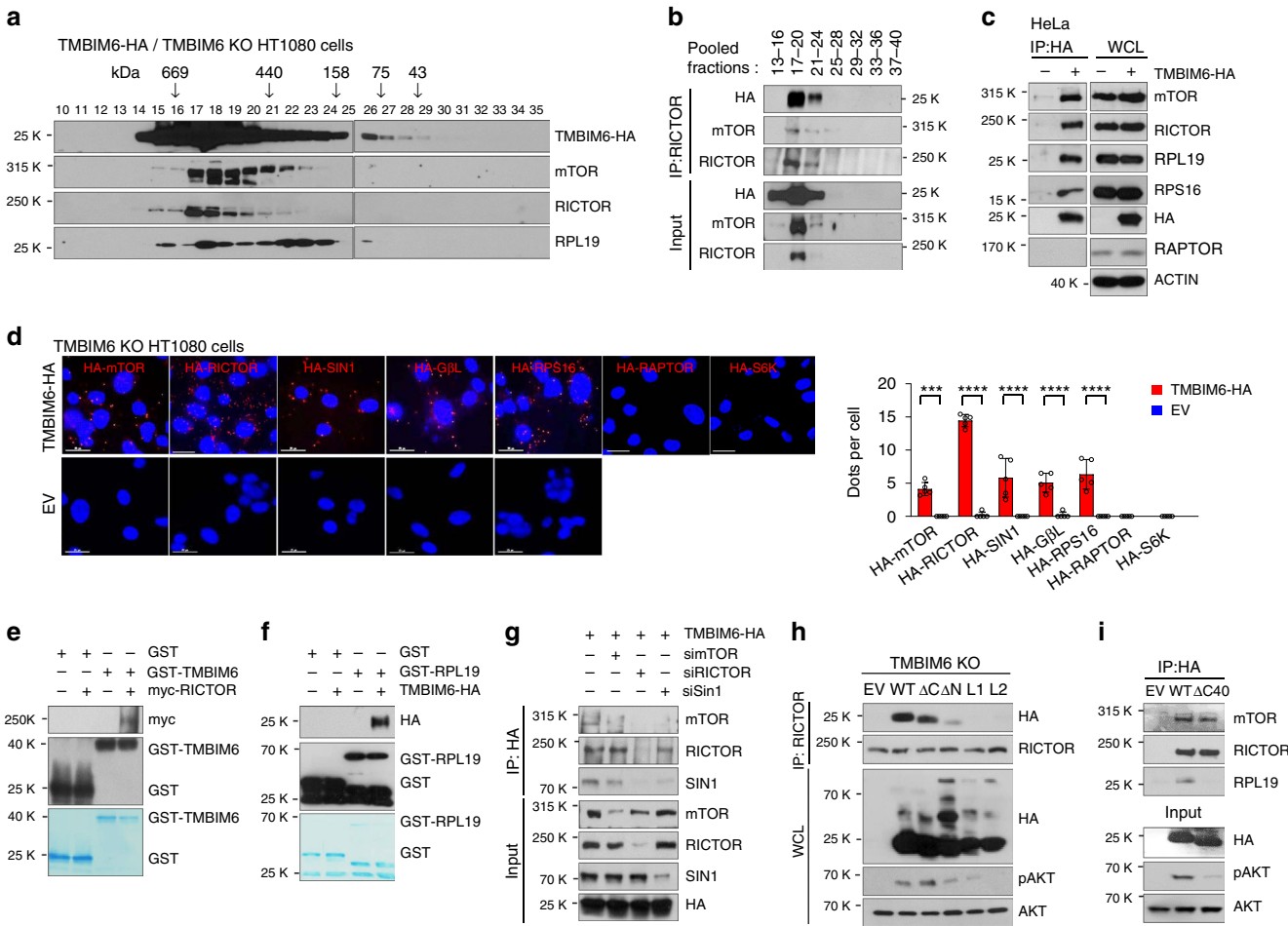

**Fig. 5 TMBIM6 directly binds to mTORC2 and ribosomes. a** Gel filtration assay of lysates of TMBIM6 KO HT1080 cells transiently overexpressing TMBIM6-HA. **b** Anti-RICTOR immunoprecipitate (IP) of pooled fractions of TMBIM6 KO HT1080 cells transiently overexpressing TMBIM6-HA. The inputs were analyzed by immunoblotting. **c** Immunoblot analysis of anti-HA IP and whole cell lysate (WCL) of TMBIM6-HA overexpressing HeLa cells. **d** PLA between TMBIM6-HA and mTORC2 components (red dots) in TMBIM6 stably overexpressing HT1080 cells. Right, quantification of red dots ($n = 5$ independent experiments). Scale bar, 15 μm. Data are presented as means ± SD. ***$p < 0.001$, ****$p < 0.0001$, two-way ANOVA followed by Bonferroni's post hoc test. **e** GST pull-down assay between GST-TMBIM6 and myc-RICTOR. **f** GST pull-down assay in the presence or absence of HA-TMBIM6. **g** Immunoblot analysis of WCL of HT1080 cells stably expressing TMBIM6 and transfected with scrambled, mTOR, RICTOR, or SIN1 siRNA and immunoprecipitated with anti-HA antibody. **h** Immunoblot analysis of the immunoprecipitates with anti-RICTOR antibody and whole cell lysates of HT1080 cells transiently transfected with TMBIM6 and TMBIM6 mutant constructs. **i** Immunoblot analysis of the immunoprecipitates with anti-HA antibody and input of HT1080 cells transiently transfected with TMBIM6 and TMBIM6 mutant constructs.

immunoprecipitation of the pooled samples using anti-RICTOR antibody revealed that TMBIM6 was directly bound to mTORC2 (Fig. 5a, b). Moreover, an association between TMBIM6 and endogenous mTORC2 or ribosomes (60S RPL19 and 40S RPS16) in TMBIM6-HA-overexpressing HeLa and HT1080 cells, but not with RAPTOR as mTORC1 subunit, was confirmed by immunoprecipitation and PLA analysis (Fig. 5c, d). We replicated the direct binding of TMBIM6 to RICTOR and RPL19 by glutathione S-transferase (GST) pull-down assay, in which TMBIM6 was associated with RPL19 and RICTOR (Fig. 5e, f). To validate whether TMBIM6 is associated with mTORC2, we performed immunoprecipitation with anti-RICTOR antibody and then analyzed the binding proteins using liquid chromatography-tandem mass spectrometry (LC-MS/MS). The results showed that TMBIM6 is one of the binding partners of mTORC2 (Supplementary Data 2). We next examined the interaction between TMBIM6 and mTORC2 components. RICTOR is located close to the FKBP12-rapamycin-binding domain of mTOR and is bound by SIN1, whereas the mTOR kinase domain is bound by mLST8[40]. Silencing RICTOR by siRNA abrogated the interaction

between mTORC2 and TMBIM6-HA, which was not observed by mTOR disruption (Fig. 5g).

TMBIM6 is composed of six or seven transmembrane regions with mostly α-helical structures, which C-terminus of TMBIM6 resides in the cytosol by TMHMM or in ER intraluminal space by the bacterial homolog BsYetJ[41–45] (Supplementary Fig. 8A). Although BsYetJ is a bacterial protein related to hTMBIM6, it has only 23.77% amino acid identity by BLASTp (Supplementary Fig. 8B). To further understand TMBIM6 topology, we performed immunofluorescence using cells overexpressing TMBIM6 tagged with the N-terminal (HA-TMBIM6) and C-terminal (TMBIM6-HA) HA tag by modified previous reports[41]. While Triton X-100 permeabilizes all membranes and leads to staining of both luminal and cytosolic epitopes, while digitonin makes only accessible to cytosolic epitopes for antibodies. The PDI retained in ER lumen was used as a negative control. Immunofluorescences of HA-TMBIM6 and TMBIM6-HA were detected in all conditions, whereas PDI fluorescence was not detected in the presence of digitonin (Supplementary Fig. 8C). These results suggest that N- and C-terminal of TMBIM6 is cytosolic exposed

in six-transmembrane structure condition, although we cannot exclude a possibility that topology of TMBIM6 might be altered by the fusion with HA itself as previously mentioned[41].

To identify which TMBIM6 domain interacts with RICTOR, we generated TMBIM6 mutant constructs, including deletion of 29 amino acids (AA) of the N-terminal (ΔN) and 9AA of the C-terminal (ΔC), and changing all residues of cytosolic loop 1 (L1) and loop 2 (L2) to alanine residues for all the six or seven transmembrane structures. The association between TMBIM6 and RICTOR was reduced in TMBIM6-ΔN or almost blocked in TMBIM6-L1 and L2 by co-IP assay (Fig. 5h). To further study which TMBIM6 domain interacts with RPL19, we generated TMBIM6 mutant constructs with deletion of 40AA of the C-terminal (ΔC40). Immunoprecipitation assay showed that the association with RPL19 to TMBIM6-ΔC40 was abrogated, whereas the interaction with RICTOR or mTOR was not altered (Fig. 5i). The phosphorylation of AKT (pAKT-S473) was also decreased in RPL19 or RICTOR non-associated TMBIM6 mutants (Fig. 5h, i). Taken together, these data indicate that TMBIM6 interacts with mTORC2 and ribosomes, and that this interaction is important for the kinase activity of mTORC2.

We then performed T4 phage display screening using a cDNA library of human tissues and the 50AA cytosolic domain of TMBIM6 as bait. The 60S RPL19 was identified as a ligand of TMBIM6 (Supplementary Fig. 9). Overall, these results suggest that the physical interaction between TMBIM6 and RICOTR or ribosome is required to enhance mTORC2 activity.

**TMBIM6-associated ER Ca$^{2+}$ release regulates mTORC2 activation.** Our data suggested that mTORC2 activation requires the binding of RICTOR to TMBIM6—a Ca$^{2+}$ channel-like protein—on the ER membrane. Therefore, we investigated the role of Ca$^{2+}$ in mTORC2 activation. Ca$^{2+}$ depletion by BAPTA acetoxymethyl ester (BAPTA-AM), not BAPTA or a Ca$^{2+}$ chelator with slower binding kinetics (i.e., EGTA-AM), blocked the association between mTORC2 and ribosomes as determined by the PLA (Supplementary Fig. 10A), suggesting that local Ca$^{2+}$ concentration controls mTORC2 activation.

To assess whether Ca$^{2+}$ released from TMBIM6 influences the interaction between mTORC2 components and ribosome, we first checked Ca$^{2+}$ leaky characteristics of TMBIM6 using a TMBIM6-GCaMP3 construct, which is based on the finding that leaky calcium but not ER lumen is detected upon binding of Ca$^{2+}$ in the cytosol of this protein (Supplementary Fig. 10B). TMBIM6-associated Ca$^{2+}$ release as determined by the intensities of fluorescence was detected in HT1080 cells transiently transfected with WT TMBIM6-GCaMP3 but not in Ca$^{2+}$ channel mutant TMBIM6 (TMBIM6$^{D213A}$)-GCaMP3[41,46] (Fig. 6a). Next, PLA assay showed that the interaction between RICTOR and mTOR, or RICTOR and RPL119 increased in HT1080 cells ectopically expressing WT but not TMBIM6$^{D213A}$ cells (Fig. 6b). In co-IP assay, RICTOR binding to TMBIM6 was not obviously different between cells expressing WT and TMBIM6$^{D213A}$ (Fig. 6c). However, mTOR binding to TMBIM6 was slightly decreased in TMBIM6$^{D213A}$ cells. Surprisingly, RPL19 and RPS16 binding to TMBIM6 was markedly decreased in the mutant cells (Fig. 6c). In addition, immunoblot and immunofluorescence analysis revealed that the phosphorylation of AKT was also decreased in TMBIM6$^{D213A}$ cells (Fig. 6d, e). To identify whether differential retention of WT or TMBIM6$^{D213A}$ in ER affects mTORC2 assembly and AKT phosphorylation, we performed immuno-fluorescence assay by ectopic expression with WT and TMBIM6$^{D213A}$ in HT1080 cells. As shown Supplementary Fig. 10C, WT and TMBIM6$^{D213A}$ were retained in ER at a comparable level. Thus, TMBIM6-associated AKT activation is

based upon the following characteristics of protein interactions: the binding of TMBIM6 with RICTOR was independent of Ca$^{2+}$ leakage, whereas the interaction of TMBIM6 with mTOR or ribosome was dependent on local Ca$^{2+}$ leakage (Supplementary Fig. 10D).

To investigate the importance of Ca$^{2+}$ leakage through TMBIM6, we stably rescued the expression of WT or TMBIM6$^{D213A}$ in TMBIM6 KO HT1080 cells, and then determined cellular metabolism. Cell proliferation was restored in TMBIM6-rescued KO cells, not TMBIM6$^{D213A}$, without differences of TMBIM6 expression (Fig. 6f, Supplementary Fig. 11A). Furthermore, the expression of genes related to the glycolysis and PPP were recovered in TMBIM6-rescued KO cells, indicating recovery of glucose consumption and lactate production (Fig. 6g–i). Consistently, the expression of genes related to GSH biosynthesis, and de novo lipogenesis were restored in TMBIM6-rescued KO cells, not TMBIM6$^{D213A}$ (Supplementary Fig. 11B, C). By mass spectrometry, the levels of metabolites from glycolysis, TCA, PPP, and HBP have restored in TMBIM6-rescued cells compared with TMBIM6$^{D213A}$ cells (Fig. 6j, Supplementary Fig. 11D). Moreover, the patterns of ribosome profiling were same in all the cells (Supplementary Fig. 11E), indicating that TMBIM6 is independent of ribosome maturation. Collectively, these results suggest that TMBIM6 regulates metabolic pathways through its characteristics "Ca$^{2+}$ leakage-associated mTORC2 activation".

**TMBIM6 antagonist reduces mTORC2 activity, inhibiting TMBIM6-associated tumorigenicity.** Initially, to identify small molecule TMBIM6 antagonists, we performed high-throughput screening from materials of the Korea Chemical Bank and elicited chalcone scaffold (Supplementary Fig. 12A). From the optimization of R1 and R2 position with diverse substituents, BIA was developed as a tool compound dependent cell viability using HT1080 cells, which have been used extensively in biomedical research (Supplementary Fig. 12B–D). We first examined TMBIM6 expression in breast cancer cell lines, which showed high correlation between TMBIM6 expression and cancer prognosis (Fig. 1f). The results showed that MCF7 and MDA-MB-231 cells highly expressed TMBIM6, whereas SKBR3 cells have low expression relative to HT1080 cells (Supplementary Fig. 13A). Next, the proliferation and cell viability of all cell lines were inhibited by treatment with 5 μM BIA (Fig. 7a, Supplementary Fig. 13B). The IC50 values at 3 days were $1.7 \pm 0.1$ μM for HT1080, $2.6 \pm 0.4$ μM for MCF cells, $2.6 \pm 0.5$ μM for MDA-MB-231 cells, and $2.4 \pm 0.4$ μM for SKBR3 cells. Moreover, HT1080 cells stably overexpressing TMBIM6 showed high sensitivity to BIA (Supplementary Fig. 13C).

To identify whether BIA decreases the binding between TMBIM6 and mTORC2, we performed a gel filtration assay in TMBIM6-overexpressing HT1080 cells after treatment with BIA during 24 h. As shown in Fig. 7b, the co-elution pattern with mTORC2 components (mTOR and RICTOR) and ribosomes (RPL19) was delayed in the BIA-treated HT1080 cells. PLA and immunoprecipitation assay showed that endogenous protein interaction between mTORC2 and ribosome, or the binding of TMBIM6 to mTORC2 and ribosome, inhibited by BIA (Fig. 7c–e, Supplementary Fig. 13D), resulting in the phosphorylation of AKT was fully decreased in all cells (Fig. 7c, d). Taken together, these data demonstrate that BIA reduced cell proliferation by inducing the dissociation of TMBIM6 from mTORC2 and ribosome, result in cell death.

To clarify whether anti-proliferation effect by BIA at that dose is an off-target effect, we examined cell proliferation in TMBIM6 KO HT1080 cells. The cell proliferation rate and AKT

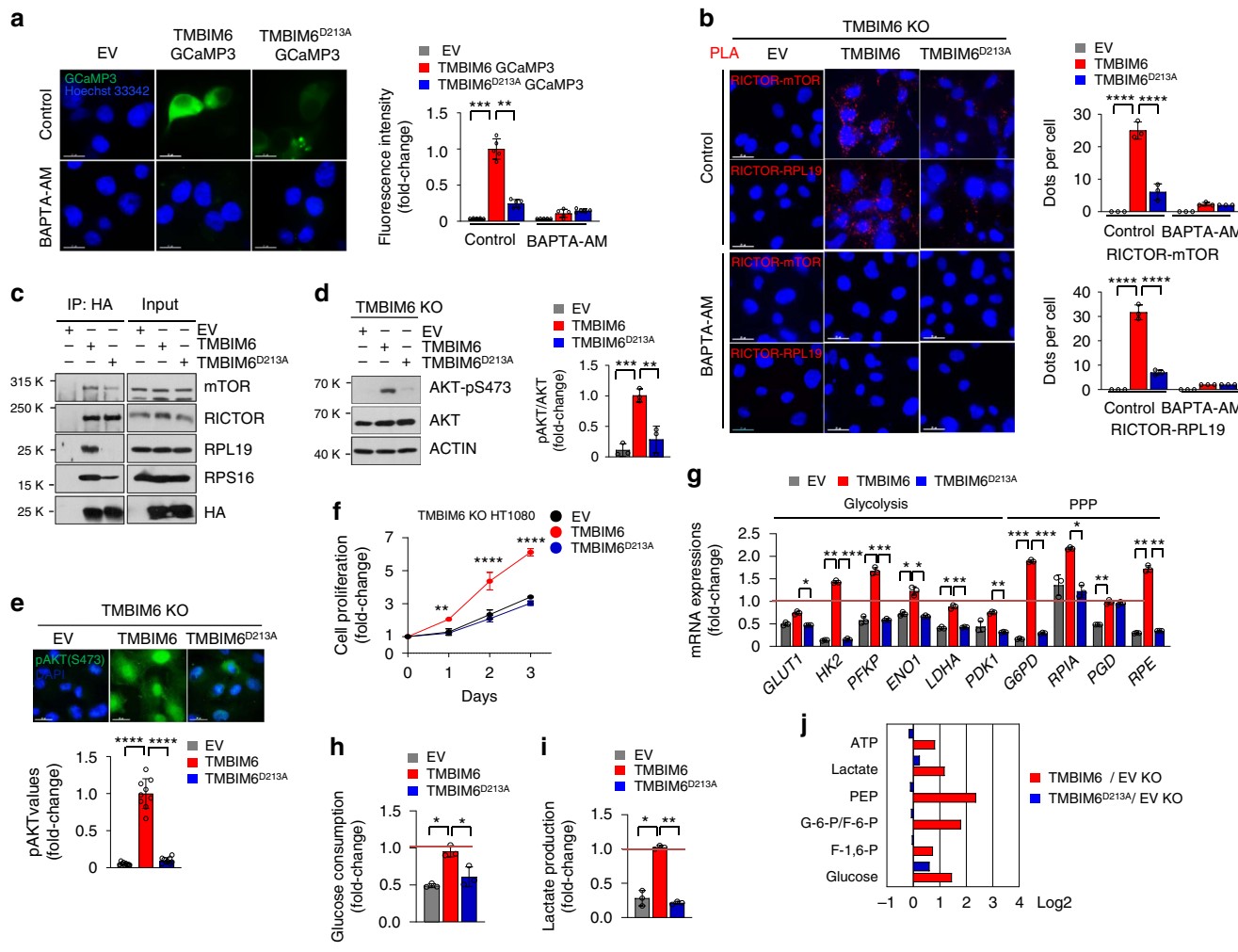

**Fig. 6 TMBIM6-induced Ca$^{2+}$ leakage affects mTORC2 assembly and the association between mTORC2 and ribosomes. a** Immunofluorescence images and fluorescence intensity (right) of TMBIM6-GCaMP3 and TMBIM6 D213A-GCaMP3 in the presence or absence of 10 μM BAPTA-AM. ($n = 5$ independent experiments). Scale bar, 15 μm. Data are presented as means ± SD. \*\**p* < 0.01, \*\*\**p* < 0.001, two-way ANOVA followed by Bonferroni's post hoc test. **b** PLA between Rictor and mTOR or between Rictor and RPL19 (red dots) in empty vector, TMBIM6, or D213A-transfected HT108 cells. ($n = 3$ independent experiments). Scale bar, 15 μm. Data are presented as means ± SD. \*\*\*\**p* < 0.0001, two-way ANOVA followed by Bonferroni's post hoc test. **c** Immunoblot analysis of the immunoprecipitates with Anti-HA antibody and input of cell lysates with the indicated antibodies. **d** Immunoblotting and quantification of phosphorylation of AKT in TMBIM6 KO HT1080 cells transfected with empty vector, TMBIM6 WT, and TMBIM6 D213A. ($n = 3$ independent experiments). Data are presented as means ± SD. \*\**p* < 0.01, \*\*\**p* < 0.001, one-way ANOVA followed by Tukey's post hoc test. **e** Immunofluorescence images and quantification of AKT phosphorylation ($n = 3$ independent experiments, total of nine images). Scale bar, 15 μm. Data are presented as means ± SD. \*\*\*\**p* < 0.0001, one-way ANOVA followed by Tukey's post hoc test. **f** Proliferation analysis of empty vector, TMBIM6, and TMBIM6 D213A-expressed HT1080 cells ($n = 3$ independent experiments). Data are presented as means ± SD. \*\**p* < 0.01, \*\*\*\**p* < 0.0001, two-way ANOVA followed by Bonferroni's post hoc test. **g** The quantification analysis of mRNA levels of glycolysis- and PPP-related genes in empty vector, TMBIM6, and TMBIM6 D213A-rescued TMBIM6 KO HT1080 cells, as determined by qRT-PCR. Quantification data represent the expression level of genes compared with those in normalized WT HT1080 cells (red line, $n = 3$ independent experiments). Data are presented as means ± SD. \**p* < 0.05, \*\**p* < 0.01, \*\*\**p* < 0.001, two-way ANOVA followed by Bonferroni's post hoc test. **h, i** Glucose consumption and lactate production in empty vector, TMBIM6, and TMBIM6 D213A-rescued TMBIM6 KO HT1080 cells ($n = 3$ independent experiments). Data are presented as means ± SD. \**p* < 0.05, \*\**p* < 0.01, one-way ANOVA followed by Tukey's post hoc test. **j** Metabolite analysis in empty vector, TMBIM6, and TMBIM6 D213A-rescued TMBIM6 KO HT1080 cells. Quantification data represent the metabolite level compared with those in empty vector-rescued TMBIM6 KO HT1080 cells ($n = 2$ independent experiments).

phosphorylation in TMBIM6 KO HT1080 cells were same in the presence or absence of BIA (Fig. 7a, Supplementary Fig. 13E) with the exceptions of high concentrations "20 and 30 μM" (Supplementary Fig. 13F), suggesting that BIA has on-target effect on TMBIM6 up to 10 μM.

In the above our results, since TMBIM6 regulates mTORC2 activation through ER Ca$^{2+}$ release, we verified whether BIA inhibits Ca$^{2+}$ release from TMBIM6. The TMBIM6-GCaMP3 green fluorescence showed a decreasing pattern in BIA-treated

cells (Fig. 7f). In addition, we demonstrated ER calcium status using the ER lumen calcium indicator (G-CEPIAer)[47] by application of 10 μM BIA. The fluorescence intensities were increased by BIA treatment as compared with those in untreated control cells (Fig. 7g), suggesting that BIA suppresses ER release of Ca$^{2+}$ from TMBIM6.

Cell migration and invasion are representative in vitro markers for cancer characteristics. BIA treatment decreased cell migration in HT1080, MCF7, MDA-MB-231, and SKBR3 cells (Fig. 8a), not

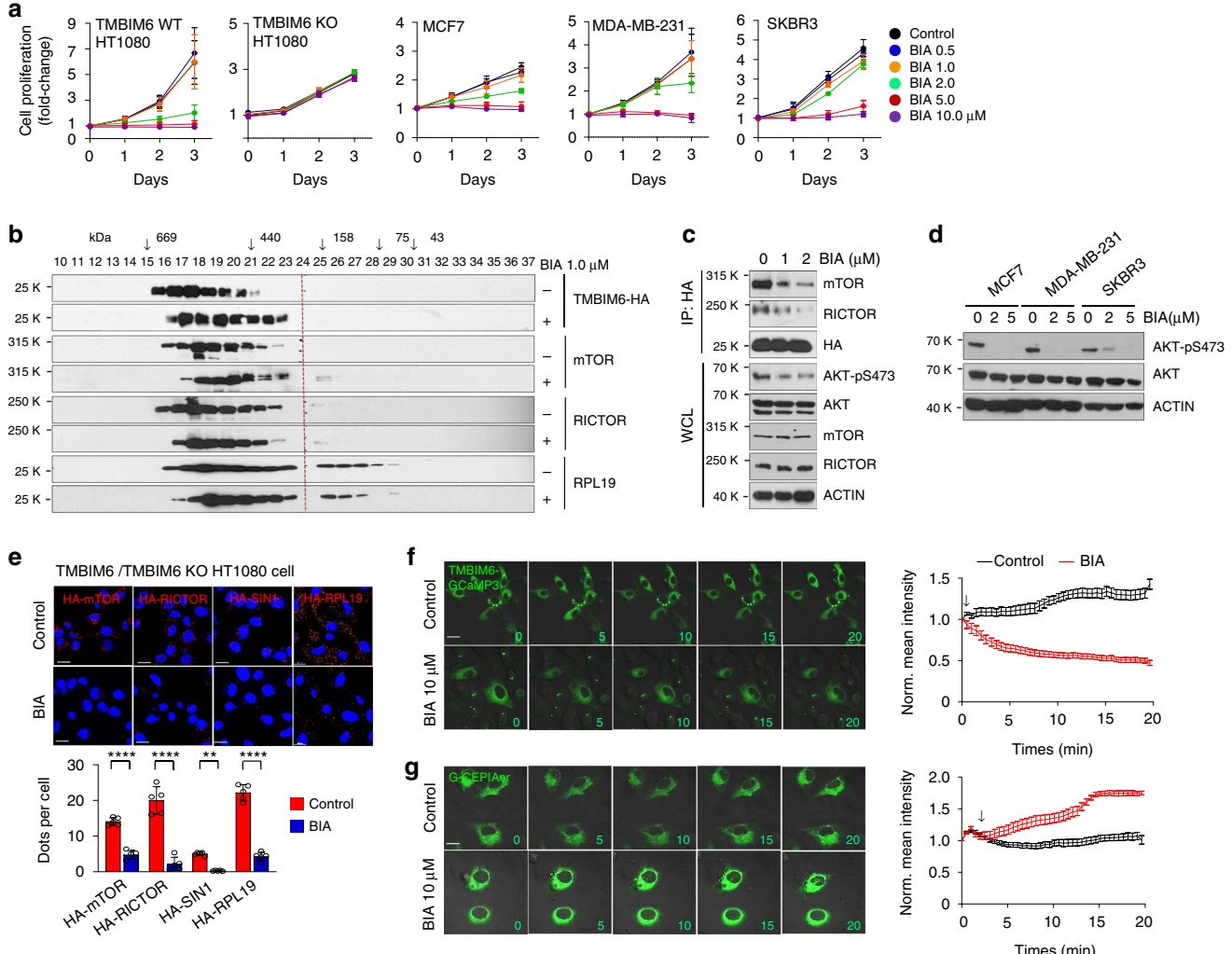

**Fig. 7 BIA, a TMBIM6 antagonist, suppresses tumor growth. a** Proliferation of TMBIM6 WT HT1080, TMBIM6 KO HT1080, MCF7, MDA-MB-231, and SKBR3 cells treated with BIA ($n = 3$ independent experiments). Data are presented as means ± SD. **b** Gel filtration assay of HT1080 cells treated with 1.0 μM BIA. The red line represents the size marker. **c** Immunoblot analysis of anti-HA immunoprecipitate (IP) and whole cell lysate (WCL) of HT1080 cells transiently overexpressing TMBIM6-HA and treated with BIA. **d** Immunoblotting of p-AKT, AKT, and actin in the indicated cell lines following treatment with BIA. **e** PLA between TMBIM6-HA and mTORC2 components or between TMBIM6-HA and RPL19 (red dots) in BIA-treated or non-treated TMBIM6 stably overexpressing HT1080 cells. Bottom, quantification of red dots ($n = 5$ independent experiments). Scale bar, 20 μm. Data are presented as means ± SD. **p < 0.01, ****p < 0.0001, two-way ANOVA followed by Bonferroni's post hoc test. **f, g** Real-time lapse images of HT1080 cells stably expressing TMBIM6-GCaMP3 and G-CEPIAer treated with BIA. Right, mean green intensity of every cells normalized to untreated cells ($n = 5$ independent experiments, total of 20 cells for TMBIM6-GCaMP3; total of 16 cells for G-CEPIAer). Scale bar, 15 μm. Data are presented as means ± SD.

TMBIM6 KO HT1080 cells (Supplementary Fig. 13G, H). Cell invasion in MDA-MB-231 and HT1080 cells was also decreased in the BIA-treated cells (Fig. 8b). Furthermore, the numbers and the size of spheroids were formed by the three-dimensional cultured cells, not showing multi-acinar structures, that was impaied by BIA (Fig. 8c).

We then established a zebrafish tumor model[48–51] in which 48 h post-fertilization embryos were implanted with human breast cancer cells labeled with DiI dye via injection into the perivitelline cavity (Fig. 8d). On day 3 after implantation, control tumor cells had migrated away from the primary sites, whereas nearly all tumor cells in the BIA treatment group remained at the site of injection.

To further determine whether BIA regresses tumor growth in vivo, we subcutaneously injected HT1080 and MDA-MB-231 cells into immunocompromised mice, which were further injected with 1 mg/kg BIA or vehicle (0.1% DMSO with saline) for 5 days per week during 25 days. The xenograft results showed that BIA

markedly impaired cell-driven tumor growth (Fig. 8e–h, Supplementary Fig. 13I, J). These results suggest that BIA-induced inhibition of AKT activity and tumor progression is due to the dissociation of TMBIM6 from mTORC2.

PIK3CA-AKT-mTOR signaling pathway is frequently activated in human cancers, and many small-molecule inhibitors have been developed that target various nodes in the pathway. However, the mTOR mutation in breast cancer resulting in mTORC1 inhibition leads to AKT activation via upregulation of receptor tyrosine kinases, leading to resistance to these inhibitors. To determine whether BIA is effective against HT1080, PANC-1 pancreatic cancer cells resistant to mTOR inhibitor, and other pancreatic cancer cells including Capan-1 and MIA PaCa-2 cells, we compared with mTOR inhibitors such as AZD8055, INK128, Omipalisib, OSI-027, and Voxtalisib. Cell viability was reduced to a greater extent by the treatment with BIA as compared with the mTOR inhibitors. Especially, BIA almost abrogated live cells in PANC-1 cells, which have 30–40% cell viability by the other

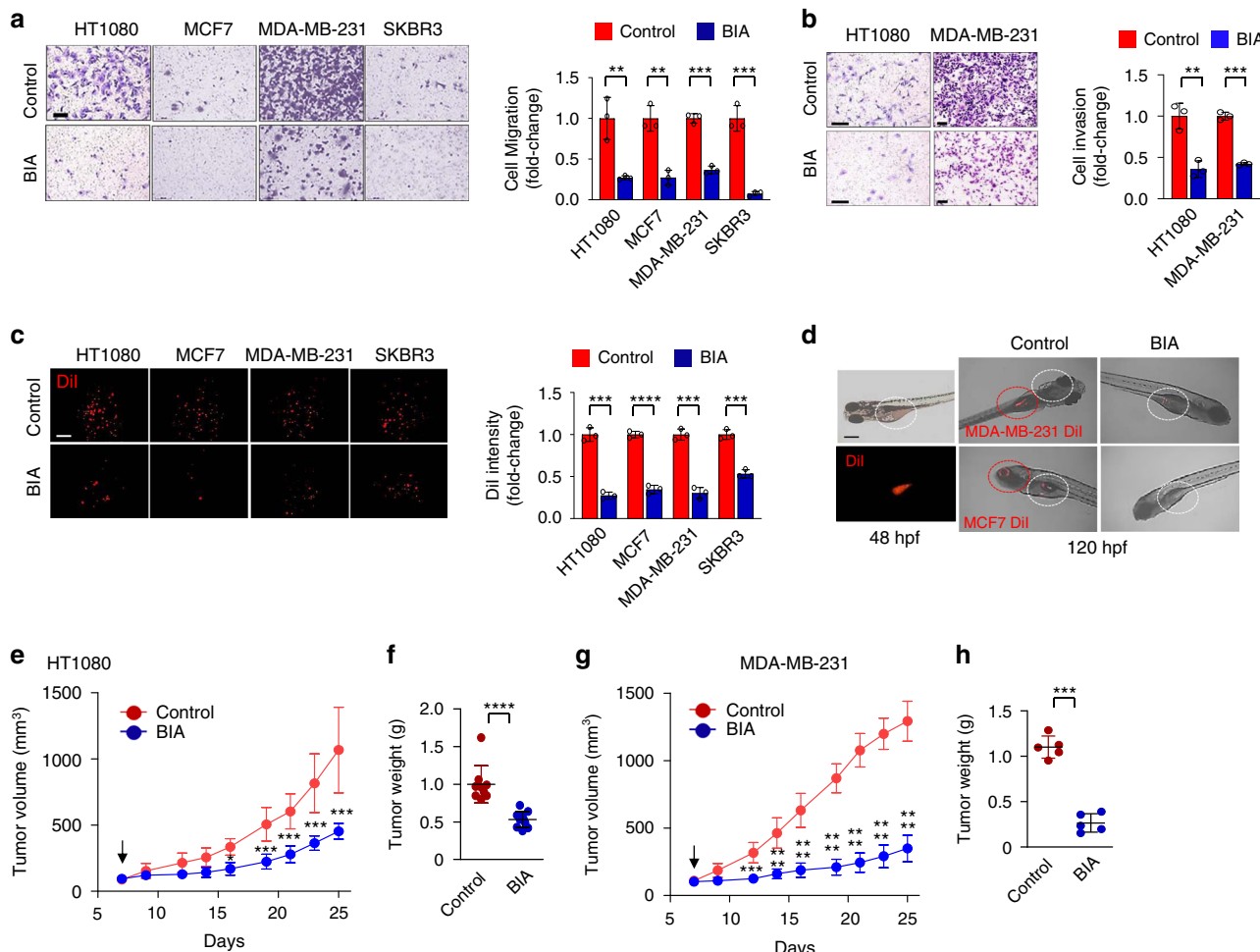

**Fig. 8 BIA inhibits tumor growth. a** Images of migrated cells in HT1080, MCF7, MDA-MB-231, and SKBR3 cells treated with 2.0 μM BIA. Right, quantification of migrated cells in the BIA-treated cells normalized to control cells ($n = 3$ independent experiments). Data are presented as means ± SD. **$p < 0.01$, ***$p < 0.001$, two-tailed unpaired $t$-test. Scale bar, 15 μm. **b** Images of invasive cells in HT1080 and MBA-MB-231 cells treated with 2.0 μM BIA. Right, quantification of invasive cells in BIA-treated cells normalized to control cells ($n = 3$ independent experiments). Data are presented as means ± SD. **$p < 0.01$, ***$p < 0.001$, two-tailed unpaired $t$-test. Scale bar, 15 μm. **c** DiI fluorescence images of the indicated cell lines after 7 days of 3D culture. Right, quantification of the intensity of fluorescence in the BIA-treated cells normalized to control cells ($n = 3$ independent experiments). Scale bar, 15 μm. Data are presented as means ± SD. ***$p < 0.001$, ****$p < 0.0001$, two-tailed unpaired $t$-test. **d** Images of control and BIA-treated zebrafish after injection of indicated cell lines into embryos. White and red circles represent cell injection and migration sites, respectively. Images represent one of nine experiments, with similar results obtained. Scale bar, 100 μm. **e**, **f** Volume and weight of tumors derived from HT1080 cells injected into nude mice treated with vehicle or 1 mg/kg BIA (vehicle; $n = 9$, BIA; $n = 11$ mice). Data are presented as means ± SD. *$p < 0.05$, ***$p < 0.001$, ****$p < 0.0001$, two-way ANOVA followed by Bonferroni's post hoc test in (**e**) and two-tailed unpaired $t$-test in (**f**). **g**, **h** Volume and weight of tumors derived from MDA-MB-231 cells injected into nude mice treated with vehicle or 1 mg/kg BIA ($n = 5$ mice per group). Data are presented as means ± SD. Data are presented as means ± SD. ***$p < 0.001$, ****$p < 0.0001$, two-way ANOVA followed by Bonferroni's post hoc test in (**g**) and two-tailed unpaired $t$-test in (**h**).

mTOR inhibitors (Supplementary Fig. 14A). In PLA assay, BIA diminished association between RICTOR and mTOR or between RICTOR and RPL19, but mTOR inhibitors did not affect any association in PANC-1 cells (Supplementary Fig. 14B), suggesting that BIA has potential as an effective anticancer agent controlling cancer cell survival although the experiment is only in vitro state.

## Discussion

The results of this study revealed that TMBIM6 enhances mTORC2 activity and assembly through direct binding and stimulation of $Ca^{2+}$ release. TMBIM6 disruption restricted primary tumor growth and impaired cancer cell metabolism. Also, we identified the small molecule compound BIA as a potential anticancer agent that prevents the binding between mTORC2 and TMBIM6.

Our finding indicated that TMBIM6 modulates glucose metabolism by regulating glycolysis and the PPP, which is critical for mTORC2 activity and signaling (Fig. 4a–d). Given the observed upregulation of TMBIM6 in breast, prostate, cervical, and lung cancers (Fig. 1a–d), changes in metabolism mediated by mTORC2 and especially AKT are likely a major mechanism for cancer progression. Moreover, TMBIM6 deletion suppressed GSH biosynthesis, which might be more susceptible to reactive oxygen species as well as lipid biosynthesis, inhibiting tumorigenesis. This study indicates that TMBIM6 is an important regulator of mTORC2 activity and tumor cell bioenergetics.

We determined that TMBIM6 serves as a signaling scaffold that recruits mTORC2 to the ER and thus promotes cell survival. This effect of TMBIM6 on cell growth differs from its classical role, which regulates ER stress-induced cell death[1,52]. The overall decrease of metabolic pathways and protein synthesis by

TMBIM6 disruption should have relieved ER stress at least partially. However, the absence of TMBIM6 rather increases ER stress, enhancing UPR signaling[1,52]. Since the characteristics of TMBIM6 have been examined only in stress conditions by ER stress inducer, TMBIM6 alteration does not affect ER stress response under non-stressed conditions[1,52]. At least in resting condition, TMBIM6 is not a simple ER stress regulator but rather a core protein enhancing mTORC2 recruitment and assembly, ultimately affecting AKT activation and cell proliferation, especially in cancer. Moreover, mTORC2 activity is associated with ribosomes and mitochondria-associated ER membranes[29,53]. In this study, we confirmed that mTORC2 is mainly localized at the ER membrane through its direct binding to TMBIM6 (Fig. 5, Supplementary Fig. 6), where it interacts with ER-bound ribosomes. AKT, PDK1, PTEN, and PI3K substrate have been observed in various organelles including the ER[31], indicating that the first step in AKT activation may also occur at the ER. Although mTORC2 also exists at the plasma membrane, ER-associated mTORC2 appears to be linked to prolonged AKT activation. Thus, the intracellular localization of mTORC2 may influence its activity toward AKT.

The ER is a $Ca^{2+}$-sequestering organelle; TMBIM6 is a $Ca^{2+}$ channel-like integral membrane protein located in the ER[1]. The blockade of $Ca^{2+}$ leakage caused by the $TMBIM6^{D213A}$ inhibited the recruitment of mTOR and ribosome (60S RPL19) to TMBIM6, resulting in a decrease of the phosphorylation of AKT (pAKT-S473) (Fig. 6b–e). It was also clearly shown that RPL19 specifically interacts with TMBIM6 upon $Ca^{2+}$ response, implying that local TMBIM6-induced $Ca^{2+}$ leakage is a requisite for mTORC2 recruitment to ribosomes. The interaction of TMBIM6 with RICTOR seems stable even in the $Ca^{2+}$ channel mutation condition, compared with that mTOR and RPL19 (Fig. 6c), indicating that the leaky $Ca^{2+}$ from TMBIM6 is related to the efficiency of mTORC2 complex formation and its subsequent AKT activation, but not directly related to RICTOR (Supplementary Fig. 10).

Similar to mTORC1 studies[54–56], we found that mTORC2 activity is sensitive to inhibition by BAPTA-AM (Fig. 6a, b, Supplementary Fig. 10A), suggesting that intracellular calcium is required for mTORC2 activation. In a recent study about the correlation of $Ca^{2+}$ channel and mTORC2, mTORC2 negatively regulates Mid1, an ER/plasma membrane-localized calcium channel regulatory protein. Decreased signaling of mTORC2 and its downstream target protein kinase, Ypk1, induced Mid1 activation[57]. The mTORC2 signaling regulates the $Ca^{2+}$-associated Mid1 as downstream effector under amino acid starvation, whereas the $Ca^{2+}$-leaky TMBIM6 first affects mTORC2 activation under basal condition. Our point is not general cytosolic $Ca^{2+}$ but the local $Ca^{2+}$ from the $Ca^{2+}$ leaky protein, TMBIM6, which is abrogated in the presence of BAPTA-AM.

The precise mechanism of regulation of mTORC2 by calcium through TMBIM6 $Ca^{2+}$ leaky characteristics is distinct in our model. First, we demonstrated that the ER leaky calcium plays a unique and critical role in mTORC2 activation and the resultant AKT in mammalian cells. Second, $TMBIM6^{D213A}$ had no effect on the interaction between TMBIM6 and RICTOR, ruling out the involvement of the local $Ca^{2+}$ in the direct binding with the mTORC2 subunit protein. However, $TMBIM6^{D213A}$ had a strong controlling effect on the interaction between mTOR and RICTOR or between RICTOR and ribosome, suggesting the involvement of the local $Ca^{2+}$ in the assembly of mTORC2 and further the association to ribosomal proteins.

We identified BIA as an inhibitor of the interaction between TMBIM6 and mTORC2, which ultimately blocks AKT activation and cancer progression. Mutations in PIK3CA or loss of PTEN constitutively activates PI3K-AKT-mTOR signaling in various human cancers[58,59]. To overcome the clinical limitation of 1st and 2nd mTORC1 antagonists including temsirolimus and everolimus[16,60], another candidate, "mTORC2"- controlling agent has been suggested in our study. Based on the interaction between TMBIM6 and mTORC2 and its biological effects (Fig. 5), the TMBIM6–mTORC2 axis is an actual signaling mechanism that can be target for cancer therapy. BIA has emerged as a potential candidate as an antagonist regulating TMBIM6-mTORC2 interaction and its related tumor growth even in cases that are resistant to mTOR inhibitors. Specifically, the characteristics of BIA indicate the dissociation between RICTOR and TMBIM6 through the regulation of TMBIM6-leaky $Ca^{2+}$ and the resultant inhibition of AKT activation impeding cancer formation.

In conclusion, our results demonstrate that since a state of high proliferation and metabolic activity is a hallmark of cancer, therapeutic strategies that disrupt the mTORC2–TMBIM6 interaction and/or inhibit TMBIM6 expression may be effective in the treatment of cancers characterized by AKT activation.

## Methods

**Plasmids.** The GST-RPL19 plasmid was generated by subcloning RPL19 cDNA from HeLa cells into the pGEX backbone vector. The pcDNA3-TMBIM6-HA, pcDNA3-TMBIM6-GCaMP3, and pLenti CMV/TO puro DEST (670-1)-TMBIM6-GCaMP3 were constructed by cloning. pCMV G-CEPIAer plasmid (#58215) and pRK-5-myc-RICTOR (#11367) were obtained from Addgene. Mutant constructs were generated using the QuikChange XL Site-Directed Mutagenesis Kit (200521, Stratagene, La Jolla, CA, USA) or Muta-Direct™ Site-Directed Mutagenesis Kit (15071, iNtRON biotechnology, Seongnam, Korea) according to the manufacturer's instructions.

**Antibodies and chemicals.** Antibodies against the following proteins were used in this study: TMBIM6/BI-1 (1:100, ab51905), and RPL19 (1:1000, ab128648) (all from Abcam, Cambridge, UK); RICTOR (1:2000, A300-459A) (Bethyl Laboratories, Montgomery, TX, USA); AKT (1:1000, #9272), GST (1:1000, #2622), mTOR (1:1000, #2972), mTOR (1:1000, #4517), NDRG1 (1:1000, #9408), phospho-Ser473-AKT (1:1000, #9271), phospho-Ser939-TSC2 (1:1000, #3615), phospho-Thr308-AKT (1:1000, #4056), TSC2 (1:1000, #4308), phospho-Thr346-NDRG1 (1:1000, #3217), SIN1 (1:1000, #12860), p70 S6 Kinase (1:100, #9202), RICTOR (1:1000, #2114), RICTOR (Sepharose bead conjugate, #5379), and RAPTOR (1:1000, #2280) (all from Cell Signaling Technology, Danvers, MA, USA); HA (1:2000, 11867423001) (Roche Diagnostics, Basel, Switzerland); actin (1:1000, sc-47778), Ki-67 (1:100, sc-15402), phospho-Thr450-AKT (1:1000, sc-293094), RPL19 (1:1000, sc-100830), and RPS16 (1:1000, sc-102087), (all from Santa Cruz Biotechnology, Santa Cruz, CA, USA). Secondary antibodies (1:10,000) for immunoblotting (Jackson ImmunoResearch, West Grove, PA, USA) and immunoprecipitation (1:5000, sc-2006) (Santa Cruz Biotechnology) were also used. Insulin (I3769), EGF (E9644), and IGF1 (I3769) were from Sigma-Aldrich. BAPTA-AM (B6769), BAPTA (B1212), and EGTA-AM (E1219) were from Thermo Fisher Scientific (Waltham, MA, USA).

**Cell culture and transfection.** HeLa, HT1080, and MEFs cells were maintained in Dulbecco's modified Eagle's medium (DMEM high glucose) supplemented with 10% fetal bovine serum (FBS) (Life Technologies, Grand Island, NY, USA) and 100 U/ml penicillin-streptomycin at 37 °C in a humidified 5% $CO_2$ incubator. Breast cancer cell lines (MCF7, MDA-MB-231, and SKBR3) and pancreatic cancer cell lines (PANC-1, Capan-1, MIA PaCa-2) were purchased from the Korean Cell Line Bank (Seoul, Korea) and were maintained in Roswell Park Memorial Institute 1640 medium supplemented with 10% FBS and 100 U/ml penicillin-streptomycin at 37 °C in a humidified 5% $CO_2$ incubator. T-REx™-293 cell lines was purchased from Thermo Fisher Scientific and maintained in DMEM supplemented with 10% FBS, 2 mM L-glutamine, and 100 U/ml penicillin-streptomycin. The cell lines were negative for mycoplasma contamination.

For transient transfection, $2–3 \times 10^6$ cells were seeded in 100 mm² dishes and cultured until they reached 70% confluence, then transfected using Lipofectamine 3000 (Invitrogen, Carlsbad, CA, USA) according to the manufacturer's instructions. To investigate signaling mechanisms, $5–10 \times 10^5$ cells were seeded in 100 mm² dishes. After serum starvation, cells were stimulated with insulin (100 ng/ml), IGF1 (100 ng/ml), or EGF (100 ng/ml). Cells subcultured for fewer than ten passages were used in the experiments.

For establishment of stable cell lines of human fibrosarcoma cells (HT1080) expressing pLenti CMV/TO puro DEST (670-1)-TMBIM6-GCaMP3 (TMBIM6-GCaMP3), cells were incubated with 8 µg/mL of Polybrene (Santa Cruz Biotechnology) and lentiviral particles harboring each gene followed by selection with puromycin dihydrochloride (Santa Cruz Biotechnology) for 1 week. Fresh

puromycin-containing medium was replaced every 3–4 days. For G-CEPIAer stable cell lines, cells were transfected with the G-CEPIA ER plasmid, and selected with G418 for 2 weeks.

**Immunoblotting**. Cells were lysed in radioimmunoprecipitation assay buffer (R4100; GenDEPOT, Katy, TX, USA) containing protease and phosphatase inhibitors (Sigma-Aldrich). The soluble fraction of cell lysates was isolated by centrifugation at 13,000 rpm for 30 min at 4 °C. The protein concentration was measured using a protein assay (Bio-Rad, Hercules, CA, USA) and 20–40 µg was resuspended in Laemmli buffer and incubated for 5 min at 95 °C before proteins were separated by sodium dodecyl sulfate polyacrylamide gel electrophoresis (SDS–PAGE) and transferred to a polyvinylidene difluoride membrane (Bio-Rad). After incubation in Tris-buffered saline containing 0.1% Tween 20 and 5% bovine serum albumin (BSA) for 1 h, the blot was incubated overnight at 4 °C with primary antibody. After washing, the membrane was incubated with horseradish peroxidase-conjugated secondary antibody for 1 h, and protein bands were visualized by enhanced chemiluminescence (ElpisBiotech, Daejeon, Korea). Uncropped images of blots are shown in Supplementary Fig. 15.

**Immunoprecipitation**. TMBIM6-HA was immunoprecipitated by the soft elution method[61]. Briefly, cell lysates were prepared in lysis buffer composed of 20 mM Tris-HCl (pH 8.0), 135 mM NaCl, 1.5 mM MgCl$_2$, 1 mM EGTA, 1% Triton X-100, and complete protease and phosphatase inhibitor cocktails (Sigma-Aldrich). Crude lysates (500 µg for each immunoprecipitation) were incubated with antibody (1–2 µg) for 6 h at 4 °C, followed by addition of protein A/G sepharose beads (Sigma-Aldrich) and incubation for an additional 1 h. Immunoprecipitates were washed five times with phosphate-buffered saline (PBS) containing 0.1% Triton X-100 or PBS before SDS–PAGE and immunoblotting.

**Reverse transcription quantitative real-time (qRT-)PCR**. Total RNA was extracted from cancer cells using TRIzol reagent (Invitrogen), and 3 µg was used to generate cDNA with the SuperScript III First-Strand Synthesis Kit (Invitrogen) according to the manufacturer's protocol. The sequences of the primer pairs used in this study are listed in Supplementary Data 3. P2220810 for mTOR, P130485 for SIN1, and P257029 for GβL were purchased from BIONEER (Daejeon, Korea). qRT-PCR was performed using the SYBR Green Reagent Kit (Applied Biosystems, Foster City, CA, USA) on an ABI PRISM 7700 Sequence Detection System (Applied Biosystems) under the following conditions: 95 °C for 5 min, followed by 40 cycles of 94 °C for 10 s, 51 °C–55 °C for 10 s, and 72 °C for 30 s. Reactions were performed in triplicate runs for each sample and were normalized to the level of actin (*ACTB*) or glyceraldehyde 3-phosphate dehydrogenase (*GAPDH*).

**Generation of TMBIM6 KO cells by CRISPR/Cas9 genome editing**. The CRISPR/Cas9 genome editing method was used to generate the TMBIM6 KO HT1080 cell line. The plasmid containing sequences targeting human TMBIM6 were designed and constructed from the pRGEN_TMBIM6 expression vector by ToolGen (Seoul, Korea). The guide sequence targeting exon 3 of human TMBIM6 was 5′-TGCAGGGGCCTATGTCCATATGG-3′. pRGEN_Scramble vector as a negative control was constructed using scrambled sequence (5′-GCACTACCAGA GGCTAACTCA-3′), which informed from Origene (#GE100003, pCas-Scramble Vector). The pRGEN_TMBIM6 vector or pRGEN_Scramble was mixed with pRGEN_Cas9-CMV and co-transfected into HT1080 and HeLa cells using Lipo-fectamine 3000; 48 h later, the cells were trypsinized and seeded in 96-well plates for isolation of individual clones by the limiting dilution method. The cells were cultured for over 1 week in DMEM containing 10% FBS and antibiotics. Single clones were expanded, and genomic DNA was purified from clones and used as template for PCR-based screening with the following three primers, one annealing outside and the others flanking the target region: F1, 5′-CGTTGCTGTGTGGTT ATTGG-3′; R1, 5′-TCAATCCTGCCTCTCCTGAT-3′; and Ftarget, 5′-TGCAGG GGCCTATGTCCATATGG-3′. KO clones produced only one PCR product, while normal clones produced two[62]. The PCR product of KO clones was purified using the JETsorb DNA Extraction Kit (Genomed, Leinfelden-Echterdingen, Germany) and the deletion was confirmed by sequence analysis.

**RNAi transfection**. Cells were transfected with small interfering (si)RNA using RNAiMax (Invitrogen) in Opti-MEM I Reduced Serum Medium (Invitrogen) and incubated for 12 h. The medium was changed and the cells were incubated for 48 h before analysis. mTOR, RICTOR, and SIN1 siRNAs were obtained from Santa Cruz Biotechnology. Negative control siRNAs were obtained from Bioneer (Daejeon, Korea) and used at 100 nM for effective knockdown without toxicity.

**Proliferation assay and cell viability assay**. Cell proliferation and cell viability assay were evaluated with crystal violet[63]. Briefly, cells were seeded at $2$–$5 \times 10^4$/ well in 24-well plates and treated with various chemical next day. After desired days, cells were fixed with 4% formaldehyde at room temperature (RT) for 5 min, and stained with crystal violet for 5 min. After washing and drying, the cells were solubilized with 1% SDS and the absorbance of the dissolved solution was measured at 595 nm.

**Gel filtration chromatography analysis**. Gel filtration chromatography was performed as previously described[64]. Briefly, TMBIM6 KO HT1080 cells were transiently transfected with HA-TMBIM6 plasmid in eight 10 cm dishes for each sample per gel filtration experiment; 24 h later, the cells were lysed in 1.0 ml CHAPS buffer [25 mM HEPES (pH 7.4), 150 mM NaCl, 1 mM EDTA, and 0.3% CHAPS] containing protease inhibitors (Sigma-Aldrich) and phosphatase inhibitors (phosphatase inhibitor cocktail sets 2 and 3; Sigma-Aldrich). The cell lysates were filtered through a 0.45 µm syringe filter; total protein concentration was adjusted to 5 mg/ml with CHAPS buffer, and 500 µl of the lysate was loaded onto a Superdex 200 Increase 10/300 GL column (GE Healthcare, Little Chalfont, UK; #28-9909-44). Chromatography was performed on an ÄKTAFPLC fast protein LC system (GE Healthcare; #18-1900-26) with CHAPS buffer. One column volume of the eluate was fractionated with 500 µl in each fraction at an elution speed of 0.3 ml/min. Aliquots (30 µl) of each fraction were separated by SDS–PAGE and proteins were detected by immunoblotting. The molecular weight resolution of the column was estimated using a gel filtration calibration kit (GE Healthcare; #28-4038-42) to determine the retention time.

**Glucose consumption and lactate production assay**. Glucose consumption and lactate production rates were evaluated using commercial kits (BioVision, Mountain View, CA, USA; K666-100 and K627-100, respectively) according to the manufacturer's instructions.

**Glycoprotein assay**. Glycoproteins were detected using a glycoprotein isolation kit (89804; Thermo Fisher Scientific) according to the manufacturer's instructions.

**GST pull-down assay**. The GST pull-down assay was performed using a commercial kit (21516; Thermo Fisher Scientific) according to the manufacturer's instructions. Briefly, GST-RPL19 was expressed in *Escherichia coli* and purified using GSH beads; the purified protein was bound to a GSH sepharose column. Soluble lysate (500 µg) from HeLa cells transfected with TMBIM6-HA or empty vector was loaded onto the GST-RPL19-bound column and rotated for 2 h at 4 °C. The samples were washed three times with wash buffer and then eluted with elution buffer and resolved by SDS–PAGE followed by immunoblotting.

**Protein synthesis assay**. The protein synthesis assay was performed using the Click-iT HPG Alexa Fluor 488 Protein Synthesis Assay Kit (C10428; Thermo Fisher Scientific) according to the manufacturer's instructions.

**3D culture assay**. The 3D culture assay was performed using a Cellrix 3D Culture System Kit (B1000-096; MediFab, Seoul, Korea) according to the manufacturer's instructions. Briefly, cultured cells were trypsinized and stained with DiI (2 g/ml) for 10 min, and then resuspended at $1 \times 10^6$ cells/ml in Cellrix Bio-Gel. The casting mold was removed from the gel and cells contained in Cellrix Bio-Gel were loaded onto the casting gel. After incubation on ice for 15 min, the gel was transferred to 96-well plates, and medium containing premixed vehicle [0.001% DMSO] and/or inhibitor was added, with medium change every 3 days. DiI fluorescence images were obtained with an Eclipse C1 confocal microscope (Eclipse C1; Nikon, Tokyo, Japan) and analyzed with AxioVision v.4.3 software (Carl Zeiss, Oberkochen, Germany).

**Wound healing assay**. Cells were cultured in a 6-well plate until they reached 70–80% confluence. A scratch was made through the cell monolayer using a sterile 200 µl pipette tip, and floating cells were removed by washing with culture medium. Fresh medium with or without small molecule inhibitor was added and the cells were allowed to migrate into the wound area for 24 h. Photomicrographs were acquired with an Axiovert 200 M fluorescence microscope (Carl Zeiss) immediately after wounding and after 24 h of incubation. The percent change in migration was determined by evaluating the change in wound width using ImageJ software (National Institutes of Health, Bethesda, MD, USA).

**Transwell cell migration and invasion assay**. Cell migration was evaluated using Transwell inserts (BD Biosciences, Franklin Lakes, NJ, USA) with an 8.0 µm pore polycarbonate membrane. Cells were trypsinized and washed in DMEM without serum, and $2 \times 10^4$ cells were added to the upper chamber and allowed to migrate for 12 h. Non-migrated cells were removed from the upper chamber using cotton swabs, and migrated cells were fixed and stained with crystal violet. For the invasion assay, BD BioCoat Matrigel invasion chambers with an 8.0 µm pore polyethylene terephthalate membrane in 24-well cell culture inserts (BD Biosciences) were used with 5% FBS added as the chemoattractant to the lower chamber. Cells were allowed to migrate for 12 h and then fixed, stained, and counted.

**Animal studies**. Six- to eight-week BklNbt:BALB/c/nu/nu old mice (Damul, Daejeon, Korea) were used for tumor xenografts. Mice were housed ($n = 5$/cage) in a fully climate-controlled room at constant temperature and humidity on a 12:12 h light/dark cycle with free access to food and water. Animal experiments were

performed in accordance with the Guide for the Care and Use of Laboratory Animals of Chonbuk National University Institutional Animal Care and Use Committee (Jeonju, Korea; approval nos. CBNU 2015-064, CBNU 2016-56, CBNU 2017-0026, and CBNU 2020-033) and related ethics regulations of the university.

TMBIM6 KO and WT HT1080 cells growing exponentially in culture were trypsinized and quantified by trypan blue exclusion, and $3–5 \times 10^6$ cells were resuspended in 0.1 ml PBS. The cells were subcutaneously injected into the flank of each mouse. For SMAiRNA or BIA treatment, $5 \times 10^6$ cells in 0.1 ml PBS were injected. When the tumor reached a weight of ~100 mg (7–10 days after inoculation), mice were randomly assigned to receive 1 mg/kg TMBIM6 SAMiRNA diluted in saline, 1 mg/kg BIA diluted in DMSO (final concentration: 10% v/v), or vehicle (saline or DMSO, 10% v/v). SAMiRNA was administered by tail vein injection every 3 days and BIA was intraperitoneally injected 5 days per week over 3 weeks. TMBIM6 SAMiRNA with the sequence 5′-AAGGCACUGCAUUGAUCUCUU-3′ and negative control SAMiRNA were obtained from Bioneer. After 25–28 days, mice were euthanized and solid tumors were dissected and tumor volume was recorded. Tumor size was measured with calipers. Tumor volume ($mm^3$) was calculated with the formula: [(shortest diameter)$^2 \times$ longest diameter]/2. Mice were evaluated twice weekly and were sacrificed by cervical dislocation when they showed signs of terminal illness such as hind leg paralysis and inability to eat or drink, and/or were moribund.

**Zebrafish experiments**. Fertilized zebrafish (*Danio rerio*) eggs were incubated at 28 °C in Danieau's solution and cultured under standard laboratory conditions. At 48 h post-fertilization, the embryos were dechorionated using forceps and anesthetized with 0.04 mg/ml tricaine. The embryos were then transferred to a modified agarose gel for microinjection. Before injection, tumor cells were labeled in vitro with 2 g/ml DiI, and ~100–500 cells were resuspended in serum-free DMEM. A 5 nl volume of the tumor cell solution was injected into the perivitelline cavity of each embryo using a Pv830 Pneumatic Picopump (World Precision Instruments, Sarasota, FL, USA) and non-filamentous borosilicate glass capillary needles (1.0 mm diameter; World Precision Instruments). The embryos were immediately transferred to culture water and maintained at 28 °C. Tumor growth and invasion were monitored every other day under a SM2645 fluorescence microscope (Nikon).

**Immunohistochemistry**. Formalin-fixed, paraffin-embedded cancer, and normal tissue samples from tissue array (BC081120d, PR1921c, CR1001a, and BC04002b, Biomax, Rockville, MD, USA) were analyzed by immunohistochemistry[65]. Briefly, after de-paraffinization and rehydration, tissue sections were subjected to 1× Target Retrieval Solution, pH 6.0 (DAKO, Glostrup, Denmark) and then incubated with peroxidase blocking solution (DAKO) for 10 min at RT. They were then washed with 1× TBST buffer (Scytek Lab, Logan, UT, USA) followed by a protein block (0.25% casein in PBS, DAKO) for 10 min at RT and incubated overnight at 4 °C with primary antibodies. Sections were incubated with secondary antibodies for 1 h at RT after rinsing in TBST buffer. AEC substrate chromogen (DAKO) was added and washed with deionized water. The slides counter stained with Mayer's hematoxylin (Sigma-Aldrich) were rinsed with tap water and mounted using an aqueous medium (Scytek Lab, USA). Ki-67 expression in xenograft tumors was determined as the percentage of positive cells per field and normalized by the total number of cells in each field.

**Immunofluorescence analysis**. Cells were seeded onto Lab-Tek II chamber slides (Thermo Fisher Scientific), rinsed in PBS, fixed in ice-cold methanol for 20 min, and then washed twice in PBS. The cells were blocked with 0.1% BSA (Sigma-Aldrich) for 30 min and then incubated overnight at 4 °C with primary antibody. After washing, cells were incubated with fluorescein isothiocyanate- or tetramethylrhodamine-conjugated secondary antibody for 1 h. The specimens were mounted with ProLong Gold anti-fade reagent and stained with 4-,6-diamidino-2-phenylindole (Invitrogen). Images were obtained on an LSM 510 confocal microscope (Carl Zeiss) and analyzed with AxioVision software.

**Proximity ligation assay (PLA)**. The PLA assay was performed using the Duolink in situ reagents (Sigma) according to the manufacturer's protocol. For image analyses, cells were acquired with the same laser parameters using the same image magnification.

**Live cell imaging**. Live cell imaging with BI-GCaMP3 and G-CEPIAer was performed using LSM 880 microscopy. Briefly, $2 \times 10^5$ HT1080 cells stably expressing GCaMP3-ML1 or G-CEPIAer were cultured in a 35 mm confocal dish. Changes in fluorescence levels were monitored for 20 min upon addition of BIA in $Ca^{2+}$-free external solution containing 145 mM NaCl, 5 mM KCl, 3 mM $MgCl_2$, 10 mM glucose, 1 mM EGTA, and 20 mM HEPES (pH 7.4). The intensity of fluorescence was measured using ZEN software.

**Human phospho-kinase array**. Human Phospho-kinase arrays were performed according to manufacturer's instructions (ARY003B, R&D Systems, Minneapolis, MN, USA). The quantification of pixels was performed using Fiji ImageJ software.

**Polysome profiling and poly(A) pull-down assay**. Polysome profiles were performed as described previously with minor adjustments. In brief, cells were incubated with a final concentration of 100 μg/ml cycloheximide for 10 min before harvest. The cells were then washed with 100 μg/ml of cycloheximide in PBS, collected in tubes, and resuspended in 1 ml of polysome lysis buffer (20 mM Tris-HCl pH 7.5, 100 mM NaCl, 10 mM $MgCl_2$, 0.4% IGEPAL, and 100 μg/ml cycloheximide) with 10 unit/ml RiboLock RNase Inhibitor (EO0381, Thermo scientific) and Xpert protease inhibitor cocktail (P3100, genDEPOT, Katy, TX, USA). The clarified lysates were loaded onto 10 ml linear 10–50% (w/v) sucrose gradients (prepared in 20 mM Tris-HCl pH 7.5, 100 mM NaCl, 10 mM $MgCl_2$, 100 μg/ml cycloheximide, 1X protease inhibitor cocktail, and 10 units/ml RNase inhibitor) and separated by centrifugation at 36,000 rpm for 2 hr at 4 °C in a P40ST swing rotor (Hitachi, JAPAN). The gradients were then fractionated with a Fluorinert FC-40 (F9755, Sigma-Aldrich) and 750 μl of the fractions were collected in tubes using an ISCO density gradient fractionation system.

For poly(A) pull-down assay, cells were lysed in buffer A (50 mM Tris-HCl [pH 7.4], 100 mM NaCl, 30 mM $MgCl_2$, 0.3% CHAPS, 40 U/ml RNase inhibitor, protease inhibitor cocktail, and 100 ug/ml cycloheximide) as previously report[29]. Lysates were clarified at 4 °C, 10 min at $8000 \times g$ and then were incubated with oligo (dT) cellulose (NEB) for 1 h at RT. The oligo (dT) cellulose was pelleted by centrifugation, and washed five times with buffer A. The bound fraction was eluted with elution buffer (100 mM Tris [pH 7.4], 500 mM NaCl, 10 mM EDTA, 1% sodium dodecyl sulfate (SDS), and 5 mM DTT). Purified ribosome fractions and the bound and unbound fractions were concentrated with Vivaspin 500 (Sartorius Stedim).

**Microarray analysis of HT1080 cells stably overexpressing TMBIM6**. Total RNA was extracted from transfected human cells using TRI reagent (MRC, Cincinnati, OH, USA) according to the manufacturer's instructions. Each RNA sample (30 μg) was labeled with cyanine (Cy)3- or Cy5-conjugated dCTP (Amersham, Piscataway, NJ, USA) by a reverse transcription reaction using SuperScript II reverse transcriptase (Invitrogen). The labeled cDNA mixture was concentrated by ethanol precipitation and resuspended in 20 μl of hybridization solution (Geno-Check, Daejeon, Korea), then mixed and applied to the OpArray Human Genome 35K array (OPHSV4; Operon Biotechnologies, GmbH) and covered with a MAUI FL hybridization chamber (Biomicro Systems, Salt Lake City, UT, USA). After incubation for 12 h at 62 °C, the slides were washed in 2× sodium chloride–sodium citrate (SSC) with 0.1% SDS for 2 min, followed by 1× SSC for 3 min and 0.2× SSC for 2 min at RT. The slides were dried by centrifugation at 3000 rpm for 20 s.

Hybridized slides were scanned with the GenePix 4000B scanner (Axon Instruments, Sunnyvale, CA, USA), and scanned images were analyzed with GenePix Pro v.5.1 (Axon Instruments) and GeneSpring GX v.7.3.1 (Silicon Genetics, Redwood City, CA, USA) software. Spots that were judged as substandard by visual examination of each slide—including those with dust artifacts or spatial defects—were excluded from further analysis. To filter out unreliable data, spots with a signal-to-noise ratio below 10 were also excluded. Data were normalized by global, lowess, print-tip, and scaled approaches for data reliability. Fold change filters included the requirement that up- and downregulated genes be present in at least 200% and 50% of controls, respectively. The data included groups of genes that behaved similarly across time course experiments and were clustered using GeneSpring GX v.7.3.1. We used an algorithm based on the Pearson correlation to identify genes with similar patterns.

**Microarray analysis of TMBIM6 KO and WT HT1080 cells**. Microarray analyses of TMBIM6 KO and WT HT1080 cells were performed by Ebiogen (Seoul, Korea). Global gene expression analyses were performed using GeneChip Human Gene 2.0 ST oligonucleotide arrays (Affymetrix, Santa Clara, CA, USA). Total RNA was isolated using TRIzol reagent. RNA quality was assessed with a 2100 Bioanalyzer (Agilent Technologies, Santa Clara, CA, USA) and quantified with an ND-1000 spectrophotometer (NanoDrop Technologies, Wilmington, DE, USA), and 300 ng of each RNA sample was used as input. Total RNA was converted to double-stranded cDNA. Using a random hexamer containing a T7 promoter, amplified cRNA was generated from the double-stranded cDNA template by in vitro transcription and purified with the Affymetrix sample cleanup module. cDNA was regenerated through random primer reverse transcription using a dNTP mix containing dUTP. The cDNA was fragmented by uracil DNA glycosylase and apurinic/apyrimidinic endonuclease 1 restriction endonucleases and end-labeled with biotinylated dideoxynucleotide in a terminal transferase reaction. Fragmented end-labeled cDNA was hybridized to the array for 16 h at 45 °C and 60 rpm according to the manufacturer's protocol. After hybridization, the array was labeled with streptavidin/phycoerythrin, washed in GeneChip Fluidics Station 450 (Affymetrix), and scanned using a GeneChip Array scanner 3000 7G (Affymetrix).

Image data were extracted from the scanned array using Command Console v.1.1 software (Affymetrix). The raw CEL file generated by the above procedure yielded expression intensity data that were analyzed with Expression Console v.1.1 software (Affymetrix). The Robust Multi-Average algorithm in the software was used for normalization. To classify co-expressed gene groups with similar expression patterns, we performed hierarchical clustering using Multi-Experiment Viewer v.4.4 software (www.tm4.org). The web-based Database for Annotation, Visualization, and Integrated Discovery tool was used to interpret the biological

function of DEGs, which were classified based on gene function information in the Gene Ontology and Kyoto Encyclopedia of Genes and Genomes databases (http://david.abcc.ncifcrf.gov/home.jsp).

**Chemical screening and synthesis of BIA**. High-throughput screening of the chemical library of the Korea Chemical Bank identified the chalcone scaffold as a potential TMBIM6 antagonist. BIA, one of the molecules in the chalcone scaffold, was selected for experiments. For the synthesis of BIA, a mixture of Aq. NaOH solution (NaOH 36 mg/$H_2O$ 0.25 mL) in EtOH (10 mL) maintained below 10 °C in an ice bath, 1-(2-aminophenyl)ethan-1-one (100 mg, 0.740 mmol) was added. The reaction mixture was stirred for 30 min at 10 °C. To the resulting mixture was then added 3-nitrobenzaldehyde (111.8 mg, 0.740 mmol). After being stirred for an additional 1 h at 10 °C, the reaction mixture was stirred at RT for further 26 h. The EtOH was removed in vacuo and the residue was dissolved in EtOAc. The organic layer was washed with 3N HCl and water. The combined organic layer was washed with brine, dried over sodium sulfate, and concentrated. A crude product was purified by column chromatography to obtain (E)-1-(2-aminophenyl)-3-(3-nitrophenyl)prop-2-en-1-one (164 mg, 82%). 1H NMR (300 MHz, DMSO-d6): δ 8.74 (s, 1H), 8.40–8.11 (m, 4H), 7.82–7.68 (m, 2H), 7.46 (s, 2H), 7.38–7.24 (m, 1H), 6.89–6.76 (m, 1H), 6.68–6.55 (m, 1H). 13C NMR (100 MHz, DMSO-d6): δ 190.6 (C), 152.7 (C), 148.9 (C), 139.9 (CH), 137.5 (C), 135.3 (CH), 135.0 (CH), 132.2 (C), 130.7 (CH), 126.7 (CH), 124.7 (CH), 123.3 (CH), 117.7 (CH), 117.4 (CH), 114.9 (CH). LC-MS (m/z): 269.1 [M + H]+. HPLC Purity 99.63%.

**Bioinformatics and statistical analysis**. Publicly available Gene Expression Omnibus (GEO) datasets for fibrosarcoma (GSE2719), cervical cancer (GSE63678), breast cancer (GSE31448), lung cancer (GSE19804), and prostate cancer (GSE69223) were used for bioinformatics analysis. GEO2R was used for TMBIM6 expression analysis. Overall survival analysis of cancer patient samples in TCGA datasets was performed using a web tool OncoLnc (http://www.oncolnc.org) and GEPIA2 (http://gepia2.cancer-pku.cn). One-way analysis of variance (ANOVA) with Tukey post hoc test, Two-Way ANOVA followed by Bonferroni post hoc test, and Student's unpaired $t$ test were performed using Prism v.8 software (GraphPad, San Diego, CA, USA). Data are expressed as mean ± SD, and $*p < 0.05$; $**p < 0.01$; $***p < 0.001$, $****p < 0.0001$, were considered statistically significant. In each case, the statistical test used is indicated, and the number of experiments is stated in the legend of each figure.

**Reporting summary**. Further information on research design is available in the Nature Research Reporting Summary linked to this article.

## Data availability

All data generated or analyzed in this study are included in the published article and Supplementary Information file or available from the authors upon reasonable request. All raw data used for generating figures are provided as a Source Data file. The microarray data has been deposited to GEO under accession numbers GSE153716. Source data are provided with this paper.

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

## Acknowledgements
The chemical library used in this study was kindly provided by the Korea Chemical Bank (http://www.chembank.org/) of the Korea Research Institute of Chemical Technology. This work was supported by the National Research Foundation of Korea, Republic of Korea (NRF-2017R1E1A1A01073796, 2017M3A9G7072719, 2017M3A9E4047243, and 2016R1A6A3A01013278).

## Author contributions
H.K.K., K.R.B. and R.P.J. performed the experiments. J.H.A. and S.H.P. screened the small-molecule inhibitor. H.J.Y. performed the metabolite analysis. J.S.H. and D.G.L. performed the ribosome profiling assay. K.W.K. performed the T4 phage display screening. H.R.K., H.K.K. and H.J.C. reviewed all data and prepared the paper. All authors reviewed the paper.

## Competing interests
The authors declare no competing interests.
