## [Peer Review File · Nature Communications]

Reviewers' comments:

Reviewer #1 (Remarks to the Author) (Expertise: mTOR, cancer metabolism):

In their manuscript, Kim et al identify the gene TMBIM6 as being important in tumorigenesis through its impact on mTORC2:ribosome assembly and therefore mTORC2 activity. The authors characterize transcriptional, metabolic, and transformation changes that occur upon TMBIM6 deletion or its suppression by a putative inhibitor, BIA. These experiments provide evidence for an interesting novel novel of this protein in regulating mTORC2 activity.

The most convincing experiment is the lack of rescue with the D213R mutant combined with rescue of the wild-type protein. The use of this system should be expanded, as described below. While a variety of assays are performed in support of their hypotheses, often times critical controls are missing, as described below.

Major Comments

1. Viability data reported in Figure 2 and metabolic and gene expression data presented in Figure 4 needs to be rescued by re-expression of the TMBIM6 cDNA, as opposed to the D213R mutant.
2. The authors should demonstrate that BIA does not have anti-proliferative or anti-cell migration activity in cell lines lacking TMBIM6, and does not impact AKT signaling in these lines. If a loss of cell proliferation is observed in KO cells, that result would indicate an off-target effect of the compound at that dose.

Minor Comments

1. The authors should show data validating their IHC technique provided in Figure 1. For example, by staining WT and KO HT-1080 cells using the same fixation protocol as used for tissues.
2. Survival data provided in Figure 1 are all from microarray using the same probe. The authors should report similar public data, readily available, from more recent studies that use RNASeq methods.
3. Figure 1G appears to be from a single replicate. The authors should provide additional replicates of this transcriptomic data. Primary transcriptomic data should be provided in a supplementary table and deposited in a public database.
4. Authors should indicate whether the enriched pathways reported in Figure 1H are significantly enriched.
5. Authors need to address whether the re-expression of TMBIM6 in Figure 3C is at physiologic levels and include blots where WT, KO, and HA-TMBIM are on the same membrane.
6. Supplementary Figure 3 is incorrectly referenced in the text as supplementary figure 2 at page 7-8
7. Authors claim that Supplementary Figure 3E shows dose dependent increase in pAKT, but total AKT levels seem to increase as well.
8. The statement "TMBIM6 is required for AKT activity and signaling" (Page 7) is overstated based on the data provided.
9. PLA assays (Figure 3 and 5) require a negative control for an uninvolved cytoplasmic protein.
10. The authors should describe how the BIA compound was identified.

Reviewer #2 (Remarks to the Author) (Expertise: mTOR signalling, ribosomes) :

TMBIM6 is a calcium channel like protein and is upregulated in many cancer types. Suppression of TMBIM6 promotes cell death and decreases tumor growth. In the current study, the authors now elucidate a mechanism as to how TMBIM6 could promote tumor growth. By knocking out TMBIM6 in HT1080 cells via CRISPR, they found overall defects in metabolism and decreased mTORC2 signaling. By gel filtration, they found that rictor and GbL fractionated at lower molecular weights

in the KO suggesting that mTORC2 dissociates from a large complex in the absence of TMBIM6. They then conducted studies to determine association of mTORC2 with ribosomes by proximity ligation assay (PLA) and immunoprecipitation. They found that the mTORC2/ribosome (via expression of RPL19) association is disrupted in TMBIM6 KO. TMBIM6 also cofractionated and associated with mTORC2 components and RPL19. Knockdown of rictor specifically abolished interaction of TMBIM6 with mTOR and SIN1. Mutagenesis of TMBIM6 revealed that the N-terminus and the cytosolic loop residues are required for interaction with rictor and Akt activation. Since TMBIM6 is a calcium channel-like protein, they then analyzed if calcium release from TMBIM6 could affect the interaction of mTORC2 with ribosomes. By PLA, they found that there was decreased rictor/mTOR and rictor/rpl19 interaction in the TMBIM6 mutant (D213A) that affects calcium release. Although rictor association with TMBIM6 was similar in the WT and mutant TMBIM6, there was a slight decrease in mTOR and strong decrease in rpl19 and rpl16 association. By screening a chemical library, they identified B1A as a TMBIM6 antagonist that decreases tumor cell proliferation at rather high concentrations. Akt phosphorylation, total expression of mTORC2 components and rpl19 are also decreased by this drug as well as release of calcium from the ER. B1A also regressed tumor growth in vivo. Based on these results, the authors propose that TMBIM6 promotes cancer progression via association with mTORC2 and Akt activation.

Overall, the results are interesting and support a role for TMBIM6 inhibition in preventing tumor growth. The role of TMBIM6 in promoting mTORC2/ribosome assembly is also interesting but would need further clarifications as detailed below.

1. Is the association of TMBIM6 with mTOR unique to mTORC2? Authors should also blot for raptor.
2. In Figure 4, the authors found decreased expression of proteins involved in glycosylation. They should also include the analysis of metabolites of the hexosamine pathway. mTORC2 has been shown to control this metabolic pathway (authors should cite PMID: 27570073 by Moloughney et al Mol Cell on page 9).
3. Akt phosphorylation should also be analyzed in Fig 5I.
4. While the PLA and immunoprecipitation assays to show association of mTORC2 and rpl19, the authors should more carefully analyze how TMBIM6 promotes association of mTORC2 with the ribosomes and whether this association is occurring in translating ribosomes. Polysome purification and analysis of mTORC2 association or cofractionation with TMBIM6 should be conducted.
5. The IC50 of B1A should be assessed. In Fig 7, it seems that high concentrations are needed to prevent cell proliferation of all the cell lines examined.
6. B1A (Fig 7B) seems to decrease total protein expression of mTOR and rpl19, as well as TMBIM6. There seems to be no strong effect on disruption of the complexes, other than slight decrease in TMBIM6. The authors should quantitate this figure more carefully to reflect comparative amount of total proteins.
7. Expression of total mTOR and rictor proteins should be included in Fig 7C.
8. The authors state in the Discussion that "This effect of TMBIM6 on cell growth differs from its classical role in ER stress." This has really not been addressed in the current study. Previous studies have already shown a role for mTORC2 in negative regulation of the ER calcium channel regulator Mid1 (in yeast) as a negative regulator of starvation response (PMID 27899413 Vlahakis et al JCB 2016). They should also cite this work to support the relationship of calcium signaling to mTORC2. Furthermore, they should verify that the disruption of TMBIM6 does not cause ER stress. The overall decrease in metabolic pathways and possible decrease in total ribosomal proteins (see comment 6) suggest that translation could also be decreased (which would relieve ER stress at least partially).

Reviewer #3 (Remarks to the Author) (Expertise: stress, ER) :

NCOMMS-19-31425-T

Comments to the authors,

In this work, Kim and colleagues show that TMBIM6 –a member of the highly conserved TMBIM family of proteins –interacts with components of the mTORC2 complex to regulate cancer progression. Indeed, they first show that TMBIM6 is overexpressed in several different types of carcinomas and its expression correlates with poorer patient overall survival. Using loss-of-function approaches, the authors show that TMBIM6 promotes cell migration and invasion in vitro and tumor growth in vivo. These experiments are well performed and highlight the known role of TMBIM6 as a protumorigenic protein. In fact, work by this same group has previously shown that TMBIM6 overexpression promotes metastasis through the regulation of cell migration and invasion (PMID: 20118983). At the molecular level they show that TMBIM6 is required for the assembly of the mTORC2 complex, leading to improved glucose homeostasis and promoting cancer progression. In short, TMBIM6 interacts and tether the mTORC2 complex and the ribosomes to the ER where it would promote its assembly through TMBIM6 calcium leak channel function. Finally, the authors described a putative TMBIM6 antagonist with anti-tumor properties.

Overall, the in vitro and in vivo results are very interesting and well-presented. However, I have some concerns regarding the proposed role of TMBIM6 as an mTORC2 regulator and its connection to cancer progression. Some key control experiments are currently missing, making the interpretation of the data difficult. The statistical analysis used throughout the manuscript should also be revised.

A revised version of this manuscript should still be suitable for publication in Nature Communications.

Major concerns:

1. Model of choice and generality: In Figure 1, the authors showed that TMBIM6 is overexpressed in several different cancer types, including cervical, endometrial and vulvar, breast, lung and prostate cancer. They also show that high TMBIM6 expression correlates with poorer prognosis in patients with squamous cell carcinoma and endocervical adenocarcinoma, esophageal carcinoma, skin cutaneous melanoma, head and neck squamous carcinoma and brain lower grade glioma, most of them of epithelial origins (Figure 1). However, they perform most of the loss-of-function experiments (migration, invasion, and tumor growth and mTORC2 biochemical experiments) in HT1080 cell lines, which is a fibrosarcoma cell line of mesenchymal origin. Two comments on this: (1) the authors should show, if it were, TMBIM6 expression and survival data on fibrosarcoma, since it will correlate better with the observations performed in HT1080 TMBIM6 KO cells and (2) the authors should repeat some key experiments (mTORC2 assembly, AKT phosphorylation and tumor growth) in an additional TMBIM6 KO cell line cancer model directly related to the cancers shown in figure 1. Most of the experiments connecting TMBIM6 with mTORC2 were performed in only one cancer cell line.

2. Is the TMBIM6 KO HT1080 cell line a pool or was derived from a single clone? In the case of it being a clonal population, the authors should repeat some key experiments (i.e. AKT phosphorylation and the assembly of the mTORC2 complex) with additional clones. Throughout the manuscript, the authors compare the HT1080 TMBIM6 KO cells with wild-type controls when they should have been compared with a matched CRISPR/Cas9 scramble or mock control. I am worried about CRISPR/Cas9 off-target effects.

3. From the data shown in Figures 3F (PLA assay), 3G and supplementary figure 3F (IP assays) the authors suggest that "TMBIM6 regulates the assembly of mTORC2 components and promotes the

physical association between mTORC2 and the ribosomes". However, the authors did not rule out the possibility that TMBIM6 expression may regulate the concentration of proteins forming the mTORC2 complex (the authors also showed that TMBIM6 KO reduces global protein synthesis). The same seems to be the case with mTOR and RICTOR co-localization with PDI (Supplementary Figure 4): TMBIM6 KO cells seem to exhibit decreased levels of these proteins, hence, decreased co-localization. At the very least, the authors should compare the endogenous protein levels of mTOR, RICTOR, RPL19 and RPS16 between HT1080 WT and TMBIM6 KO cell lines. If willing, they should also check the mRNA levels of these genes. Is it reduced assembly, reduced protein expression or both?

4. To assess the putative role of calcium released from TMBIM6 on mTORC2 assembly, the authors use a TMBIM6-GCaMP3 construct that they say "... is based on the finding that leaky calcium but not ER lumen [calcium] is detected upon binding of Ca²⁺ to the cytosol of this protein (Supplementary Fig. 6B)". To what side of TMBIM6 was GCaMP3 attached? The latest structural models based on the crystallization of the bacterial homolog BsYetJ suggest that TMBIM proteins are composed by seven transmembrane domains with the N-terminus facing the cytosol and the C-terminus facing the intraluminal space (see for example: PMID:24904158 and PMID:30930064). The authors should comment and cite these structural works and incorporate them into their models presented in Supplementary Figure 6C, where they only show the 6 transmembrane models for TMBIM6. How are the authors completely positive that they are not measuring ER calcium content?

5. In Figure 6, the authors reconstituted TMBIM6 KO cells with either TMBIM6-HA or the channel mutant TMBIM D213A and assessed calcium leak (Figure 6A), mTORC2 assembly (Figure 6B) and AKT phosphorylation (Figures 6D and 6E). However, to correctly interpret these experiments, a comparison of the total reconstituted levels of TMBIM6 and TMBIM6 D213A proteins in KO cells is necessary. The authors should also check the ER localization of the WT and the D213A mutant proteins by immunofluorescence. Differential partial reconstitution of these proteins may account for the differences in mTORC2 assembly and AKT phosphorylation.

6. Specificity of BI: from the data presented in Figures 7 and 8, and Supplementary Figure 7, the authors suggest that BIA acts as a TMBIM6 antagonist, blocking ER calcium release, leading to mTORC2 disassembly and inhibition of TMBIM6-associated tumorigenicity. Although it is clear that BIA reduces cell proliferation, migration and invasion, it is not so clear that these effects dependent on BIA's function as a TMBIM6 antagonist. Does BIA work when TMBIM6 is knocked out? There are also several key control experiments missing from figures 7 and 8 that preclude the interpretation of these experiments. For example, total mTOR and RICTOR levels are missing from the IP shown in Figure 7C. The authors should also report the effect of BIA on cell viability under the conditions reported in Figure 7A. The data supporting the role of BIA as an inhibitor of the interaction between TMBIM6 and mTORC2 is currently not convincing enough. The authors could perform PLA as in previous experiments to strengthen this data.

7. Overstatements and unsupported evidence: In page 18 the authors state that "BIA is a newly identified TMBIM6 antagonist that disrupts the TMBIM6-mTORC2 interaction, leading to the inhibition of tumor growth even in cases that are resistant to mTOR inhibitors". However, the authors have not provided evidence suggesting this. They have only shown that BIA decreases tumor growth in vitro and in vivo and that it works in combination with other mTORC2 inhibitors to kill HT1080 cells. They have not shown that these effects are a direct consequence of BIA's putative role as a TMBIM6 antagonist or as a disruptor of TMBIM6-mTORC2 interaction. Throughout the manuscript there are several instances where the authors jump from a phenotypic observation to a molecular mechanism whose causal connection to the observation has not been directly demonstrated.

8. Statistics: The authors should review the statistical procedures used throughout the manuscript. When there are two independent variables (e.g. genotype and time) the authors should use a Two-

Way ANOVA followed by post hoc test instead of multiple two-tailed unpaired Student's t-tests. Figures where Two-Way ANOVA should be used include Figures 1E, 2A, 2D, 2E, 2H, 3B, 3F, 4A, 4E, 4H, 8E and 8G. When there is one independent variable but more than two groups are compared, the authors should use a One-Way ANOVA followed by post hoc test. Examples of these include Figures 2I, 2K, 6D and 6E. Finally, the authors should specify p-values as follows: *, $p < 0.05$; **, $p < 0.01$; ***, $p < 0.001$.

Minor comments:

1. For in vitro migration and invasion experiments shown in Figures 2B and C, the authors should also express the data as the percentage of migrating and invading cells and not only as fold changes compared to WT.
2. In Figure 3C the authors should show the blot results of the second mTORC2 target NDRG1 (pT346) in the HT1080 TMBIM6 KO cells reconstituted with TMBIM6-HA.
3. In page 7, the authors describe the effects of insulin referencing supplementary figure 2D and 2E. This is not correct; it should read Supplementary Figure 3D and 3E respectively.
4. There are many misplaced conclusions throughout the manuscript. For example, in page 11 the authors say that "...TMBIM6 interacts with mTORC2 and ribosomes and that this interaction is important for the kinase activity of mTORC2". However, the effects of TMBIM6-mTORC2 interaction in AKT phosphorylation are explored in the following paragraph (Figure 5H).
5. How does calcium leak from the ER increase mTORC2 assembly efficiency? The authors should discuss this in more detail.

Reviewer #4 (Remarks to the Author) (expertise: cancer, zebrafish model) :

This is a continuous work of the authors' previous claims that TMBIM6/BI-1 promotes tumor growth and progression (ref 9). In this study, the authors provide new evidence of potential signaling events of the TMBIM6/BI-1-mTORC2-AKT axis in different cancer types. First, they show that in several cancer types TMBIM6/BI-1 expression is elevated, which is largely confirmed by GCTA analysis. Using in vitro KO technology, they show that deletion of TMBIM6/BI-1 in cancer cells retards cancer cell proliferation, migration, and tumor formation. They then defined signaling pathways that potential involved in these tumorigenic activities. Finally, they screened a chemical library to identify a potential inhibitor BIA, which inhibits tumor growth in mice and in fish.

Comments:

- 1) This work covers almost all respects of cancer development, including cancer cell proliferation, migration, survival, tumor formation, and metastasis. From the provided data, it is hard to believe that TMBIM6/BI-1 is a master regulator of cancer development. In particular, the characterized signaling events do not support TMBIM6/BI-1 as a master regulator. The authors should focus on a particular signaling and activity to obtain an in-depth mechanistic insight, but not a wikipedia-associated cancer development.
- 2) The TCGA data and their own experimental findings do not completely match. What about lung cancer, pancreatic cancer, and prostate cancer in TCGA? If TMBIM6/BI-1 is also highly expressed in these cancers, but not associated with poor survival, what does this mean? I am almost certain that some cancer types express high levels of TMBIM6/BI-1, but lack clinical correlation. The authors should not only choose the results favor to your findings.

3) The zebrafish cancer model is useless in this experimental setting. Are BIA-treated cancer cells dead in the fish body? How do they discriminate living cancer cells from dead cancer cells? How do they know that BIA is active in fish body?

4) BIA looks like a non-specific chemical compound as most chemical compounds do. These data can only be used as indirect supportive data.

5) The manuscript needs language editing.

Reviewers' comments:

Reviewer #1 (Remarks to the Author) (Expertise: mTOR, cancer metabolism):

In their manuscript, Kim et al identify the gene TMBIM6 as being important in tumorigenesis through its impact on mTORC2:ribosome assembly and therefore mTORC2 activity. The authors characterize transcriptional, metabolic, and transformation changes that occur upon TMBIM6 deletion or its suppression by a putative inhibitor, BIA. These experiments provide evidence for an interesting novel novel of this protein in regulating mTORC2 activity.

Response: We thank the reviewer for his/her insightful comments and criticisms.

The most convincing experiment is the lack of rescue with the D213R mutant combined with rescue of the wild-type protein. The use of this system should be expanded, as described below. While a variety of assays are performed in support of their hypotheses, often times critical controls are missing, as described below.

Major Comments

1. Viability data reported in Figure 2 and metabolic and gene expression data presented in Figure 4 needs to be rescued by re-expression of the TMBIM6 cDNA, as opposed to the D213R mutant.

Response: We agree with the reviewer's comments. In accordance with the reviewer's recommendation, we performed cell proliferation, metabolic, and gene expression analysis by re-expression of the TMBIM6 and TMBIM6-D213A. As shown in the revised Fig. 6, the inhibition of cell proliferation was rescued by re-expression of TMBIM6, not TMBIM6 D213A. Consistently, the expression of genes related to glycolysis, pentose phosphate pathway (PPP), GSH biosynthesis, and *de novo* lipogenesis was rescued by re-expression of TMBIM6, but not by that of TMBIM6 D213A. Furthermore, the levels of metabolites from glycolysis, tricarboxylic acid cycle, PPP, and hexosamine biosynthesis pathway (HBP) in TMBIM6 KO cells were rescued in the re-expressing condition of TMBIM6, not TMBIM6 D213A. Since the presence of D213A starts from the original Fig. 6, the reviewer's comment was answered in the revised Figure 6. The updated Fig. 6, and Supplementary Fig. 8, and their matched explanation in this manuscript are given below.

Figure 6

Figure 6. TMBIM6-induced Ca^{2+} leakage affects mTORC2 assembly and the association between mTORC2 and ribosomes. (A) Immunofluorescence images and fluorescence intensity (right) of TMBIM6-GCaMP3 and TMBIM6 D213A-GCaMP3 in the presence or absence of 10 μM BAPTA-AM. ($n = 3$ independent experiments). Scale bar, 15 μm . (B) PLA between Rictor and mTOR or between Rictor and RPL19 (red dots) in empty vector, TMBIM6, or D213A-transfected HT1080 cells. ($n = 3$ independent experiments). Scale bar, 15 μm . (C) Immunoblot analysis of the immunoprecipitates with Anti-HA antibody and input of cell lysates with the indicated antibodies. (D) Immunoblotting and quantification of phosphorylation of AKT in TMBIM6 KO HT1080 cells transfected with empty vector, TMBIM6 WT, and TMBIM6 D213A. ($n = 3$ independent experiments). (E) Immunofluorescence images and quantification of AKT phosphorylation ($n = 3$ independent experiments, total of 9 images). Scale bar, 15 μm . (F) Proliferation analysis of empty vector, TMBIM6, and TMBIM6 D213A-expressed HT1080 cells ($n = 3$ independent experiments). (G) The quantification analysis of mRNA levels of glycolysis- and PPP-related genes in empty vector, TMBIM6, and TMBIM6 D213A-rescued TMBIM6 KO HT1080 cells, as determined by qRT-PCR. Quantification data represent the expression level of genes compared with those in normalized WT HT1080 cells (red line, $n = 3$ independent experiments). (H-I) Glucose consumption and lactate production in empty vector, TMBIM6, and TMBIM6 D213A-rescued TMBIM6 KO HT1080 cells ($n = 3$ independent experiments). (J) Metabolite analysis in empty vector, TMBIM6, and TMBIM6 D213A-rescued TMBIM6 KO HT1080 cells. Quantification data represent the metabolite level compared with those in empty vector-rescued TMBIM6 KO HT1080 cells ($n = 2$ independent experiments). Data represent mean \pm SD. Statistical differences were detected with two-tailed unpaired Student's t-tests (J), one-Way ANOVA followed by Tukey's test (D, E, H, I), and two-Way ANOVA followed by Bonferroni's test (A, B, F, G).

Supplementary Fig. 8

Supplementary Fig. 8

Supplementary Fig. 8 TMBIM6 regulates mTORC2-dependent metabolism. (A) Immunoblotting of TMBIM6-HA in empty vector, TMBIM6, TMBIM6 D213A-rescued TMBIM6 KO HT1080 cells. (B-C) mRNA levels of GSH biosynthesis genes (B) and *de novo* lipid biosynthesis genes (C) in empty vector, TMBIM6, TMBIM6 D213A-rescued TMBIM6 KO HT1080 cells, as determined by qRT-PCR. Quantification data represent the expression level of genes compared with those in normalized WT HT1080 cells (red line, $n = 3$ independent experiments). (D) Metabolite analysis in empty vector, TMBIM6, TMBIM6 D213A-rescued TMBIM6 KO HT1080 cells. Quantification data represent the metabolite level compared with those in empty vector-rescued TMBIM6 KO HT1080 cells ($n = 2$ independent experiments). (E) Polysome profiling performed in empty vector, TMBIM6, TMBIM6 D213A-rescued TMBIM6 KO HT1080 cells by sucrose gradient fractionation. Data represent mean \pm SD. Statistical differences were detected with one-Way ANOVA followed by Tukey's test (D), and two-Way ANOVA followed by Bonferroni's test (B, C).

Manuscripts

To investigate the importance of Ca^{2+} leakage through TMBIM6, we stably rescued TMBIM6 expression of TMBIM6 or TMBIM6 D213A in TMBIM6 KO HT1080 cells, and then determined cell proliferation rates. TMBIM6 and D213A were expressed at a comparable level in the cells (Supplementary Fig. 8A). We found that cell proliferation was restored in TMBIM6-rescued KO cells, not D213A (Fig. 6F). Furthermore, the expression of genes related to the glycolysis and PPP was recovered in TMBIM6-rescued KO cells, not D213A, recovering glucose consumption and lactate production (Fig. 6G-I). We next

analyzed the expression of genes related to GSH biosynthesis, and *de novo* lipogenesis. Consistently, the expression of genes was restored in TMBIM6-rescued KO cells, not D213A (Supplementary Fig. 8B-C). By mass spectrometry, the levels of metabolites from glycolysis, tricarboxylic acid cycle, PPP, and HBP have restored in TMBIM6 rescued cells compared with TMBIM6 D213A mutant rescued cells (Fig. 6J, Supplementary Fig. 8D). Moreover, the patterns of ribosome profiling were same in all the cells, that is consistent to Supplementary Fig. 8E; TMBIM6 is independent of ribosome maturation. These results suggest that TMBIM6 regulates metabolic pathways through its characteristics “Ca²⁺ leakage-associated mTORC2 activation”.

2. The authors should demonstrate that BIA does not have anti-proliferative or anti-cell migration activity in cell lines lacking TMBIM6, and does not impact AKT signaling in these lines. If a loss of cell proliferation is observed in KO cells, that result would indicate an off-target effect of the compound at that dose.

Response: In accordance with the reviewer’s suggestion, we performed cell proliferation, migration assay, and immunoblotting for AKT signaling in BIA-treated TMBIM6 KO cell lines. As shown in Fig. 7A, and Supplementary Fig. 10E-G, the proliferation, migration, and the status of AKT phosphorylation in TMBIM6 KO HT1080 cells did not change in the presence of BIA. However, the cell viability at high concentrations of BIA was significantly decreased (Supplementary Fig. 10E), indicating that BIA has an on-target effect against TMBIM6 up to 10 μ M. The updated Fig. 7 and Supplementary Fig. 10E-G, and their explanation in the manuscripts are given below.

Figure 7

Figure 7. BIA, a TMBIM6 antagonist, suppresses tumor growth. (A) Proliferation of TMBIM6 WT HT1080, TMBIM6 KO HT1080, MCF7, MDA-MB-231, and SKBR3 cells treated with BIA (n = 3 independent experiments).

Supplementary Fig. 10

Supplementary Fig. 10

Supplementary Fig. 10 Inhibitory function of BIA in cancer progression. (A) TMBIM6 mRNA levels in various cancer cell lines. mRNA levels in each cells were normalized to the level of β -actin. (B) Cell viability was measured in the indicated concentrations of BIA-treated cancer cells at three days ($n = 3$ independent experiments). (C) Proliferation of cells stably expressing TMBIM6 or the vector was analyzed after 1 day of treatment with indicated concentrations of BIA. (D) PLA between RICTOR and the following protein, mTOR, RPL19 and RPS 16 was performed in HT1080 cells with 10 μ M BIA. Scale bar, 20 μ m. Right, quantification of red dots ($n = 5$ independent experiments). (E-G) Immunoblotting of phosphorylation of AKT (E), cell viability assay (F), and images of migrated cells and its quantification assay (G) were performed in TMBIM6 KO HT1080 cells treated with indicated concentrations of BIA. Quantification of cell viability and migrated cells in the BIA-treated cells normalized to control cells ($n = 3$ independent experiments). (H) The wound healing assay was performed with HT1080 cells treated with 2 μ M BIA. Representative images (left) and quantification (right; $n = 3$ independent experiments) are shown (I and J). Representative images of *in vivo* tumors from xenograft experiments were shown in Figure 8 ($n = 9$ and 11 for control and treatment groups for HT1080 cells (I), and n

= 6 mice per group for MDA-MB-231 cells (J)). Data represent mean \pm SD. Statistical differences were detected with one-Way ANOVA followed by Tukey's test (F), and two-Way ANOVA followed by Bonferroni's test (B, C, D, H).

Manuscripts

To clarify whether anti-proliferation effect by BIA at that dose is an off-target effect, we examined cell proliferation in TMBIM6 KO HT1080 cells. The cell proliferation rate and AKT phosphorylation in TMBIM6 KO HT1080 cells were same in the presence or absence of BIA (Fig. 7A, Supplementary Fig. 10E) with the exceptions of high concentrations "20 and 30 μ M" (Supplementary Fig. 10F), suggesting that BIA has on-target effect on TMBIM6 up to 10 μ M.

Cell migration and invasion are representative *in vitro* markers for cancer characteristics. BIA treatment decreased cell migration in HT1080, MCF7, MDA-MB-231, and SKBR3 cells (Fig. 8A), not TMBIM6 KO HT1080 cells (Supplementary Fig. 10G). Consistently, the wound healing assay showed that BIA significantly inhibited cell migration compared with DMSO-treatment (Supplementary Fig. 10H). Moreover, cell invasion in MDA-MB-231 and HT1080 cells was also decreased in the BIA-treated cells (Fig. 8B). We also evaluated the effect of BIA on cell proliferation in a three-dimensional (3D) culture, which more accurately recapitulates *in vivo* tumor growth. Under the BIA treatment condition, the numbers and the size of spheroids formed by the 3D cultured cells were significantly decreased, not showing multi-acinar structures (Fig. 8C).

Minor Comments

1. The authors should show data validating their IHC technique provided in Figure 1. For example, by staining WT and KO HT-1080 cells using the same fixation protocol as used for tissues.

Response: In accordance with the reviewer's suggestion, we performed IHC in WT and KO HT1080 cells using the same protocol as used for tissues. In the revised manuscript, we have included a new image in Figure 1F, showing that TMBIM6 expression is only detected in the TMBIM6 WT cells, and not in TMBIM6 KO cells. The updated Fig. 1F is given below.

Figure 1. TMBIM6 expression increased in cancer patient samples. (A-E) TMBIM6 expression was analyzed using the Gene Expression Omnibus database from NCBI. fibrosarcoma (GSE2719), cervix (GSE63678), breast (GSE31448), lung GSE19804, and prostate (GSE69223) datasets are presented. (F) Representative immunohistochemical staining of TMBIM6 on tissue microarrays containing fibrosarcoma, cervix, breast, lung, and prostate tissue and adjacent normal tissues. TMBIM6 WT and KO HT1080 cells were used as a control for validation of the method. Right; quantification data of TMBIM6 expression.

Scale bar, 100 μm . (brown: positive antibody staining, blue: hematoxylin for nuclear staining).

2. Survival data provided in Figure 1 are all from microarray using the same probe. The authors should report similar public data, readily available, from more recent studies that use RNASeq methods.

Response: We performed an analysis of survival data from GEPIA2 tool (<http://gepia2.cancer-pku.cn>) and OncoLnc (<http://www.oncolnc.org>). GEPIA2 (Gene Expression Profiling Interactive Analysis 2) is developed by Zefang Tang and colleagues for analyzing the RNA sequencing expression data of 9,736 tumors and 8,587 normal samples from the TCGA and the GTEx projects (Tang et al., 2019). OncoLnc contains survival data for 8,647 patients from 21 cancer studies performed by The Cancer Genome Atlas (TCGA), along with RNA-SEQ expression for mRNAs from TCGA (Anaya, 2016). Both tools allow researchers to study a specific gene and facilitate quick investigation in a single click. The updated Figure 1G and Supplementary Fig. 1A and corresponding text are given below.

Figure 1G

Figure 1. TMBIM6 expression increased in cancer patient samples. (G) Kaplan-Meier curves showing the overall survival analysis in patients with high and low expression of TMBIM6 using GEPIA2 tool. P value with log-rank analysis. BRCA; breast invasive carcinoma, CESC; cervical squamous cell carcinoma and endocervical adenocarcinoma, SARC; sarcoma, LUAD; lung adenocarcinoma.

Supplementary Fig. 1

Manuscripts

To further examine whether the TMBIM6 expression level in tumors is associated with prognosis, we analyzed the correlations between TMBIM6 expression and overall survival

(OS) using Gene Expression Profiling Interactive Analysis 2 tool (GEPIA2, <http://gepia2.cancer-pku.cn>) from the TCGA and the GTEx projects and OncoLnc (<http://www.oncolnc.org>) from The Cancer Genome Atlas (TCGA). We found that patients with high TMBIM6 expression had poor survival, compared with those with low TMBIM6 expression in breast invasive carcinoma (BRCA, $P = 0.017$ by GEPIA2, $P = 0.0123$ by OncoLnc), cervical squamous cell carcinoma and endocervical adenocarcinoma (CESC, $P = 0.00043$ by GEPIA2, $P = 0.0486$ by OncoLnc), sarcoma (SARC, $P = 0.034$ by GEPIA2), and lung adenocarcinoma (LUAD, $P = 0.031$ by GEPIA2, $P = 0.0367$ by OncoLnc), as analyzed by the Kaplan–Meier method with log-rank test (Fig. 1G, Supplementary Fig. 1A). Additionally, we confirmed OS in several cancers including prostate adenocarcinoma (PRAD, $P = 0.028$ by GEPIA2), pancreatic adenocarcinoma (PAAD, $P = 0.028$ by GEPIA2), esophageal carcinoma (ESCA, $P = 0.0129$ by OncoLnc), skin cutaneous melanoma (SKCM, $P = 0.0336$ by OncoLnc), head and neck squamous cell carcinoma (HNSC, $P = 0.0008$ by OncoLnc), and brain lower-grade glioma (LGG, $P = 0.0194$ by OncoLnc) (Supplementary Fig. 1B). These data suggest that TMBIM6 has a potential clinical value as a predictive biomarker for disease outcome in several cancers.

3. Figure 1G appears to be from a single replicate. The authors should provide additional replicates of this transcriptomic data. Primary transcriptomic data should be provided in a supplementary table and deposited in a public database.

Response: We added additional replicates transcriptome data, and detail primary transcriptomic data and showed them as supplementary table 1 as per the reviewer's suggestion. The expressed genes related to apoptotic process, proliferation, and metabolic pathways were greatly changed in TMBIM6 KO HT1080 cells compared with WT cells. The updated Figure 1H is given below. Figure 1G in the original version is changed into Figure 1H in the revised version.

Figure 1H

4. Authors should indicate whether the enriched pathways reported in Figure 1H are significantly enriched.

Response: We selected Gene Ontology related to cancer characteristics on Quick GO (<https://www.ebi.ac.uk/QuickGO/>) supplied at EMBL's European Bioinformatics Institute (EMBL-EBI). Angiogenesis (GO:0001525), apoptotic process (GO:0006915), cell cycle (GO:0007049), cell death (GO:0008219), cell migration (GO:0016477), cell proliferation (GO:0008283), DNA repair (GO:0006281), extracellular matrix (GO:0031012), secretion

(GO:0046903), glycosylation (GO:0070085), lipid biosynthesis (GO:0008610), metabolic pathway (GO:0008152), and transport (GO:0006810) were confirmed in WT and KO cells. The updated Figure 1I is given below. Figure 1H in the original version is changed into Figure 1I in the revised version.

Figure 1

5. Authors need to address whether the re-expression of TMBIM6 in Figure 3C is at physiologic levels and include blots where WT, KO, and HA-TMBIM6 are on the same membrane.

Response: In accordance with the reviewer's suggestion, we performed immuno-blotting in TMBIM6 KO HT1080 cells with re-expression of TMBIM6 using pLenti-vector for expression of physiologic levels, and confirmed the expression of TMBIM6 by using qRT-PCR. In the revised manuscript, we have included a new image in Figure 3C, showing that TMBIM6 expression is the same between TMBIM6 WT cells and the re-expression of TMBIM6 in KO cells. The updated Figure 3C and corresponding text are given below.

Figure 3. TMBIM6 regulates mTORC2 activation. (A) Expression and phosphorylation of 43 proteins were examined by Proteome Profile Human Phospho-Kinase Array in WT and

TMBIM6 KO HT1080 cells. Right; the relative phosphorylation of indicated proteins was quantitated by ImageJ program. (B) pAKT, pTSC2, and pNDRG1 were analyzed in TMBIM6 KO and WT HT1080 cells using Western blots (left) and were normalized to total proteins of WT cells (right; n = 5 independent experiments). (C) Immunoblotting, RT-PCR and its gene quantification analysis were performed in TMBIM6-HA-stably expressing or non-expressing TMBIM6 KO cells (n = 3 independent experiments).

6. Supplementary Figure 3 is incorrectly referenced in the text as supplementary figure 2 at page 7-8

Response: We apologized for the incorrectly referenced Supplementary Figure in the text, and appropriately referenced “Supplementary Fig. 3F and Supplementary Fig. 3G” in the updated text, respectively.

7. Authors claim that Supplementary Figure 3E shows dose dependent increase in pAKT, but total AKT levels seem to increase as well.

Response: We carefully repeated immunoblotting for pAKT and AKT. We confirmed the dose-dependent increase in pAKT levels without change in total AKT levels. The updated Supplementary Figure 3E is given below. Supplementary Fig. 3E in the original version is changed into Supplementary Fig. 3G in the revised version.

8. The statement “TMBIM6 is required for AKT activity and signaling” (Page 7) is overstated based on the data provided.

Response: This is a very reasonable concern. In this study, we demonstrated TMBIM6 mediates mTORC2 activation through its binding to ribosome, thereby inducing AKT activation as mTORC2 substrate. In the revised version, the statement was updated “TMBIM6 is one of the essential genes for mTORC2 signaling, which regulates AKT activity”.

9. PLA assays (Figure 3 and 5) require a negative control for an uninvolved cytoplasmic protein.

Response: In accordance with the reviewer’s suggestion, we performed PLA assays using ribosomal protein S6 kinase beta-1 (S6K1) as an uninvolved cytoplasmic protein. The updated Figure 3F and 5D, and their corresponding text are given below.

Figure 3F

Figure 3. TMBIM6 regulates mTORC2 activation. (A) Expression and phosphorylation of 43 proteins were examined by Proteome Profile Human Phospho-Kinase Array in WT and TMBIM6 KO HT1080 cells. Right; the relative phosphorylations of indicated proteins were quantitated by ImagJ. (B) pAKT, pTSC2, and pNDRG1 were analyzed in TMBIM6 KO and WT HT1080 cells using Western blots (left) and normalized to total proteins of WT cells (right; n = 5 independent experiments). (C) Immunoblot, RT-PCR and quantification of indicated genes in TMBIM6 KO cells stably expressing TMBIM6-HA (n = 3 independent experiments). (D) After serum starvation for 12 h, TMBIM6 KO and WT HT1080 cells were stimulated with insulin (100 ng/ml), IGF1 (100 ng/ml), or EGF (100 ng/ml), and immunoblotting was performed with indicated antibodies. (E) Gel filtration assay of extracts of TMBIM6 KO and WT MEFs. The red line represents the size marker. (F) PLA between indicated proteins (red dots) in TMBIM6 KO and WT HT1080 cells. For the PLA, **the ribosomal protein S6 kinase beta-1 (S6K1) was also applied as a negative control**. Scale bar, 15 μ m. Right, quantification of red dots (n = 5 independent experiments).

Figure 5D

10. The authors should describe how the BIA compound was identified.

Response: To search for novel small molecules, TMBIM6 antagonists, we first performed high-throughput screening (HTS) from materials of the Korea Chemical Bank, and elicited chalcone scaffold. Next, total of 44 substituents through modification of R1 and R2 position of chalcone scaffold were synthesized, and were tested for cell viability. BIA was developed as a tool compound, and confirmed dissociation between mTORC2 and RPL19, decreasing AKT phosphorylation, and inhibiting Ca^{2+} release. The updated supplementary Figure 9 and its corresponding text are given below.

Supplementary Fig. 9

Supplementary Fig. 9 Synthesis of BIA. (A) Schematic representation of chalcone structure. (B) Cell viability of HT1080 cells with 10 μ M of chalcone substituents (n = 3 independent experiments). (C) Schematic representation of the BIA synthesis. (D) NMR data of BIA.

Manuscripts

TMBIM6 antagonist reduces mTORC2 activity, inhibiting TMBIM6-associated tumorigenicity

Initially, to identify novel small molecule TMBIM6 antagonists, we performed high-throughput screening (HTS) from materials of the Korea Chemical Bank and elicited chalcone scaffold (Supplementary Fig. 9A). From the optimization of R1 and R2 position with diverse substituents, BIA was developed as a tool compound dependent cell viability (Supplementary Fig. 9B-D).

Reviewer #2 (Remarks to the Author) (Expertise: mTOR signalling, ribosomes) :

TMBIM6 is a calcium channel like protein and is upregulated in many cancer types. Suppression of TMBIM6 promotes cell death and decreases tumor growth. In the current study, the authors now elucidate a mechanism as to how TMBIM6 could promote tumor growth. By knocking out TMBIM6 in HT1080 cells via CRISPR, they found overall defects in metabolism and decreased mTORC2 signaling. By gel filtration, they found that rictor and GbL fractionated at lower molecular weights in the KO suggesting that mTORC2 dissociates

from a large complex in the absence of TMBIM6. They then conducted studies to determine association of mTORC2 with ribosomes by proximity ligation assay (PLA) and immunoprecipitation. They found that the mTORC2/ribosome (via expression of RPL19) association is disrupted in TMBIM6 KO. TMBIM6 also cofractionated and associated with mTORC2 components and RPL19. Knockdown of rictor specifically abolished interaction of TMBIM6 with mTOR and SIN1. Mutagenesis of TMBIM6 revealed that the N-terminus and the cytosolic loop residues are required for interaction with rictor and Akt activation. Since TMBIM6 is a calcium channel-like protein, they then analyzed if calcium release from TMBIM6 could affect the interaction of mTORC2 with ribosomes. By PLA, they found that there was decreased rictor/mTOR and rictor/rpl19 interaction in the TMBIM6 mutant (D213A) that affects calcium release. Although rictor association with TMBIM6 was similar in the WT and mutant TMBIM6, there was a slight decrease in mTOR and strong decrease in rpl19 and rpl16 association. By screening a chemical library, they identified B1A as a TMBIM6 antagonist that decreases tumor cell proliferation at rather high concentrations. Akt phosphorylation, total expression of mTORC2 components and rpl19 are also decreased by this drug as well as release of calcium from the ER. B1A also regressed tumor growth in vivo. Based on these results, the authors propose that TMBIM6 promotes cancer progression via association with mTORC2 and Akt activation.

Response: We thank the reviewer for his/her enthusiasm and insightful summary for this study.

Overall, the results are interesting and support a role for TMBIM6 inhibition in preventing tumor growth. The role of TMBIM6 in promoting mTORC2/ribosome assembly is also interesting but would need further clarifications as detailed below.

1. Is the association of TMBIM6 with mTOR unique to mTORC2? Authors should also blot for raptor.

Response: We appreciate the reviewers' comments. To identify whether TMBIM6 is associated with mTOR unique to mTORC2, not mTORC1, we performed immunoblot from immunoprecipitation lysates and PLA assay between TMBIM6 and RAPTOR, mTORC1 subunit, or between TMBIM6 and RICTOR, mTORC2 subunit. As shown in Figure 5C and 5D, TMBIM6 is associated with mTORC2, not mTORC1. The updated Figure 5C and 5D, and their corresponding text are given below.

Figure 5C

Figure 5D

Manuscripts

We also detected an association between TMBIM6 and endogenous mTORC2 or ribosomes (60S RPL19 and 40S RPS16) in TMBIM6-HA-overexpressing HeLa cells, **but not with raptor as mTORC1 subunit** (Fig. 5C). The localization of TMBIM6-HA in close proximity to mTORC2 components and ribosomes was observed by confocal microscopy (Fig. 5D).

2. In Figure 4, the authors found decreased expression of proteins involved in glycosylation. They should also include the analysis of metabolites of the hexosamine pathway. mTORC2 has been shown to control this metabolic pathway (authors should cite PMID: 27570073 by Moloughney et al Mol Cell on page 9).

Response: We appreciate the reviewer's comments. In accordance with the reviewer's suggestion, we performed an analysis of metabolites of hexosamine biosynthesis pathway (HBP) in TMBIM6 WT and KO HT1080 cells. In the revised manuscript, we have included new data in Figure 4D, showing that metabolites of HBP such as Uridine diphosphate N-Acetylglucosamine (UDP-GlcNAc), N-Acetyl-D-Glucosamine 6-Phosphate (GlcNAc-6-P), glucosamine-6-phosphate, and N-Acetyl-D-Glucosamine were decreased in TMBIM6 KO cells compared to TMBIM6 WT HT1080 cells. The updated Fig. 4D and its corresponding text are given below.

Figure 4

Manuscripts

mTORC2 activation by TMBIM6 regulates cellular metabolism

mTORC2 regulates cellular bioenergetics by modulating glycolytic gene expression, aerobic glycolysis, glutathione (GSH) biosynthesis, **hexosamine biosynthesis pathway (HBP)**, and glycosylation³⁶⁻³⁹. In this study, TMBIM6 KO cells showed downregulation of glycolytic genes (Fig. 4A), resulting in reduced glucose consumption and lactate production (Fig. 4B-C). The expression of genes related to the pentose phosphate pathway (PPP) was also decreased in TMBIM6 KO, which was reversed in TMBIM6-overexpressing HeLa cells (Fig. 4A and Supplementary Fig. 6). An MS analysis showed that the levels of metabolites from

glycolysis, tricarboxylic acid cycle, PPP, and HBP were decreased in TMBIM6 KO cells relative to those in WT cells (Fig. 4D) indicating that metabolic pathways are dysregulated in the absence of TMBIM6, where is linked to the inhibition of mTORC2 activity.

3. Akt phosphorylation should also be analyzed in Fig 5I.

Response: In accordance with the reviewer's comments, we performed immunoblotting in TMBIM6 WT and C-terminal 40 amino acids-deleted HT1080 cells. The phosphorylation of AKT Ser473 was reduced in the mutant cells. The updated Fig 5I is given below.

Manuscripts

Under the expression of TMBIM6 with deletion of 40 C-terminal amino acids, the association with RPL19 was abrogated, whereas the interaction with RICTOR or mTOR was not altered (Fig. 5I). The phosphorylation of AKT Ser473 was also decreased in RPL19-non-associated TMBIM6 mutants (Fig. 5H, I).

4. While the PLA and immunoprecipitation assays to show association of mTORC2 and rp119, the authors should more carefully analyze how TMBIM6 promotes association of mTORC2 with the ribosomes and whether this association is occurring in translating ribosomes. Polysome purification and analysis of mTORC2 association or cofractionation with TMBIM6 should be conducted.

Response: In accordance with reviewer comments, we performed polysome, and ribosome fractionation in a sucrose gradient and purification of mRNA bound ribosomes by pull-down of poly(A) mRNA with oligo(dT) cellulose. As shown in Supplementary Fig. 4, there is no difference in polysome maturation between TMBIM6 KO and WT cells. However, the expressions of mTOR, RICTOR, and SIN1; mTORC2 components were lower in polysome and ribosome fractionation samples from TMBIM6 KO cells than WT cells, indicating that the physical association between mTORC2 and ribosomes is regulated by TMBIM6. Moreover, TMBIM6 was co-purified along with mTOR and RICTOR in polysome and ribosome fractionation samples, suggesting that TMBIM6 promotes the association of mTORC2 with the ribosomes.

Supplementary Fig. 4

Supplementary Fig. 4 TMBIM6 regulates the association of mTORC2 and ribosome. (A) Polysome profiling was performed in TMBIM6 WT and KO HT1080 cells by sucrose gradient fractionation. The polysomal (P) and ribosomal (M) fractions are indicated. (B) Immunoblot analysis with the indicated antibodies of fractions from A. (C) Immunoblot analysis with the indicated antibodies of fractions from empty-vector and TMBIM6 rescued KO HT1080 cells. (D) Immunoblot analysis with the indicated antibodies was performed in the purified poly(A) mRNA-bound ribosomes from HT1080 cell with stably expressing TMBIM6 by oligo(dT) pull-down. The bound fraction (B) and the supernatant (S) are indicated.

Manuscripts

To identify whether reducing mTORC2 activity in TMBIM6 KO cells is related to impairment of ribosome maturation, we performed fractionation in a sucrose gradient assay to separate polysomes from 80S, 60S, and 40S ribosomes. As shown in Supplementary Fig. 4A, the pattern of ribosome profiling was same between TMBIM6 WT and KO cells, indicating TMBIM6 is not related with ribosome maturation. In addition, mTOR, RICTOR, and SIN1 were found in both the polysomal and ribosomal fractions in TMBIM6 WT HT1080 cells (Supplementary Fig. 4B). However, mTORC2 components were relatively less detected in both fractions from TMBIM6 KO HT1080 cells compared to those from WT cells. In TMBIM6-rescued cells, TMBIM6 was co-purified with polysome and ribosome fractions (Supplementary Fig. 4C). Since mTORC2 physically interacts with translating (mRNA-bound) and non-translating 80S ribosomes, and TMBIM6 binds to the mTORC2, we next determined whether TMBIM6 is copurified with mTORC2 at mRNA-bound ribosomes. In mRNA bound ribosomes purified by pull-down of poly(A) mRNA with oligo(dT) cellulose, TMBIM6 was copurified with mTOR, RICTOR, and RPL19 (Supplementary Fig. 4D). Collectively, these results suggest that TMBIM6 regulates the assembly of mTORC2 components and promotes the physical association between mTORC2 and ribosomes.

Methods

Polysome profiling and poly(A) pull-down assay

Polysome profiles were performed as described previously with minor adjustments. In brief, cells were incubated with a final concentration of 100 µg/ml of cycloheximide for 10 min before harvest. The cells were then washed with 100 µg/ml of cycloheximide in PBS, collected in tubes, and resuspended in 1 ml of polysome lysis buffer (20 mM Tris-HCl pH 7.5, 100 mM NaCl, 10 mM MgCl₂, 0.4% IGEPAL, and 100 µg/ml cycloheximide) with 10 unit/ml RiboLock RNase Inhibitor (EO0381, Thermo scientific) and Xpert protease inhibitor cocktail (P3100, genDEPOT, Katy, TX, USA). The clarified lysates were loaded onto 10 ml linear 10-50% (w/v) sucrose gradients (prepared in 20 mM Tris-HCl pH 7.5, 100 mM NaCl, 10 mM MgCl₂, 100 µg/ml cycloheximide, 1X protease inhibitor cocktail, and 10 units/ml RNase inhibitor) and separated by centrifugation at 36,000 rpm for 2 hr at 4 °C in a P40ST swing rotor (Hitachi, JAPAN). The gradients were then fractionated with a Fluorinert FC-40 (F9755, Sigma Aldrich) and 750 µl of the fractions were collected in tubes using an ISCO density gradient fractionation system. For poly(A) pull-down assay, cells were lysed in buffer A (50 mM Tris-HCl [pH 7.4], 100 mM NaCl, 30 mM MgCl₂, 0.3% CHAPS, 40 U/ml RNase inhibitor, protease inhibitor cocktail, and 100 µg/ml cycloheximide) as previously report²⁹. Lysates were clarified at 4 °C, 10 min at 8000 x g and then were incubated with oligo (dT) cellulose (NEB) for 1 hr at room temperature. The oligo (dT) cellulose was pelleted by centrifugation, and washed five times with buffer A. The bound fraction was eluted with elution buffer (100 mM Tris [pH 7.4], 500 mM NaCl, 10 mM EDTA, 1% sodium dodecyl sulfate (SDS), and 5 mM DTT). Purified ribosome fractions and the bound and unbound fractions were concentrated with Vivaspin 500 (Sartorius Stedim).

5. The IC₅₀ of B1A should be assessed. In Fig 7, it seems that high concentrations are needed to prevent cell proliferation of all the cell lines examined.

Response: In accordance with the reviewer's comments, we calculated IC₅₀ of BIA using GraphPad 8.0 software. In Fig 7A, the IC₅₀ values at 3 days were 1.7 ± 0.1 µM for HT1080, 2.6 ± 0.4 µM for MCF cells, 2.6 ± 0.5 µM for MDA-MB-231 cells, and 2.4 ± 0.4 µM for SKBR3 cells.

Manuscripts

Next, the proliferation of all three cell lines containing HT1080, MCF7, and MDA-MB-231 cells was inhibited by treatment with 5 µM BIA (Fig. 7A). The IC₅₀ values at 3 days were 1.7 ± 0.1 µM for HT1080, 2.6 ± 0.4 µM for MCF cells, 2.6 ± 0.5 µM for MDA-MB-231 cells, and 2.4 ± 0.4 µM for SKBR3 cells. We also checked cell viability at three days, and confirmed inhibition by BIA at all cell lines (Supplementary Fig. 10B). We also checked cell viability at three days, and confirmed inhibition by BIA at all cell lines (Supplementary Fig. 10B). Moreover, HT1080 cells stably overexpressing TMBIM6 showed high sensitivity to BIA (Supplementary Fig. 10C).

6. B1A (Fig 7B) seems to decrease total protein expression of mTOR and rpl19, as well as TMBIM6. There seems to be no strong effect on disruption of the complexes, other than slight decrease in TMBIM6. The authors should quantitate this figure more carefully to reflect comparative amount of total proteins.

Response: We appreciate the reviewer's comments. The total expressions of TMBIM6, mTOR, RICTOR, and RPL19 were not changed under the BIA, which was confirmed by immunoblot assay from size-fractionation samples. It is indicated that the BIA-induced effect

is not related to the mTOR and rpl19 protein expression but more related to the regulatory effect on the complex formation between mTOR/RICTOR and RPL19. The updated Fig 7B is given below.

7. Expression of total mTOR and rictor proteins should be included in Fig 7C.

Response: We appreciate the reviewer comments. The updated Fig 7C is given below.

8. The authors state in the Discussion that “This effect of TMBIM6 on cell growth differs from its classical role in ER stress.” This has really not been addressed in the current study. Previous studies have already shown a role for mTORC2 in negative regulation of the ER calcium channel regulator Mid1 (in yeast) as a negative regulator of starvation response (PMID 27899413 Vlahakis et al JCB 2016). They should also cite this work to support the relationship of calcium signaling to mTORC2. Furthermore, they should verify that the disruption of TMBIM6 does not cause ER stress. The overall decrease in metabolic pathways and possible decrease in total ribosomal proteins (see comment 6) suggest that translation could also be decreased (which would relieve ER stress at least partially). BI-1 Ko condition is not related with ER stress, document it. They concern the possibility “The KO of BI-1 reduces ER stress, a possibility

Response: We appreciate the reviewer’s thoughtful comments, and respond to each comment individually.

About the issue “the correlation between mTORC2 and Ca^{2+} ”, our data support that leaky calcium from TMBIM6 affects directly mTORC2 activation through inducing the physical association of RICTOR with core proteins of the ribosome, RPL16, and RPS proteins. In a

previous study about a role for mTORC2 in negative regulation of the ER calcium channel regulator Mid1 (in yeast) (Vlahakis et al. JCB 2016), the impairment of mTORC2-Ypk1 signaling activates calcineurin via Mid1, and the resultant autophagy flux disturbance by inhibition of GAAC response under conditions of amino acid starvation. However, we also reported that TMBIM6 rather enhances autophagy flux through Ca²⁺ leaky characteristics (Kim et al., Autophagy 2020). Due to the TMBIM6 Ca²⁺ leaky channel characteristics, mTORC2 binding affinity, and the recruitment of mTORC2 to the local Ca²⁺ enriched area is the main thing to explain the correlation between mTORC2 and Ca²⁺.

In relation ER stress, the absence of TMBIM6 increases ER folding impairment and ER stress, enhancing UPR signaling (Chae et al., 2004; Lee et al., 2011). Most of these reports have examined the characteristics of TMBIM6 only in stress condition “in the presence of ER stress inducer”. However, the presence or absence of TMBIM6 does not affect ER stress response under non-stressed conditions or in resting conditions (Chae et al., 2004; Lee et al., 2011). Therefore, TMBIM6 is not a simple ER stress regulator but rather an AKT activator enhancing cell proliferation, especially in cancer conditions.

In the revised discussion, we have updated as following.

Manuscripts

mTORC2 was found to be associated with the ER through its direct binding to TMBIM6. We determined that TMBIM6 serves as a signaling scaffold that recruits mTORC2 to the ER and thus promotes cell survival. This effect of TMBIM6 on cell growth differs from its classical role in ER stress. TMBIM6 regulates ER stress-induced cell death^{1,50}. The overall decrease of metabolic pathways and protein synthesis in the absence of TMBIM6 should have relieved ER stress at least partially. However, the absence of TMBIM6 increases ER stress, enhancing UPR signaling^{1,50}. Most of these reports have examined the characteristics of TMBIM6 only in stress conditions “in the presence of ER stress inducer”. However, the presence or absence of TMBIM6 does not affect ER stress response under non-stressed conditions or in resting conditions^{1,50}. At least in a resting condition, TMBIM6 is not a simple ER stress regulator but rather a core protein enhancing protein recruitment and assembly, including mTORC2, ultimately affecting AKT activation and cell proliferation, especially in cancer condition.

Similar to mTORC1 studies⁵²⁻⁵⁴, we found that mTORC2 activity is sensitive to inhibition by BAPTA-AM (Figure 6A-B, supplementary Fig. 7A), suggesting that intracellular calcium is required for mTORC2 activation. In a recent study about the correlation of Ca²⁺ channel and mTORC2, mTORC2 negatively regulates Mid1, an ER/plasma membrane (PM)-localized calcium channel regulatory protein. Decreased signaling of TORC2 and its downstream target protein kinase, Ypk1, induced Mid1 activation⁵⁵. The mTORC2 signaling regulates the Ca²⁺-associated Mid as downstream effector under amino acid starvation, whereas the Ca²⁺-leaky TMBIM6 first affects mTORC2 activation through the releasable Ca²⁺ from TMBIM6 under basal condition. Our point is not general cytosolic Ca²⁺ but the local Ca²⁺ from the Ca²⁺ leaky protein, TMBIM6, which is also abrogated in the presence of BAPTA-AM, not BAPTA, a non- cell permeable Ca²⁺ chelating agent. The precise mechanism of regulation of mTORC2 by calcium through TMBIM6 Ca²⁺ leaky characteristics is distinct in our model. First, we demonstrated that the ER leaky calcium plays a unique and critical role in mTORC2 activation and the resultant AKT in mammalian

cells. Second, D213A mutant had no effect on the interaction between TMBIM6 and RICTOR, ruling out the involvement of the local Ca^{2+} in the direct binding with the mTORC2 subunit protein. However, we also found that D213A mutant had a strong controlling effect on the interaction between mTOR and RICTOR or between RICTOR and the ribosomal proteins, RPL19 and RPS16 suggesting the involvement of the local Ca^{2+} in the assembly of mTORC2 and further the recruitment of ribosomal proteins also.

Reviewer #3 (Remarks to the Author) (Expertise: stress, ER) :

NCOMMS-19-31425-T

Comments to the authors,

In this work, Kim and colleagues show that TMBIM6 –a member of the highly conserved TMBIM family of proteins –interacts with components of the mTORC2 complex to regulate cancer progression. Indeed, they first show that TMBIM6 is overexpressed in several different types of carcinomas and its expression correlates with poorer patient overall survival. Using loss-of-function approaches, the authors show that TMBIM6 promotes cell migration and invasion in vitro and tumor growth in vivo. These experiments are well performed and highlight the known role of TMBIM6 as a protumorigenic protein. In fact, work by this same group has previously shown that TMBIM6 overexpression promotes metastasis through the regulation of cell migration and invasion (PMID: 20118983). At the molecular level they show that TMBIM6 is required for the assembly of the mTORC2 complex, leading to improved glucose homeostasis and promoting cancer progression. In short, TMBIM interacts and tether the mTORC2 complex and the ribosomes to the ER where it would promote its assembly through TMBIM6 calcium leak channel function. Finally, the authors described a putative TMBIM6 antagonist with anti-tumor properties.

Overall, the in vitro and in vivo results are very interesting and well-presented. However, I have some concerns regarding the proposed role of TMBIM6 as an mTORC2 regulator and its connection to cancer progression. Some key control experiments are currently missing, making the interpretation of the data difficult. The statistical analysis used throughout the manuscript should also be revised.

A revised version of this manuscript should still be suitable for publication in Nature Communications.

Response: We appreciate that the reviewer found the significance of this study, and thank for his/her insightful comments and criticisms

Major concerns:

1. Model of choice and generality: In Figure 1, the authors showed that TMBIM6 is overexpressed in several different cancer types, including cervical, endometrial and vulvar, breast, lung and prostate cancer. They also show that high TMBIM6 expression correlates with poorer prognosis in patients with squamous cell carcinoma and endocervical adenocarcinoma, esophageal carcinoma, skin cutaneous melanoma, head and neck squamous

carcinoma and brain lower grade glioma, most of them of epithelial origins (Figure 1). However, they perform most of the loss-of-function experiments (migration, invasion, and tumor growth and mTORC2 biochemical experiments) in HT1080 cell lines, which is a fibrosarcoma cell line of mesenchymal origin. Two comments on this: (1) the authors should show, is it were, TMBIM6 expression and survival data on fibrosarcoma, since it will correlate better with the observations performed in HT1080 TMBIM6 KO cells and (2) the authors should repeat some key experiments (mTORC2 assembly, AKT phosphorylation and tumor growth) in an additional TMBIM KO cell line cancer model directly related to the cancers shown in figure 1. Most of the experiments connecting TMBIM6 with mTORC2 were performed in only one cancer cell line.

Response: We appreciate the reviewer's comments, and responded to each comment individually.

(1) the authors should show, is it were, TMBIM6 expression and survival data on fibrosarcoma, since it will correlate better with the observations performed in HT1080 TMBIM6 KO cells

Response: As per the reviewer's first comments, we searched TMBIM6 expression data in NCBI Gene Expression Omnibus database (NCBI/GEO), and confirmed that TMBIM6 expression was increased in sarcoma samples (GSE2719). Since fibrosarcoma is part of a wider family of sarcomas, we used sarcoma for survival analysis. Survival analysis of sarcomas (SARC) was represented in Fig. 1G, showing the correlation of the high expression of TMBIM6 with poor prognosis.

Figure 1

(2) the authors should repeat some key experiments (mTORC2 assembly, AKT phosphorylation and tumor growth) in an additional TMBIM KO cell line cancer model directly related to the cancers shown in figure 1.

Response: To solve the second comments, we generated scrambled and two TMBIM6 KO HeLa cell lines directly related to cervical cancer shown in Figure 1 using CRISPR/Cas9 systems. Next, we performed several experiments, including cell proliferation, PLA for mTORC2 assembly, immunoblotting for detection of AKT phosphorylation, and *in vivo* xenograft for tumor growth. TMBIM6 KO HeLa cells showed that cell proliferation, AKT phosphorylation, and tumor growth were decreased. The updated data and its corresponding text are given below.

Additional Supplementary Item 1. Generation of TMBIM6 KO cells by CRISPR/Cas9 genome editing. (A) Schematic illustration of genome editing via CRISPR/Cas9-mediated RNA-guided site-specific DNA cleavage. The sequence of the mutated allele in TMBIM6 harboring the insertions/deletions is shown. (B-C) *TMBIM6* mRNA levels in WT and TMBIM6-KO HT1080 (B) and HeLa cells (C), as determined by qRT-PCR.

Figure 2

Figure 2. TMBIM6 enhances tumor growth. (A) Proliferation of TMBIM6 KO and WT HT1080 cells, HeLa cells, and MEFs (n = 3 independent experiments). (B) Images of migrated cells in TMBIM6 KO and WT HT1080. Right, quantification of WT cells

normalized to KO cells (n = 6 independent experiments). Scale bars, 100 μ m. (C) Images of invasive cells in TMBIM6 KO and WT HT1080. Right, quantification of WT cells normalized to KO cells (n = 5 independent experiments). Scale bars, 100 μ m. (D-F) Tumor volume, weight, and size derived from TMBIM6 KO or WT HT1080 cells injected into the flank of 6-week-old nude mice (n = 6 mice per group). (G) Histological images by immunohistochemical detection of Ki-67. Right, quantification of Ki-67-positive cells in xenograft tumors derived from TMBIM6 KO and WT HT1080 cells (n = 6 mice per group). Scale bars, 100 μ m. (H-J) Tumor volume, weight, and size derived from TMBIM6 KO or WT HeLa cells injected into the flank of 6-week-old nude mice (n = 6 mice per group). (K) Histological images by immunohistochemical detection of Ki-67. Right, quantification of Ki-67-positive cells in xenograft tumors derived from TMBIM6 KO and WT HeLa cells (n = 6 mice per group). Scale bars, 100 μ m. Data represent mean \pm SD. Statistical differences were detected with two-tailed unpaired Student's *t*-tests (B, C, E, G), one-Way ANOVA followed by Tukey's test (I, K), and two-Way ANOVA followed by Bonferroni's test (A, D, H)

Supplementary Fig. 3 TMBIM6 regulates mTORC2 activity. (A-C) Immunoblot analysis for the indicated proteins in HT1080, HeLa, and MEFs cells. (D) Immunofluorescence images of pAKT. Right, mean pAKT intensity in the KO cells normalized to WT cells (n = 10 independent experiments). (E) Immunoblot analysis for the indicated proteins in HeLa cells transfected with HA-TMBIM6. (F) After serum starvation for 12 h, TMBIM6 KO MEF cells with transfection of TMBIM6-HA were stimulated with or without insulin (100 ng/ml), and immunoblotting was performed using the indicated antibodies. (G) Immunoblot of the indicated proteins in TMBIM6 T-Rex 293 cells treated with various concentrations of doxycycline for 24 h. (H) PLA between indicated proteins (red dots) in TMBIM6 KO and WT HeLa cells. The ribosomal protein S6 kinase beta-1 (S6K1) as negative controls for PLA

was used. Scale bar, 15 μm . Right, quantification of red dots ($n = 5$ independent experiments). (I) mRNA levels of indicated genes in TMBIM6 KO and WT HT1080 cells, as determined by qRT-PCR ($n = 3$ independent experiments). (J) Immunoblot analysis of anti-RICTOR IP and WCL of TMBIM6 KO and WT MEFs. WCL, whole cell lysates. Data represent mean \pm SD. Statistical differences were detected with two-tailed unpaired Student's *t*-tests (D), and two-Way ANOVA followed by Bonferroni's test (H, I).

Manuscripts

TMBIM6 depletion suppresses the tumorigenicity of cancer cells

To validate the above results, we performed cell proliferation, migration, and invasion assay. TMBIM6 KO HT1080, HeLa cells, and MEFs both exhibited slow growth relative to WT cells (Fig. 2A), which was restored in TMBIM6 KO cells with re-expressing TMBIM6 (Supplementary Fig. 2A-B). Cell migration and invasion—as indices of cancer progression—were inhibited in cells lacking TMBIM6 (Fig. 2B-C, Supplementary Fig. 2C-D). To investigate the role of TMBIM6 in the growth of tumor cells in animals, we subcutaneously injected TMBIM6 WT and KO HT1080 cells into the left and right flanks, to immune-compromised mice and monitored tumor growth during 27 days (Supplementary Fig. 2E). As shown in Fig. 2D, tumor formation in TMBIM6 KO was significantly reduced compared with that in WT cells over the same period. The weight of tumors originating from TMBIM6 KO HT1080 cells was decreased more than 6-folds than that of those originating from WT cells at the end of the experiment (Fig. 2E and F). Immunohistochemistry analysis of Ki-67 expression showed a significant decrease in tumors derived from TMBIM6 KO cells compared with that in those from WT cells (Fig. 2G). Consistently, we also found that tumor formation and weight, and the expressions of Ki-67 was apparently reduced in TMBIM6 KO HeLa cells compared with that in WT cells (Fig. 2H-K, Supplementary Fig. 2F). In addition, TMBIM6 knockdown by injection of self-assembled micelle inhibitory RNA (SAMiRNA), a stable siRNA silencing platform for efficient *in vivo* targeting of genes, reduced tumor formation as well as Ki-67 expression when compared with those in the control groups (Supplementary Fig. 2G-L). Taken together, these *in vitro* and *in vivo* experiments demonstrate that TMBIM6 regulates tumor growth.

2. Is the TMBIM6 KO HT1080 cell line a pool or was derived from a single clone? In the case of it being a clonal population, the authors should repeat some key experiments (i.e. AKT phosphorylation and the assembly of the mTORC2 complex) with additional clones. Throughout the manuscript, the authors compare the HT1080 TMBIM6 KO cells with wild-type controls when they should have been compared with a matched CRISPR/Cas9 scramble or mock control. I am worried about CRISPR/Cas9 off-target effects.

Response: We generated two TMBIM6 KO and scrambled HT1080 cell lines as a single clone. Since the number of gene editing sequences at genome is different on allele by CRISPR/Cas9 system, we isolated individual clones by the limiting dilution method after transfection. Therefore, as reviewer's comments, we performed cell proliferation, cell migration, cell invasion, immunoblotting for AKT phosphorylation, PLA for the assembly of the mTORC2 complex, glucose consumption, and lactate production assays in additional clones (clone #2). All results obtained from TMBIM6 KO#2 clones showed similar results as those from TMBIM6 KO#1 clones. Moreover, AKT phosphorylation from scramble clone designated similar results those from WT cells. The updated data and corresponding text are given below.

Additional Supplementary Item 1

A

B

HT1080

C

HeLa

Supplementary Fig. 3

Method Section

Generation of *TMBIM6* KO cells by CRISPR/Cas9 genome editing

The CRISPR/Cas9 genome editing method was used to generate the *TMBIM6* KO HT1080 cell line. The plasmid containing sequences targeting human *TMBIM6* were designed and constructed from the pRGEN_ *TMBIM6* expression vector by ToolGen (Seoul, Korea). The guide sequence targeting exon 3 of human *TMBIM6* was 5'-TGCAGGGGCCTATGTCCATATGG-3'. pRGEN_Scramble vector as a negative control was constructed using scrambled sequence (5'-GCACTACCAGAGGCTAACTCA-3'), which informed from Origene (#GE100003, pCas-Scramble Vector). The pRGEN_ *TMBIM6* or pRGEN_Scramble vector was mixed with pRGEN_Cas9-CMV and co-transfected into HT1080 and HeLa cells using Lipofectamine 3000

3. From the data shown in Figures 3F (PLA assay), 3G and supplementary figure 3F (IP assays) the authors suggest that “*TMBIM6* regulates the assembly of mTORC2 components and promotes the physical association between mTORC2 and the ribosomes”. However, the authors did not rule out the possibility that *TMBIM6* expression may regulate the concentration of proteins forming the mTORC2 complex (the authors also showed that *TMBIM6* KO reduces global protein synthesis). The same seems to be the case with mTOR and RICTOR co-localization with PDI (Supplementary Figure 4): *TMBIM6* KO cells seem to exhibit decreased levels of these proteins, hence, decreased co-localization. At the very least, the authors should compare the endogenous protein levels of mTOR, RICTOR, RPL19 and RPS16 between HT1080 WT and *TMBIM6* KO cell lines. If willing, they should also check the mRNA levels of these genes. Is it reduced assembly, reduced protein expression or both?

Response: We confirmed the endogenous protein levels and mRNA levels of mTOR, RICTOR, SIN1, GβL, RPL19, and RPS16 in HT1080 WT and *TMBIM6* KO cell lines. As shown in Figure 3G and Supplementary Fig 3I, the protein and mRNA expression levels of these genes were same in *TMBIM6* WT and KO cells, opening the possibility that *TMBIM6* affects physical interaction between ribosome and mTORC2.

Supplementary Fig. 3I

Supplementary Fig. 3 *TMBIM6* regulates mTORC2 activity. (I) mRNA levels of indicated genes in *TMBIM6* KO and WT HT1080 cells, as determined by qRT-PCR (n = 3 independent experiments).

Manuscripts

In an immunoprecipitation assay with RPL19, the binding of mTOR, RICTOR, SIN1, and GβL [also known as mammalian lethal with SEC13 protein (mL)8] to RPL19 was mostly abrogated in KO as compared to that in WT cells (Fig. 3G). Moreover, the expression levels of protein and mRNA of these genes were same in *TMBIM6* WT and KO cells (Fig. 3G, Supplementary Fig. 3I).

4. To assess the putative role of calcium released from TMBIM6 on mTORC2 assembly, the authors use a TMBIM6-GCaMP3 construct that they say "... is based on the finding that leaky calcium but not ER lumen [calcium] is detected upon binding of Ca^{2+} to the cytosol of this protein (Supplementary Fig. 6B)". To what side of TMBIM6 was GCaMP3 attached? The latest structural models based on the crystallization of the bacterial homolog BsYetJ suggest that TMBIM proteins are composed by seven transmembrane domains with the N-terminus facing the cytosol and the C-terminus facing the intraluminal space (see for example: PMID:24904158 and PMID:30930064). The authors should comment and cite these structural works and incorporate them into their models presented in Supplementary Figure 6C, where they only show the 6 transmembrane models for TMBIM6. How are the authors completely positive that they are not measuring ER calcium content?

Response: TMBIM6 topology analysis showed that the C-terminus of TMBIM6 resides in the cytosol in previous reports (Bultynck et al., 2012). However, in structural models based on the crystallization of the bacterial homolog BsYetJ, C-terminus of TMBIM6 resides was located in the ER intraluminal space (Chang et al., 2014). To clarify the debate issue of TMBIM6 topology, we compared the models TMBIM6 structure obtained by TMpred, TMMHMM, and BsYetJ. As shown in Additional supplementary item 2A, TMBIM6 has six or seven transmembrane domains based on the model source. Although BsYetJ is a bacterial protein related to hTMBIM6, it has only 23.77% amino acid identity by Blastp, a limitation for the application of human TMBIM6 topology. In addition, we performed immunofluorescence using cells overexpressing TMBIM6 tagged with the N-terminal (HA-TMBIM6) and C-terminal (TMBIM6-HA) HA tag, and selective membrane permeabilization reagents, by modifying previous studies (Bultynck et al., 2012). Digitonin allows epitopes in the cytosol, but not in the ER lumen, accessible to antibodies. In contrast, Triton X-100 permeabilizes all membranes, leading to staining of both luminal and cytosolic epitopes. The protein disulfide isomerase (PDI) retained in the ER lumen was used as a negative control. Under digitonin treatment, both HA-TMBIM6 and TMBIM6-HA represented similar fluorescence intensity, whereas PDI did not show fluorescence intensity. On the other hand, HA-TMBIM6, TMBIM6-HA, and PDI staining all increased fluorescence in Triton X-100 treated condition for cell membrane permeabilization. These results suggest that the N- and C-terminal of TMBIM6 is all oriented cytosol dependent on the six-transmembrane structure, although we cannot exclude the possibility that the fusion with HA itself might alter the topology of TMBIM6. However, further study is necessary for accurate structural analysis using human TMBIM6 protein. The updated data and corresponding text are given below.

Additional Supplementary Item 2

Additional Supplementary Item 2. Representative TMBIM6 topology. (A) Bioinformatical prediction of the topology of TMBIM6 according to TMpred, TMHMM, and BsYetJ. The box and number indicate transmembrane domain and amino acids, respectively. (B) Alignment of amino acid sequences between TMBIM6 and BsYetJ based on previous reports. The box and line represent same and alternative prediction sequences from (A), respectively. (C) Immunofluorescence using cells overexpressing TMBIM6 tagged with the N-terminal (HA-TMBIM6) and C-terminal (TMBIM6-HA) HA tag after permeabilization by digitonin or triton X-100.

Manuscripts

TMBIM6 is composed of six or seven transmembrane regions with mostly α -helical structures, which C-terminus of TMBIM6 resides in the cytosol by TMHMM or in ER intraluminal space by the bacterial homolog BsYetJ⁴⁰⁻⁴⁴ (Additional Supplementary Item 2A). Although BsYetJ is a bacterial protein related to hTMBIM6, it has only 23.77% amino acid identity by Blastp (Additional Supplementary Item 2B). To further understand TMBIM6 topology, we performed immunofluorescence using cells overexpressing TMBIM6 tagged with the N-terminal (HA-TMBIM6) and C-terminal (TMBIM6-HA) HA tag, and selective membrane permeabilization reagents, by modified previous reports⁴². Digitonin makes epitopes in the cytosol, but not in the ER lumen, accessible to antibodies. In contrast, treatment with Triton X-100, permeabilizes all membranes and leads to staining of both luminal and cytosolic epitopes. The protein disulfide isomerase (PDI) retained in the ER lumen was used as a negative control. In the presence of digitonin, both HA-TMBIM6 and TMBIM6-HA represented similar fluorescence intensity, whereas PDI fluorescence was not detected (Additional Supplementary Item 2C). On the other hand, HA-TMBIM6, TMBIM6-HA, and PDI fluorescence was increased in the Triton X-100-induced cell membrane permeabilization conditions. These results suggest that N-terminal and C-terminal of

TMBIM6 is cytosolic exposed in six-transmembrane structure condition, although we cannot exclude a possibility that topology of TMBIM6 might be altered by the fusion with HA itself as previously mentioned⁴².

5. In Figure 6, the authors reconstituted TMBIM6 KO cells with either TMBIM6-HA or the channel mutant TMBIM D213A and assessed calcium leak (Figure 6A), mTORC2 assembly (Figure 6B) and AKT phosphorylation (Figures 6D and 6E). However, to correctly interpret these experiments, a comparison of the total reconstituted levels of TMBIM6 and TMBIM6 D213A proteins in KO cells is necessary. The authors should also check the ER localization of the WT and the D213A mutant proteins by immunofluorescence. Differential partial reconstitution of these proteins may account for the differences in mTORC2 assembly and AKT phosphorylation.

Response: In our recent report, we documented the ER localization of TMBIM6 WT and TMBIM D213A (Kim et al., 2020). Moreover, the mutant with C-terminal nine amino acids of the protein was replaced by alanines (BI-1^{C9A}) was also localized at ER, which was consistent with a previous report (Lisbona et al., 2009). Consider that TMBIM6 has six or seven transmembrane domains, and the D213A mutant would not affect TMBIM6 topology. However, in accordance to reviewer’s suggestions, we performed immunofluorescence in TMBIM6-HA WT or TMBIM6-HA D213A-transiently overexpressed TMBIM6 KO cells. As shown in supplementary Fig. 7C, both of TMBIM6 WT and D213A expression were observed in the ER. Thus, the differences about mTORC2 assembly (Figure 6B) and AKT phosphorylation between the TMBIM6 and the mutant D213A (Figures 6D and 6E) are suggested to be related with the characteristics of TMBIM6, “Ca²⁺ leakage”, not differential reconstitution. The updated data and its corresponding text are given below.

Supplementary Fig. 7

Supplementary Fig. 7 Ca^{2+} regulates mTORC2 activation. (A) PLA between the indicated proteins (red dots) in HT1080 cells treated with BAPTA-AM (10 μM), BAPTA (10 μM), and EGTA-AM (10 μM). Scale bar, 15 μm . Right, quantification of red dots (n = 3 independent experiments). (B) Illustration of TMBIM6-GCaMP3 by a genetically-encoded Ca^{2+} indicator (GCaMP3) fused directly to the C-terminus of TMBIM6 (TMBIM6-GCaMP3). (C) TMBIM6 and D213A expression-rescued KO cells were stained for calnexin (CANX, ER marker). (D) The scheme of TMBIM6 characteristics; TMBIM6-leaky Ca^{2+} and the interaction with mTORC2 and ribosome complex. Data represent mean \pm SD. Statistical difference was detected with one-Way ANOVA followed by Tukey's test (A).

Manuscripts

To identify whether differential retention of TMBIM6 WT or D213A mutant in ER affects mTORC2 assembly and AKT phosphorylation, we performed immunofluorescence in HT1080 cells transiently transfected with WT TMBIM6-HA and D213A mutant. As shown Supplementary Fig. 7C, TMBIM6 WT and D213A were retained in ER at a comparable level. Thus, TMBIM6-associated AKT activation is based upon the following characteristics of protein interactions: the binding of TMBIM6 with RICTOR was independent of Ca^{2+} leakage, whereas the interaction of TMBIM6 with mTOR or ribosomal subunits including RPL19 was dependent on local Ca^{2+} leakage. The Ca^{2+} -relevant TMBIM6 state was schematically described (Supplementary Fig. 7D).

6. Specificity of BI: from the data presented in Figures 7 and 8, and Supplementary Figure 7, the authors suggest that BIA acts as a TMBIM6 antagonist, blocking ER calcium release, leading to mTORC2 disassembly and inhibition of TMBIM6-associated tumorigenicity. Although it is clear that BIA reduces cell proliferation, migration and invasion, it is not so clear that these effects dependent on BIA's function as a TMBIM antagonist. Does BIA work when TMBIM6 is knocked out? There are also several key control experiments missing from figures 7 and 8 that preclude the interpretation of these experiments. For example, total mTOR and RICTOR levels are missing from the IP shown in Figure 7C. The authors should also report the effect of BIA on cell viability under the conditions reported in Figure 7A. The data supporting the role of BIA as an inhibitor of the interaction between TMBIM6 and mTORC2 is currently not convincing enough. The authors could perform PLA as in previous experiments to strengthen this data.

Response: We appreciate the reviewer's comments, and responded to each comments individually.

- Does BIA work when TMBIM6 is knocked out?

As per the reviewer 1's comments, we performed cell proliferation, migration assay, and immunoblotting about AKT signaling in BIA-treated TMBIM6 KO cell lines. As shown in Fig. 7A, and Supplementary Fig. 10E-G, the proliferation, migration, and the status of AKT phosphorylation of TMBIM6 KO HT1080 cells were not significantly affected even in the presence of BIA.

- Total mTOR and RICTOR levels are missing from the IP shown in Figure 7C.

In the revised version, we added image of blot for total mTOR and RICTOR, and updated them in the Figure 7C.

- The authors should also report the effect of BIA on cell viability under the conditions reported in Figure 7A.

In the revised version, we have added cell viability data in supplementary Fig. 10B under the same condition as reported in the original Figure 7A.

- The data supporting the role of BIA as an inhibitor of the interaction between TMBIM6 and mTORC2 is currently not convincing enough. The authors could perform PLA as in previous experiments to strengthen this data.

As per the reviewer's suggestions, we performed PLA assay to elucidate the BIA effect "inhibition of the interaction between TMBIM6 and mTORC2 or between TMBIM6 and ribosome". As shown in Figure 7E and Supplementary Fig. 10D, the interaction of TMBIM6 with mTORC2 components was decreased in the BIA-treated cells. The updated data and corresponding text are given below.

Figure 7E

Figure 7. BIA, a TMBIM6 antagonist, suppresses tumor growth. (A) Proliferation of HT1080, TMBIM6 KO HT1080, MCF7, MDA-MB-231, and SKBR3 cells treated with BIA (n = 3 independent experiments). (B) Gel filtration assay of HT1080 cells treated with 1.0 μ M BIA. The red line represents the size marker. (C) Immunoblot analysis of anti-HA immunoprecipitate (IP) and whole cell lysate (WCL) of HT1080 cells transiently overexpressing TMBIM6-HA and treated with BIA. (D) Immunoblotting of p-AKT, AKT, and actin in the indicated cell lines following treatment with BIA. (E) PLA between TMBIM6-HA and mTORC2 components or between TMBIM6-HA and RPL19 (red dots) in BIA-treated or non-treated TMBIM6 stably overexpressing HT1080 cells. Bottom, quantification of red dots (n = 5 independent experiments). Scale bar, 20 μ m.

Supplementary Fig. 10

Supplementary Fig. 10 Inhibitory function of BIA in cancer progression. (A) TMBIM6 mRNA levels in various cancer cell lines. mRNA levels in each cells were normalized to the level of β -actin. (B) Cell viability was measured in the indicated concentrations of BIA-treated cancer cells at three days ($n = 3$ independent experiments). (C) Proliferation of cells stably expressing TMBIM6 or the empty vector was analyzed after 1 day of treatment with indicated concentrations of BIA. (D) PLA between RICTOR and the following protein, mTOR, RPL19 and RPS 16 was performed in HT1080 cells with 10 μ M BIA. Scale bar, 20 μ m. Right, quantification of red dots ($n = 5$ independent experiments).

Manuscripts

To identify whether BIA decreases the binding between TMBIM6 and mTORC2, we performed a gel filtration assay in TMBIM6-overexpressing HT1080 cells after treatment with BIA during 24 h. As shown in Fig. 7B, TMBIM6-HA was co-eluted with mTORC2 components (mTOR and RICTOR) and ribosomes (RPL19), whereas the co-elution pattern was delayed in the BIA-treated HT1080 cells, indicating that BIA induces the dissociation of TMBIM6 from mTORC2 and ribosome. Interaction between RICTOR and mTOR, or RICTOR and RPL119, or RICTOR and RPS16 by PLA assay was decreased in HT1080 cells with BIA compared to control cells (Supplementary Fig. 10D). In addition, BIA decreased the binding of TMBIM6 to mTORC2 and inhibited the phosphorylation of AKT (Fig. 7C). Consistently, the phosphorylation of AKT was fully decreased in three breast cancer cell lines, MCF7, MDA-MB-231, and SKBR3 cells (Fig. 7D). To elucidate whether BIA impaired interaction between TMBIM6 and mTORC2 or ribosome, we performed the PLA assay. As shown in Figure 7E, interactions of mTOR, RICTOR, or RPL19 with TMBIM6 were decreased in BIA-treated cells compared to control cells.

7. Overstatements and unsupported evidence: In page 18 the authors state that “BIA is a newly identified TMBIM6 antagonist that disrupts the TMBIM6-mTORC2 interaction, leading to the inhibition of tumor growth even in cases that are resistant to mTOR inhibitors”. However, the authors have not provided evidence suggesting this. They have only shown that BIA decreases tumor growth in vitro and in vivo and that it works in combination with other mTORC2 inhibitors to kill HT1080 cells. They have not shown that these effects are a direct consequence of BIA’s putative role as a TMBIM6 antagonist or as a disruptor of TMBIM6-

mTORC2 interaction. Throughout the manuscript there are several instances where the authors jump from a phenotypic observation to a molecular mechanism whose causal connection to the observation has not been directly demonstrated.

Response: To show these effects are a direct consequence of BIA's putative role as a TMBIM6 antagonist or as a disruptor of TMBIM6-mTORC2 interaction, we added the other data, including PLA analysis. In this study, BIA has emerged as a potential candidate as an antagonist regulating TMBIM6-mTORC2 interaction and its related tumor growth even in cases that are resistant to mTOR inhibitors. Besides, we carefully mentioned these effects and mechanisms of BIA, not jumping to the conclusion from a phenotypic observation throughout this revised manuscript. In the revised version, we mentioned about the point as following,

Supplementary Fig. 11 BIA decreases cell survival. (A) The images of crystal violet staining in HT1080, PANC-1, Capan-1, and MIA PaCa-2 cells after treatment with 10 μM BIA and mTOR inhibitors. Right; quantification of cell viability normalized to control cells. (B) PLA between the indicated proteins (red dots) in BIA or mTOR inhibitors-treated PANC-1 cells. Right, quantification of red dots (n = 5 independent experiments). Scale bar, 20 μm. Data represent mean ± SD. Statistical differences were detected with one-Way ANOVA followed by Tukey's test (A), and two-Way ANOVA followed by Bonferroni's test (B).

Manuscripts

To determine whether BIA is effective against HT1080, PANC-1 pancreatic cancer cells resistant to mTOR inhibitor, and other pancreatic cancer cells including Capan-1 and MIA PaCa-2 cells, we compared the growth of cells treated with BIA to that of cells treated with anti-mTOR inhibitors such as AZD8055, INK128, Omipalisib, OSI-027, and Voxtalisib. Cell viability was reduced to a greater extent by the treatment with BIA as compared to the other agents (Supplementary Fig. 11A). Especially, BIA almost abrogated live cells in PANC-1 cells, which have 30 ~ 40% cell viability by the other mTOR inhibitors, including AZD8055, INK128, Omipalisib, OSI-027, and Voxtalisib. In PLA assay, BIA diminished association between RICTOR and mTOR or between RICTOR and RPL19, but the other anti-mTOR inhibitors did not affect any association in PANC-1 cells (Supplementary Fig. 11B), suggesting that BIA has potential as an effective anticancer agent controlling cancer cell survival although the experiment is only *in vitro* state.

To this end, BIA has emerged as a potential candidate as an antagonist regulating TMBIM6-mTORC2 interaction and its related tumor growth even in cases that are resistant to mTOR inhibitors. Specifically, the characteristics of BIA indicate the dissociation between RICTOR and TMBIM6 through the regulation of TMBIM6-leaky Ca²⁺ and the resultant inhibition of AKT activation impeding cancer formation.

8. Statistics: The authors should review the statistical procedures used throughout the

manuscript. When there are two independent variables (e.g. genotype and time) the authors should use a Two-Way ANOVA followed by post hoc test instead of multiple two-tailed unpaired Student's t-tests. Figures where Two-Way ANOVA should be used include Figures 1E, 2A, 2D, 2E, 2H, 3B, 3F, 4A, 4E, 4H, 8E and 8G. When there is one independent variable but more than two groups are compared, the authors should use a One-Way ANOVA followed by post hoc test. Examples of these include Figures 2I, 2K, 6D and 6E. Finally, the authors should specify p-values as follows: *, $p < 0.05$; **, $p < 0.01$; ***, $p < 0.001$.

Response: In accordance with the reviewer's recommendation, we performed the statistical reanalysis. One-way analysis of variance (ANOVA) with Tukey post hoc test, Two-Way ANOVA followed by Bonferroni post hoc test, and Student's unpaired t-test were performed using Prism v.8 software. Also, *, $p < 0.05$; **, $p < 0.01$; ***, $p < 0.001$, were considered statistically significant. In each case, the statistical test used is indicated, and the number of experiments is stated in the legend of each figure.

Minor comments:

1. For in vitro migration and invasion experiments shown in Figures 2B and C, the authors should also express the data as the percentage of migrating and invading cells and not only as fold changes compared to WT.

Response: In accordance with the reviewer's suggestion, we represent two types of data "fold and percentage of migrating and invading cells in Figures 2B-C and Supplementary Fig 2C-D, respectively".

Figure 2

Supplementary Fig. 2

Supplementary Fig. 2 TMBIM6 regulates tumor growth. (A-B) Proliferation of rescue TMBIM6 expressing in TMBIM6 KO HT1080 (A) and HeLa (B) cells ($n = 3$ independent experiments). (C-D) Images and quantification of migrated cells (C) or invasive cells (D) represented in TMBIM6 KO and WT HT1080. Quantification data represented percentage of WT cells normalized to KO cells ($n = 3$ independent experiments). Scale bars, 100 μm .

2. In Figure 3C the authors should show the blot results of the second mTORC2 target NDRG1 (pT346) in the HT1080 TMBIM6 KO cells reconstituted with TMBIM6-HA.

Response: In addition to reviewer 1's comments, we performed immunoblotting of NDRG1 T346 in TMBIM6 KO HT1080 cells with re-expression of TMBIM6. The updated Figure 3C and corresponding text are given below.

Manuscripts

Immunofluorescence analysis revealed and confirmed that the phosphorylation of AKT was decreased in TMBIM6 KO as compared to that in wild-type (WT) cells (Supplementary Fig. 3D). Consistently, overexpressing TMBIM6 in HeLa cells increased mTORC2 activity (Supplementary Fig. 3E). Reintroducing TMBIM6 into TMBIM6 KO HT1080 cells restored AKT Ser473 and NDRG1 Ser939 phosphorylation comparably (Fig. 3C).

3. In page 7, the authors describe the effects of insulin referencing supplementary figure 2D and 2E. This is not correct; it should read Supplementary Figure 3D and 3E respectively.

Response: We apologized incorrect referencing Supplementary Figure 2D and 2E in the original text. In the revised version, we changed the original supplementary Fig. 3D and Supplementary Fig. 3E to Supplementary Fig. 3F and 3G, respectively.

4. There are many misplaced conclusions throughout the manuscript. For example, in page 11 the authors say that "...TMBIM6 interacts with mTORC2 and ribosomes and that this interaction is important for the kinase activity of mTORC2". However, the effects of TMBIM6-mTORC2 interaction in AKT phosphorylation are explored in the following paragraph (Figure 5H).

Response: As per the reviewer's suggestion, that specific sentence was rearranged in the following Figure 5H.

Manuscripts

The phosphorylation of AKT Ser473 was also decreased in RPL19-non-associated TMBIM6 mutants (Fig. 5H, I). Taken together, these data indicate that TMBIM6 interacts with mTORC2 and ribosomes and that this interaction is important for the kinase activity of mTORC2.

5. How does calcium leak from the ER increase mTORC2 assembly efficiency? The authors

should discuss this in more detail.

Response: This part is very important part for this study. D213A mutant role and the presence of BAPTA-AM control the mTORC2 assembly. In addition, the PLA between mTOR and rictor and between mTOR and RPL19 and RPS16 was shown as a positive evidence showing that calcium leak from the ER increased mTORC2 assembly efficiency. According to the reviewer's comment, we have updated the following part in the revised discussion.

Manuscripts

Similar to mTORC1 studies⁵²⁻⁵⁴, we found that mTORC2 activity is sensitive to inhibition by BAPTA-AM (Figure 6A-B, supplementary Fig. 7A), suggesting that intracellular calcium is required for mTORC2 activation. In a recent study about the correlation of Ca²⁺ channel and mTORC2, mTORC2 negatively regulates Mid1, an ER/plasma membrane (PM)-localized calcium channel regulatory protein. Decreased signaling of TORC2 and its downstream target protein kinase, Ypk1, induced Mid1 activation⁵⁵. The mTORC2 signaling regulates the Ca²⁺-associated Mid as downstream effector under amino acid starvation, whereas the Ca²⁺-leaky TMBIM6 first affects mTORC2 activation through the releasable Ca²⁺ from TMBIM6 under basal condition. Our point is not general cytosolic Ca²⁺ but the local Ca²⁺ from the Ca²⁺ leaky protein, TMBIM6, which is also abrogated in the presence of BAPTA-AM, not BAPTA, a not cell permeable Ca²⁺ chelating agent.

The precise mechanism of regulation of mTORC2 by calcium through TMBIM6 Ca²⁺ leaky characteristics is distinct in our model. First, we demonstrated that the ER leaky calcium plays a unique and critical role in mTORC2 activation and the resultant AKT in mammalian cells. Second, D213A mutant had no effect on the interaction between TMBIM6 and RICTOR, ruling out the involvement of the local Ca²⁺ in the direct binding with the mTORC2 subunit protein. However, we also found that D213A mutant had a strong controlling effect on the interaction between mTOR and RICTOR or between RICTOR and the ribosomal proteins, RPL19 and RPS16 suggesting the involvement of the local Ca²⁺ in the assembly of mTORC2 and further the recruitment of ribosomal proteins also.

Reviewer #4 (Remarks to the Author) (expertise: cancer, zebrafish model) :

This is a continuous work of the authors' previous claims that TMBIM6/BI-1 promotes tumor growth and progression (ref 9). In this study, the authors provide new evidence of potential signaling events of the TMBIM6/BI-1-mTORC2-AKT axis in different cancer types. First, they show that in several cancer types TMBIM6/BI-1 expression is elevated, which is largely confirmed by GCTA analysis. Using in vitro KO technology, they show that deletion of TMBIM6/BI-1 in cancer cells retards cancer cell proliferation, migration, and tumor formation. They then defined signaling pathways that potential involved in these tumorigenic activities. Finally, they screened a chemical library to identify a potential inhibitor BIA, which inhibits tumor growth in mice and in fish.

Comments:

1) This work covers almost all respects of cancer development, including cancer cell proliferation, migration, survival, tumor formation, and metastasis. From the provided data, it is hard to believe that TMBIM6/BI-1 is a master regulator of cancer development. In particular, the characterized signaling events do not support TMBIM6/BI-1 as a master

regulator. The authors should focus on a particular signaling and activity to obtain an in-depth mechanistic insight, but not a wikipedia-associated cancer development.

Response: Various oncogenic pathways are related to mTOR signaling. Among the mTOR complexes, mTORC1 function is hyperactivated in up to 70% of all human tumors (Forbes et al., 2011; Xie et al., 2016). mTORC2 is linked to PI3K signaling, also highly activated in many tumor cells. The mechanism for mTORC1 activation has been well established, but that for mTORC2 activation is not well understood.

In this study, it is aimed that the role of TMBIM6 is ultimately linked to mTORC2 and AKT, a general cancer master signaling. However, we agree with the reviewer's comment. We don't want to show the role of TMBIM6 as a master regulator in cancer progression. Through the developed mTOR inhibitors and the other anti-cancer agents' application to the resistant cancer models (Supplementary Figure 11), newly targeted molecule/protein-protein interaction would be a strong target toward the resistant cancer cases, a main theme in this study.

The unique point is that TMBIM6 is a Ca^{2+} regulator controlling local area surrounding ER, so that the binding affinity of some proteins such as mTORC2 etc. can be increased. Due to the local Ca^{2+} environment, the interaction of TMBIM6 with mTORC2 and the resultant AKT activation contributes to the cancer development, a summary in this study. In this revised version, we did our best to focus on particular signaling and activity to obtain an in-depth mechanistic insight; local Ca^{2+} -based mTORC2 and ribosome recruitment and the resultant AKT activation. We appreciate returning comments.

2) The TCGA data and their own experimental findings do not completely match. What about lung cancer, pancreatic cancer, and prostate cancer in TCGA? If TMBIM6/BI-1 is also highly expressed in these cancers, but not associated with poor survival, what does this mean? I am almost certain that some cancer types express high levels of TMBIM6/BI-1, but lack clinical correlation. The authors should not only choose the results favor to your findings.

Response: According to the reviewer's comments, we performed survival analysis about lung cancer, pancreatic cancer, and prostate cancer. TMBIM6 is highly expressed in lung cancer (502.54 TPM for tumor, 330.77 TPM for normal), pancreatic cancer (379.38 TPM for tumor, 115.42 TPM for normal), and prostate cancers (505.34 TPM for tumor, 258.46 TPM for normal) using GEPIA2. In survival analysis, lung and pancreatic cancer were also associated with poor survival. In prostate cancer, the lists of cancers in OncoLnc database were not included, so we could not examine the correlation between the TMBIM6 expressions and prostate cancer survival. Also, we were not confirmed in GEPIA2 analysis. However, the group with high expression of TMBIM6 showed poor survival in either altered or and unaltered group using all datasets in cBioPortal, suggesting that high expression of TMBIM6 may affect patient survival.

We need to further study for prostate cancer.

In the revised version, we added the data in Figure 1G and supplementary Fig. 1A-B.

Figure 1

Supplementary Fig. 1

3) The zebrafish cancer model is useless in this experimental setting. Are BIA-treated cancer

cells dead in the fish body? How do they discriminate living cancer cells from dead cancer cells? How do they know that BIA is active in fish body?

Response: Zebrafish model was applied to utilize the transparent embryo to visualize the migration pattern or behavior of the injected cancer cells. Later, BIA treatment prevented the migration of injected cancer cells to other parts, whereas in control tumor cells migrated away from the primary site of injection. Our objective from this experiment was to visualize the migration behavior to confirm the efficacy of BIA in preventing migration of tumor cells, thereby effective in the prevention of spread. Moreover, developing embryos injected with cancer cells were grown in water containing 2.0 μM BIA, which itself makes sure that the activity of BIA.

Several other experiments in this study sufficiently prove the cause thus, reviewers might have thought that zebrafish experiments as unnecessary. However, to confirm migration behaviors of cancer cells with visualization zebrafish model is the possible solution. Hence, zebrafish model was used in the study.

4) BIA looks like a non-specific chemical compound as most chemical compounds do. These data can only be used as indirect supportive data.

Response: To show the specificity, we added the application of BIA to the TMBIM6 KO cells in the revised version. Compared with the WT cells, the BIA had no effect at the TMBIM6 KO cells, indicating the specificity of BIA as an inhibitor of TMBIM6. As a screening approach for the development of TMBIM6 antagonist, BIA has emerged as a potential candidate as an antagonist regulating TMBIM6-mTORC2 interaction and its related tumor growth even in cases that are resistant to mTOR inhibitors. To show these effects are a direct consequence of BIA's putative role as a TMBIM6 antagonist or as a disruptor of TMBIM6-mTORC2 interaction, we added the other data, including PLA analysis.

5) The manuscript needs language editing.

Response: As per the reviewer's advice, we get editing help from someone with full professional proficiency in English.

Additional changes made in the revised manuscript are as follows:

1) In method section, the R1 primer sequence for confirming TMBIM6 KO cells by CRISPR/Cas9 genome editing was incorrect, so that we have been changed to R1, 5'-TCAATCCTGCCTCTCCTGAT-3'.

2) We apologize missing explanation about some method, and the updated text is as below.

Live cell imaging

Live cell imaging with BI-GCaMP3 and G-CEPIAer was performed using LSM 880 microscopy. Briefly, 2×10^5 HT1080 cells stably expressing GCaMP3-ML1 or G-CEPIAer were cultured in a 35-mm confocal dish. Changes in fluorescence levels were monitored for 20 min upon addition of BIA in Ca^{2+} -free external solution containing 145 mM NaCl, 5 mM

KCl, 3 mM MgCl₂, 10 mM glucose, 1 mM EGTA, and 20 mM HEPES (pH 7.4). The intensity of fluorescence was measured using ZEN software.

Human phospho-kinase array

Human Phospho-kinase arrays were performed according to manufacturer's instructions (ARY003B, R&D Systems, Minneapolis, MN, USA). The quantification of pixels was performed using Fiji ImageJ software.

3) We apologize missing to explanation about phospho-kinase profiling in Figure 3A. The updated text is below.

Manuscripts

To evaluate the signaling protein molecule, which regulates the cancer progression in WT and TMBIM6 KO HT1080 cells, we have performed phospho-kinase profiling. The expression of phosphorylated AKT, proline-rich Akt substrate of 40 kDa (PRAS40), mTOR, Glycogen synthase kinase-3 (GSK3- α/β), and lysine deficient protein kinase 1 (WNK1) were decreased in TMBIM6 KO HT1080 cells (Fig. 3A). Since PRAS40, GSK3- α/β , and WNK1 are the known substrates of AKT, we next investigated whether mTORC2, upstream regulator of AKT, is altered in TMBIM6 KO cells.

4) We have added four authors who helped us to conduct experiments that are necessary for the revision. New coauthors, Suvarna H Pagire, screened the new small-molecule inhibitor with Jin Hee Ahn. Hyun Ju Yoo performed the metabolite analysis such as glycolysis, TCA, PPP, and HBP. Jaeseok Han and Duckgwee Lee performed the ribosome profiling assay. Kyung-Woon Kim performed the T4 phage display screening.

5) In revised manuscripts, we added some sentences to support our explanation.

- In above our results, since TMBIM6 regulates mTORC2 activation through ER Ca²⁺ release, we measured TMBIM6-GCaMP3 green fluorescence for real-time by application of 10 μ M BIA to identify whether BIA inhibits Ca²⁺ release from TMBIM6.

- To investigate the role of TMBIM6 in the growth of tumor cells in animals, we subcutaneously injected TMBIM6 WT and KO HT1080 cells into the left and right flanks, to immune-compromised mice and monitored tumor growth during 27 days (Supplementary Fig. 2E-F).

References

- Anaya, J. (2016). OncoLnc: linking TCGA survival data to mRNAs, miRNAs, and lncRNAs. *Peerj Comput Sci*.
Bultynck, G., Kiviluoto, S., Henke, N., Ivanova, H., Schneider, L., Rybalchenko, V., Luyten, T., Nuyts, K., De Borggraeve, W., Bezprozvanny, I., *et al.* (2012). The C terminus of Bax inhibitor-1 forms a Ca²⁺-permeable channel pore. *J Biol Chem* 287, 2544-2557.
Chae, H.J., Kim, H.R., Xu, C., Bailly-Maitre, B., Krajewska, M., Krajewski, S., Banares, S., Cui, J.,

- Digicaylioglu, M., Ke, N., *et al.* (2004). BI-1 regulates an apoptosis pathway linked to endoplasmic reticulum stress. *Mol Cell* 15, 355-366.
- Chang, Y., Bruni, R., Kloss, B., Assur, Z., Kloppmann, E., Rost, B., Hendrickson, W.A., and Liu, Q. (2014). Structural basis for a pH-sensitive calcium leak across membranes. *Science* 344, 1131-1135.
- Forbes, S.A., Bindal, N., Bamford, S., Cole, C., Kok, C.Y., Beare, D., Jia, M., Shepherd, R., Leung, K., Menzies, A., *et al.* (2011). COSMIC: mining complete cancer genomes in the Catalogue of Somatic Mutations in Cancer. *Nucleic Acids Res* 39, D945-950.
- Kim, H.K., Lee, G.H., Bhattarai, K.R., Lee, M.S., Back, S.H., Kim, H.R., and Chae, H.J. (2020). TMBIM6 (transmembrane BAX inhibitor motif containing 6) enhances autophagy through regulation of lysosomal calcium. *Autophagy*, 1-18.
- Lee, G.H., Kim, D.S., Kim, H.T., Lee, J.W., Chung, C.H., Ahn, T., Lim, J.M., Kim, I.K., Chae, H.J., and Kim, H.R. (2011). Enhanced lysosomal activity is involved in Bax inhibitor-1-induced regulation of the endoplasmic reticulum (ER) stress response and cell death against ER stress: involvement of vacuolar H⁺-ATPase (V-ATPase). *J Biol Chem* 286, 24743-24753.
- Lisbona, F., Rojas-Rivera, D., Thielen, P., Zamorano, S., Todd, D., Martinon, F., Glavic, A., Kress, C., Lin, J.H., Walter, P., *et al.* (2009). BAX inhibitor-1 is a negative regulator of the ER stress sensor IRE1alpha. *Mol Cell* 33, 679-691.
- Tang, Z., Kang, B., Li, C., Chen, T., and Zhang, Z. (2019). GEPIA2: an enhanced web server for large-scale expression profiling and interactive analysis. *Nucleic Acids Res* 47, W556-W560.
- Xie, J., Wang, X., and Proud, C.G. (2016). mTOR inhibitors in cancer therapy. *F1000Res* 5.

REVIEWERS' COMMENTS:

Reviewer #1 (Remarks to the Author):

The authors have done an excellent job responding to reviewer criticisms. I have no further comments.

Reviewer #2 (Remarks to the Author):

The authors have thoroughly responded to this reviewer's comments and suggestions. They have added additional data to support the role of TMBIM6 in mTORC2 assembly and association with ribosomes and provided clarifications as requested.

Reviewer #3 (Remarks to the Author):

NCOMMS-19-31425A (revision 2)

Comments to the authors,

My main concerns with the paper were (1) the model of choice and generality of the results and the (2) possibility of CRIPR/cas9 off-target effects, since the authors only showed results derived from a single TMBIM6 KO clone and no scramble controls. In the revised version of this manuscript, the authors have completely addressed these concerns. First, they have shown that high levels of TMBIM correlates with poorer prognosis in sarcoma, They have also generated two different TMBIM6 KO clones (together with the scrambled CRISPR/Cas9 control) in HeLa cells and further showed that these cells exhibited decreased proliferation, impaired mTORC2 assembly and decreased tumor growth in a xenograft mouse model. These experiments are very well performed and clearly support the role of TMBIM6 on mTORC assembly and tumor progression. Second, the authors have also added an additional TMBIM6 KO clone and scrambled control to their experiments in HT1080 cells, decreasing the probability of off-target effects.

Additionally, the requested missing controls for the total protein levels of the mTORC complex and the ER localization of TMBIM6-HA and the mutant D213A have also been performed.

Another contentious issue was regarding the topology of TMBIM6 within the ER membrane. Here, the authors have performed an additional experiment suggesting that TMBIM6 has a six-transmembrane topology and have acknowledged previous reports suggesting a seven transmembrane topology for TMBIM6. I still believe that the authors could, if they are willing, provide a more balanced discussion regarding this issue.

Overall, the authors have greatly improved and expanded their work and satisfactorily addressed all of my concerns. This revised version of the manuscript should be ready for publication in Nature Communications.

Suggestion:

1. The authors should discuss the effects of BIA on cell viability presented in Supplementary Figure 10B and how they relate to the cell proliferation data shown in Figure 7A. It is clear that the concentrations of BIA that inhibited proliferation are also killing the cells. Does inhibiting TMBIM6 with BIA results in a dual effect, disassembling mTORC and decreasing proliferation on one hand and inhibiting additional TMBIM6 functions required for cell survival under stress?

Reviewer #4 (Remarks to the Author):

The authors have reasonably addressed my previous concerns, although some remain. The authors should cite some of the important zebrafish work related to their models:

Nat Protoc. 2010 Dec;5(12):1911-8. doi: 10.1038/nprot.2010.150. Epub 2010 Nov 4.

PMID: 21127485

Proc Natl Acad Sci U S A. 2009 Nov 17;106(46):19485-90. doi: 10.1073/pnas.0909228106. Epub 2009 Nov 3. PMID: 19887629

Cancer Res. 2015 Jan 15;75(2):306-15. doi: 10.1158/0008-5472.CAN-14-2819. Epub 2014 Dec 9.

PMID: 25492861

Clin Cancer Res. 2017 Aug 15;23(16):4769-4779. doi: 10.1158/1078-0432.CCR-17-0101. Epub 2017 Apr 18.

PMID: 28420724

On the basis of their thorough revision and reasonable explanation, I recommend publication of this work in NC. Congratulations!

REVIEWERS' COMMENTS:

Reviewer #1 (Remarks to the Author):

The authors have done an excellent job responding to reviewer criticisms. I have no further comments.

Response: We thank the reviewer for his/her friendly decisions.

Reviewer #2 (Remarks to the Author):

The authors have thoroughly responded to this reviewer's comments and suggestions. They have added additional data to support the role of TMBIM6 in mTORC2 assembly and association with ribosomes and provided clarifications as requested.

Response: We thank the reviewer for his/her friendly decisions.

Reviewer #3 (Remarks to the Author):

NCOMMS-19-31425A (revision 2)

Comments to the authors,

My main concerns with the paper were (1) the model of choice and generality of the results and the (2) possibility of CRIPR/cas9 off-target effects, since the authors only showed results derived from a single TMBIM6 KO clone and no scramble controls. In the revised version of this manuscript, the authors have completely addressed these concerns. First, they have shown that high levels of TMBIM6 correlates with poorer prognosis in sarcoma, They have also generated two different TMBIM6 KO clones (together with the scrambled CRISPR/Cas9 control) in HeLa cells and further showed that these cells exhibited decreased proliferation, impaired mTORC2 assembly and decreased tumor growth in a xenograft mouse model. These experiments are very well performed and clearly support the role of TMBIM6 on mTORC assembly and tumor progression. Second, the authors have also added an additional TMBIM6 KO clone and scrambled control to their experiments in HT1080 cells, decreasing the probability of off-target effects.

Additionally, the requested missing controls for the total protein levels of the mTORC complex and the ER localization of TMBIM6-HA and the mutant D213A have also been performed.

Another contentious issue was regarding the topology of TMBIM6 within the ER membrane. Here, the authors have performed an additional experiment suggesting that TMBIM6 has a six-transmembrane topology and have acknowledged previous reports suggesting a seven transmembrane topology for TMBIM6. I still believe that the authors could, if they are willing, provide a more balanced discussion regarding this issue.

Overall, the authors have greatly improved and expanded their work and satisfactorily addressed all of my concerns. This revised version of the manuscript should be ready for publication in Nature Communications.

Suggestion:

1. The authors should discuss the effects of BIA on cell viability presented in Supplementary Figure 10B and how they relate to the cell proliferation data shown in Figure 7A. It is clear that the concentrations of BIA that inhibited proliferation are also killing the cells. Does inhibiting TMBIM6 with BIA results in a dual effect, disassembling mTORC and decreasing proliferation on one hand and inhibiting additional TMBIM6 functions required for cell survival under stress?

Response: We thank the reviewer for his/her friendly decisions or comments. The inhibitory action of TMBIM6 functions including directly binding with mTORC2 and association of ribosome through calcium leaky by compound such as BIA caused decreasing of AKT activation, thereby reduced cell proliferation by disrupting cellular metabolism, increased susceptibility under stress, result in killing the cells.

To make it clear, we have now added the following information in main text.

Taken together, these data demonstrate that BIA reduced cell proliferation by inducing the dissociation of TMBIM6 from mTORC2 and ribosome, result in cell death.

Reviewer #4 (Remarks to the Author):

The authors have reasonably addressed my previous concerns, although some remain. The authors should cite some of the important zebrafish work related to their models:

Nat Protoc. 2010 Dec;5(12):1911-8. doi: 10.1038/nprot.2010.150. Epub 2010 Nov 4.

PMID: 21127485

Proc Natl Acad Sci U S A. 2009 Nov 17;106(46):19485-90. doi: 10.1073/pnas.0909228106.

Epub 2009 Nov 3. PMID: 19887629

Cancer Res. 2015 Jan 15;75(2):306-15. doi: 10.1158/0008-5472.CAN-14-2819. Epub 2014 Dec 9.

PMID: 25492861

Clin Cancer Res. 2017 Aug 15;23(16):4769-4779. doi: 10.1158/1078-0432.CCR-17-0101.

Epub 2017 Apr 18.

PMID: 28420724

On the basis of their thorough revision and reasonable explanation, I recommend publication of this work in NC. Congratulations!

Response: We thank the reviewer for his/her friendly decisions, and added all of the suggests references mentioned by the reviewer.